# MesaNet: Sequence Modeling by Locally Optimal Test-Time Training

**Johannes von Oswald**[1,*]**, Nino Scherrer**[1,*]**,**

**Seijin Kobayashi**[1]**, Luca Versari**[1]**, Songlin Yang**[3]**, Maximilian Schlegel**[1]**,
Kaitlin Maile**[1]**, Yanick Schimpf**[1]**, Oliver Sieberling**[3]**, Alexander Meulemans**[1]**,
Rif A. Saurous**[1]**, Guillaume Lajoie**[1]**, Charlotte Frenkel**[1]**, Razvan Pascanu**[2]**,
Blaise Agüera y Arcas**[1] **and João Sacramento**[1]

[1] **Google, Paradigms of Intelligence Team,** [2] **Google DeepMind,** [3] **MIT CSAIL**

## Abstract

Sequence modeling is currently dominated by causal transformer architectures that use softmax self-attention. Although widely adopted, transformers require scaling memory and compute linearly during inference. A recent stream of work linearized the softmax operation, resulting in powerful recurrent neural network (RNN) models with constant memory and compute costs such as DeltaNet, Mamba or xLSTM. These models can be unified by noting that their recurrent layer dynamics can all be derived from an in-context regression objective, approximately optimized through an online learning rule. Here, we join this line of work and introduce a numerically stable, chunkwise parallelizable version of the recently proposed Mesa layer (von Oswald et al., 2024), which could only run sequentially in time and was therefore not scalable. This layer again stems from an in-context loss, but which is now minimized to optimality at every time point using a fast conjugate gradient solver. Through an extensive suite of experiments study up to the billion-parameter scale, we show that optimal test-time training enables reaching lower language modeling perplexity and higher downstream benchmark performance than previous RNNs, especially on tasks requiring long context understanding. This performance gain comes at the cost of additional flops spent during inference time. Our results are therefore intriguingly related to recent trends of increasing test-time compute to improve performance – here by spending compute to solve sequential optimization problems within the neural network itself.

## 1 Introduction

While Transformers dominate sequence modeling, their per-token computational and memory requirements scale linearly with sequence length during inference. This limitation motivates the development of efficient recurrent neural networks (RNNs) with constant complexity, particularly for autoregressive tasks like language modeling. Recent progress has focused on fast weight programming layers, which process a given sequence by representing and learning a linear model in their activations (Schmidhuber, 1992; Schlag et al., 2021a; Yang et al., 2024c; Dao & Gu, 2024). Such 'fast weights' undergo one learning step whenever the input sequence advances, following simple Hebbian (Hebb, 1949) or error-correcting (delta) rules (Widrow & Hoff, 1960). Both rules correspond to gradient descent on a suitable quadratic loss function, measured on the latest input.

Here, we take this concept one step further, and design an optimal fast weight programming layer. Following previous related work, we consider linear fast weight models, and measure how well a given context is modeled using a quadratic loss. However, instead of gradually learning through gradient descent, we design a layer that always responds with the optimal fast weights, which achieve minimum loss on all data seen so far. This allows retaining past information while adapting to new evidence quickly as a sequence unfolds. Our work builds off the recent recurrent Mesa layer (von Oswald et al., 2024), proposing a version of this layer that is parallelizable leveraging matrix multiplication accelerators, numerically stable, and that allows for context-dependent forgetting. Moreover, the layer dynamically adapts its computational cost at test time to the sequence at hand. This is because the

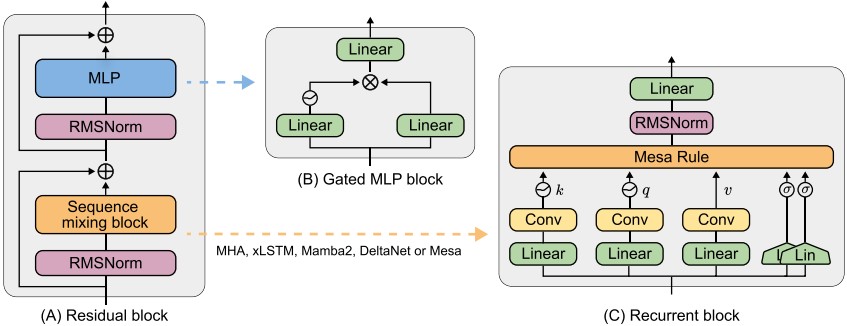

**Figure 1: Model Architecture of the MesaNet.** **(A)** We adopt the widespread decoder-only transformer architecture (Touvron et al., 2023) stacking $N$ residual blocks of a channel mixing (B) and sequence mixing (C) components. **(B)** Channel mixing is a vanilla SwiGLU MLP. **(C)** Sequence mixing is performed by the Mesa layer. From its inputs, it generates keys, queries and values as well as input and forget strengths. These are then processed according to the Mesa Rule (Equation 7). We compare the MesaNet to models which share the exact same architecture and only change the sequence mixing rule to multi-head-attention (MHA), xLSTM, Mamba2 or (Gated) DeltaNet.

layer introduced here explicitly invokes an external solver, for which the number of iterations required to reach a given stopping criterion differs across sequences. We summarize our contributions below:

- **A novel Mesa layer which is parallelizable over sequence length and flexibly allocates test-time computation**: We adapt the previously proposed Mesa layer (von Oswald et al., 2024) to allow for chunkwise parallel training. We leverage an equivalence of the conjugate gradient (CG) method over multiple time steps with gated linear self-attention, which allows using established hardware-efficient training (Yang et al., 2024a). During inference, the layer reallocates test-time compute dynamically as different sequences lead to varying CG iterations to reach a stopping criterion, allowing to trade off test-time compute and performance.

- **The MesaNet is a strong language model**: We train 140M, 440M and 1B parameters MesaNets, see Figure 1, on the SlimPajama dataset (Soboleva et al., 2023). On all of these scales, the MesaNet reaches lower validation perplexity compared to models such as Mamba2 (Gu & Dao, 2024), xLSTM (Beck et al., 2024), DeltaNet (Yang et al., 2024c), Gated DeltaNet (Yang et al., 2024a) and Transformers (Vaswani et al., 2017) with the same base architecture.

- **In-depth analyses of modern RNNs including MesaNet**: Intriguingly, we find that while reaching the same or better perplexity on language modeling, all RNN models reduce perplexity remarkably differently, namely focus on early tokens in the sequence while transformers excel at later tokens. We further disentangle downstream language benchmarks according to their need for *global* or only *local* language modeling, through controlled Sliding-Window Attention ablations. We find that MesaNet outperforms all modern RNNs on global reasoning, in-context learning & in-context recall benchmarks, but unsurprisingly still lack behind Transformers in in-context recall.

## 2 A PARALLELIZABLE MESA LAYER

We consider autoregressive sequence modeling tasks where the objective is to predict element $e_{t+1} \in \mathbb{R}^{n_e}$ given a sequence of token embeddings $e = (e_t)_{t=1}^T$. At present, autoregressive sequence modeling is dominated by architectures based on the causally-masked softmax self-attention layer, whose token updates $e_t \leftarrow e_t + \Delta e_t^{\text{sa}}$ follow the rule $\Delta e_t^{\text{sa}} = \sum_{h=1}^H P_h V_{h,t} \, \alpha(K_{h,t}^\top q_{h,t})$, where $q_{h,t} = W_{h,q} e_t \in \mathbb{R}^{n_a}$ is referred to as a query, each column $k_{h,t'} = W_{h,k} e_{t'} \in \mathbb{R}^{n_a}$ of matrix $K_{h,t} \in \mathbb{R}^{n_a \times t}$ as a key, and each column $v_{h,t'} = W_{h,v} e_{t'} \in \mathbb{R}^{n_v}$ of matrix $V_{h,t} \in \mathbb{R}^{n_v \times t}$ as a value; in this paper, we follow the convention that vectors are column vectors. The parameters of this layer are the matrices $\{(P_h, W_{h,q}, W_{h,k}, W_{h,v})\}_{h=1}^H$ for all $H$ heads; for notational simplicity, we omit positional encodings and absorb bias terms, and assume here for conciseness that all heads are equally sized. The function $\alpha$ applied to vector $a \in \mathbb{R}^t$ returns an attention weight vector: in the standard transformer, $\alpha(a)_i = \text{softmax}(a)_i := (\sum_{t'=1}^t \exp(a_{t'}))^{-1} \exp(a_i)$ (Vaswani et al., 2017). Since each head is processed independently and only interacts through the summation in $\Delta e_t^{\text{sa}}$, for simplicity we drop the head index $h$ and the projection matrix $P$ in what follows.

**Linear self-attention and test-time training.** We focus on the case where $\alpha$ is the identity function. This yields a *linear* attention layer (Schmidhuber, 1992), which as we will see next turns out to be a linear RNN (Katharopoulos et al., 2020):

$$\Delta e_t^{\text{lsa}} = \Phi_t q_t. \tag{1}$$

Unlike its softmax counterpart, linear attention can be implemented recurrently, by maintaining and updating a matrix-valued state $\Phi \in \mathbb{R}^{n_v \times n_a}$ according to the linear dynamics

$$\Phi_t = \gamma_t \Phi_{t-1} + \beta_t v_t k_t^T. \tag{2}$$

Above, we add forget gates $\gamma_t$ and input gates $\beta_t$ which have been shown to improve performance (Yang et al., 2024a). Both are usually a function of the current input $e_t$, like queries, values and keys, but bounded within $[0, 1]$. Importantly, and in contrast to softmax self-attention, linear attention only requires constant memory and compute to predict the next token. As we review below and more extensively in Appendix A, a series of recent high-performance models (e.g., Gu & Dao, 2024; Peng et al., 2023; Beck et al., 2024; Schlag et al., 2021a; Yang et al., 2024c;a) can be cast into the same basic linear self-attention model (equation 1) using variations of equation 2.

Such modern RNNs can also be seen from the unifying perspective of test-time training (Schlag et al., 2021a; Liu et al., 2025; von Oswald et al., 2024; Wang et al., 2025; Behrouz et al., 2025b). Under this view, the key-value linear map $\Phi_t : \mathbb{R}^{n_a} \to \mathbb{R}^{n_v}$ introduced in equation 1 is *learned* from the data in context $e_{1:t}$. Let us introduce a time-varying loss, from which we will derive a gradient-based dynamics for $\Phi$:

$$L_t(\Phi) = l_t(\Phi) + \frac{1}{2} \text{Tr}(\Phi \Lambda_t \Phi^\top). \tag{3}$$

Above, $l_t$ measures the *instantaneous* loss incurred at the current time step, and the second term acts as a regularizer with strength controlled by a symmetric $n_a \times n_a$ matrix $\Lambda_t$. Now, setting $l_t(\Phi) = l_t^{\text{Hopfield}}(\Phi) := -v_t^\top \Phi k_t$ and $\Lambda_t = \frac{1-\gamma_t}{\beta_t} I$, and letting $\Phi$ evolve through online gradient descent, $\Phi_t = \Phi_{t-1} - \beta_t \nabla_\phi L_t(\Phi_{t-1}) = \gamma_t \Phi_{t-1} + \beta_t v_t k_t^T$, we recover gated linear attention (equation 2). In passing, we have also connected modern linear attention to classical associative memory models (Schlag et al., 2021a): $l_t^{\text{Hopfield}}$ is the energy function that governs continuous-state Hopfield networks, and $\Phi$ is learned through Hebb's associative rule (Hopfield, 1984; Hertz et al., 1991). If we take instead the squared error loss $l_t(\Phi) = l_t^{\text{sq-err}}(\Phi) := \frac{1}{2} \|v_t - \Phi k_t\|^2$, we recover DeltaNet (Schlag et al., 2021a; Yang et al., 2024c;a), which learns a linear model with the online delta rule (Widrow & Hoff, 1960). Recent work has extended the DeltaNet to perform mini-batch updates, and to perform gradient updates on a 1-hidden-layer MLP (Sun et al., 2025), and Titans adds momentum to the mini-batched gradient update (Behrouz et al., 2024). We return to this point in Appendices A and B, where we discuss additional related work from the viewpoint of test-time regression, and derive in more detail the update rules above.

**The Mesa layer: optimal test-time regression.** In this work, we revisit the recently proposed Mesa layer (von Oswald et al., 2024), also referred to as an intention layer in the context of non-autoregressive models (Garnelo & Czarnecki, 2023). This layer again updates tokens according to the linear self-attention rule (equation 1) but now defines the linear map $\Phi_t$ as the solution of a test-time optimization problem, where a symmetric positive definite matrix $\Lambda_t \in \mathbb{R}_+^{n_k \times n_k}$ controls the strength of a quadratic regularizer:

$$\hat{\Phi}_t^{\text{mesa}} = \arg\min_\Phi \mathcal{L}_t(\Phi), \qquad \text{with} \qquad \mathcal{L}_t(\Phi) = \frac{1}{2} \sum_{t'=1}^t \zeta_{tt'} \|v_{t'} - \Phi k_{t'}\|^2 + \frac{1}{2} \text{Tr}(\Phi \Lambda_t \Phi^\top). \tag{4}$$

In all our experiments, we take a static, diagonal regularizer, with $\Lambda_t = \Lambda \ \forall_t$ and $\Lambda_{ii} > 0$. Above, the cumulative forget factor $\zeta_{tt'} = \mathbb{1}_{t \geq t'} \prod_{s=t'+1}^t \gamma_s$ causally weighs the contribution of past losses until the present ($t' = 1, \ldots, t$), taking into account the forget factors $\gamma_{t'} \in [0, 1]$ so far. The output $\Delta e_t^{\text{mesa}}$ of the Mesa layer depends on the (unique) solution $\hat{\Phi}_t^{\text{mesa}}$, which can be expressed in closed form:

$$\Delta e_t^{\text{mesa}} = \hat{\Phi}_t^{\text{mesa}} q_t = \left( \sum_{t'=1}^t \zeta_{tt'} v_{t'} k_{t'}^\top \right) \left( \sum_{t'=1}^t \zeta_{tt'} k_{t'} k_{t'}^\top + \Lambda \right)^{-1} q_t \tag{5}$$

$$= G_t (H_t + \Lambda)^{-1} q_t. \tag{6}$$

We compute $\hat{\Phi}_t^{\text{mesa}}$ step by step in Appendix D.

The Mesa layer differs from the test-time training models reviewed above in two key ways. First, instead of considering an instantaneous loss measured only at the current input $e_t$ as in equation 3, the Mesa layer optimizes the *cumulative* regularized squared-error loss taking into account all data $e_{1:t}$ so far. While at first this may seem impossible to achieve under a constant memory requirement, the Mesa layer circumvents the need to explicitly keep past tokens in memory (as in softmax self-attention) and exploits the fact that $\mathcal{L}_t$ is a quadratic function of $\Phi$ (Gauss, 1821). Second, instead of taking a single gradient descent step, the Mesa layer learns $\Phi$ to optimality at every time point. We note that the related Longhorn model (Liu et al., 2025) also derives a recurrent layer via the minimization of a quadratic loss, but its loss is evaluated only on the latest input as in equation 3, yielding a variant of DeltaNet. We further note that concurrent work (Atlas; Behrouz et al., 2025a) corresponds to a sliding-window variant of the Mesa layer, while also allowing the model to be optimized at test-time to be nonlinear, as in (Sun et al., 2025). We present the update rules and test-time objective functions of these two related works in Appendix B.

The Mesa layer is the optimal (in the squared-error sense) linear associative memory (Kohonen & Ruohonen, 1973), and it can store a new association instantaneously (one-shot), whereas DeltaNet requires in general multiple pattern presentations to reduce memorization error (Hertz et al., 1991). This fast learning property of the Mesa layer can be further understood by recasting it as a second-order online learner (cf. Appendix H); DeltaNet only uses first-order derivative information to learn.

Von Oswald et al. (2024) proposed to determine $\hat{\Phi}_t^{\text{mesa}}$ following classical recursive least-squares. Although computationally attractive at inference, we now stress two shortcomings of this approach. First, forgetting ($0 \leq \gamma_t < 1$) leads to numerical instabilities, and requires a regularization term $\Lambda$ that decays exponentially with time. Second, this original version of the layer is not parallelizable, and it therefore heavily underutilizes current matrix-matrix multiplication accelerators such as GPUs and TPUs during training. We explain this in detail in Appendix H.

**A new parallelizable Mesa layer with adaptive forgetting and regularization.** To overcome these issues, we propose a novel parallelizable version of the Mesa layer which allows for dynamic forgetting. Instead of computing $\hat{\Phi}_{h,t}^{\text{mesa}}$ recurrently, we solve a linear system of equations in parallel, for each query $q_t$:

$$\Delta e_t^{\text{mesa}} = G_t(H_t + \Lambda)^{-1}q_t = G_t\text{linsolve}(H_t + \Lambda, q_t). \tag{7}$$

The equation above can be computed by maintaining and updating two state variables, $S_t = \{G_t, H_t\}$, through the following linear recurrence relations:

$$G_t = \gamma_t G_{t-1} + \beta_t v_t k_t^\top, \qquad H_t = \gamma_t H_{t-1} + \beta_t k_t k_t^\top, \tag{8}$$

where as before $\gamma_t \in [0, 1]$ is a forget gate and $\beta_t \in [0, 1]$ is an input gate. We adopt the conjugate gradient method to obtain a solution $q_t^* = \text{linsolve}(H_t + \Lambda, q_t) = (H_t + \Lambda)^{-1}q_t$ (Lanczos, 1950; Hestenes et al., 1952). This yields a numerically stable Mesa layer as $\text{linsolve}(H_t + \Lambda, q_t)$ is stable irrespective of forgetting strength, albeit at a higher memory cost compared to single matrix state RNN models, as an additional matrix of size $n_a \times n_a$ needs to be propagated forward alongside the standard matrix of size $n_v \times n_a$. Although the RNN state size increases, this expansion amounts to less than 1% of the entire memory footprint of models trained in this paper, which includes both state and parameters.

To enable efficient training, we introduce a chunkwise parallelized (Hua et al., 2022; Yang & Zhang, 2024) algorithm to compute equation 7. Our method builds on top of established efficient implementations of GLA, that we briefly review now. First, note that the output of this layer can be written as $o_t^{\text{GLA}} = G_t q_t = \sum_{i=1}^t \zeta_{ti} v_i k_i^\top q_t$. Let us chunk a sequence of length $T$ in $T/C$ chunks of size $C$, with $c \in \{0, C, \ldots, T - C\}$. The crucial insight to enable leveraging matrix-matrix multiplication and parallelization across time for GLA is that, given a chunked state variable $G_c$, we can compute the output at time $c < t \leq c + C$ as $o_t^{\text{GLA}} = (G_c + \sum_{i=c+1}^t \zeta_{ti} v_i k_i^\top)q_t = G_c q_t + \sum_{i=c+1}^t \zeta_{ti} v_i k_i^\top q_t$, which can be done in parallel for $t \in \{c+1, \ldots c+C\}$. In matrix notation we write

$$O_c^{\text{GLA}} = G_c Q_c + V_c(Z_c \odot (K_c^\top Q_c^*)), \tag{9}$$

where $K_c = [k_c, \ldots, k_{c+C}]$ and $O_c^{\text{GLA}}, V_c, Q_c$ accordingly, and $Z_c$ is a upper triangular matrix of size $C \times C$ containing the appropriate forgetting terms.

Now, we highlight that the Mesa layer can be decomposed into two parts:

$$o_t^{\text{mesa}} = \sum_{i=1}^{t} \zeta_{ti} v_i k_i^\top q_t^*, \quad \text{and } q_t^* = (H_t + \Lambda)^{-1} q_t. \tag{10}$$

The first part is equivalent to GLA, and can therefore be computed efficiently as just described. It therefore remains to be shown how to obtain $Q_{h,c}^* = [q_{h,c}^*, \dots, q_{h,c+C}^*]$ within a given chunk of size $C$ in parallel. As we explain in detail in Appendices C & D, the key observation is that the compute-intensive part of a CG iteration boils down to $\sum_{i=1}^{t} \zeta_{ti} k_i k_i^\top p$, with $p$ its current search direction, a computation that is once again in the GLA form. Alongside its fast convergence properties, this is the reason for picking the CG method as our solver, as it allowed us to leverage existing efficient chunkwise parallel linear attention implementations. The new Mesa layer proposed in this paper therefore admits a parallel training mode with $O(T)$ complexity, alongside the recurrent inference mode with $O(1)$ complexity. In Appendix D, we further show how to efficiently compute gradients through the layer in chunkwise parallel form. Finally, we discuss details on precision within our CG solver in Appendix G.5.

## 3 TRAIN AND INFERENCE TIME OF THE MESA LAYER

**Chunkwise parallel Mesa layer leads to competitive train time.** In Figure 2, we report training and inference times on a TPUv5 and H100 for both transformers (MHA), common RNN alternatives and the MesaNet. Despite having to solve $t \cdot H$ linear systems of equations per layer during training as well as compute gradients through the found solutions, the MesaNet remains competitive at train time with respect to MHA and RNN alternatives.

**The Mesa layer, applied with static $k$, is relatively slow especially early in the sequence.** We present in Appendix Table 5 an analysis of the memory and computational costs of inference, comparing the Mesa layer to MHA as well as recently developed RNNs. This overview highlights a tension that the MesaNet faces. On the one hand, if the number of conjugate gradient (CG) steps $k$ is set to zero we obtain $q_t^* = q_t$, and so recover gated linear self-attention (GLA) and its compute and memory requirement. Thus, we require $k > 0$ for the Mesa layer to differ from GLA, which provides a lower bound for the computational cost of the Mesa layer. Note that the Mesa layer is, in terms of flops, roughly $k$ times as costly as linearized transformer models such as GLA, Mamba2 and xLSTM and $k - 1$ times more costly as (Gated) DeltaNet. Furthermore, because the total cost of executing the CG method grows with $kn_a^2$, there is a maximal value of $k$ for which the Mesa uses fewer flops than MHA for a given sequence length.

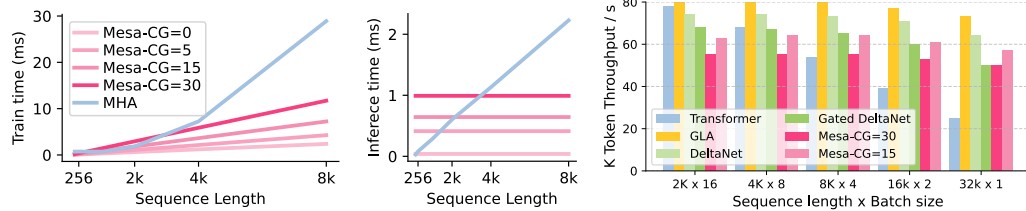

**Figure 2: Train and inference time of a Mesa layer using different number of CG steps**. *Left*: Train time of a single Mesa later on a TPUv5: output the entire sequence, compute the cross entropy loss, and gradients w.r.t. layer parameters. We use batch size of 4, key size of 128 and 8 heads. *Center*: Inference time of a single Mesa layer on a TPUv5: compute the next token given a certain context length. We use batch size of 128, key size of 128 and 8 heads. *Right*: Token throughput (in thousands) when training 1B parameter models on a H100 GPU. We compare a Flash-Attention-2 (Dao, 2023) transformer implementation with a triton-based chunkwise parallel implementation of RNN models, including the MesaNet which uses 30 or 15 CG steps across all layers. All models use a key size of 128 and share the same backbone, see Appendix G. We observe competitive token throughput on H100s of the MesaNet despite using substantially more flops.

We show this in Figure 2 (center) for a typical choice of $n_a = 128$, where we plot inference time as a function of sequence length for both MHA and the Mesa layer, when varying $k$. These numbers reflect the runtime of a single layer and might vary across inference use cases and accelerators.

| Layer | Recurrence | Memory read-out |
|---|---|---|
| Mamba2 | $G_t = \gamma_t G_{t-1} + v_t k_t^\top$ | $o_t = G_t q_t$ |
| GLA | $G_t = \gamma_t G_{t-1} + \beta_t v_t k_t^\top$ | $o_t = G_t q_t$ |
| DeltaNet | $G_t = G_{t-1}(I - \beta_t k_t k_t^\top) + \beta_t v_t k_t^\top$ | $o_t = G_t q_t$ |
| Gated DeltaNet | $G_t = G_{t-1}(\gamma_t(I - \beta_t k_t k_t^\top)) + \beta_t v_t k_t^\top$ | $o_t = G_t q_t$ |
| mLSTM | $G_t = \gamma_t G_{t-1} + \beta_t v_t k_t^\top, z_t = \gamma_t z_{t-1} + \beta_t k_t$ | $o_t = G_t q_t / \max\{1, |z_t^\top q_t|\}$ |
| Mesa | $G_t = \gamma_t G_{t-1} + \beta_t v_t k_t^\top, H_t = \gamma_t H_{t-1} + \beta_t k_t k_t^\top$ | $o_t = G_t \text{linsolve}(H_t + \Lambda, q_t)$ |

**Table 1:** Overview of recent linear recurrent models which we compare to in this work, except for LRU layers, see De et al. (2024).

**The Mesa layer allocates test-time compute dynamically.** Being a test-time optimizer, the Mesa layer offers a principled way for dynamically allocating test-time compute. The number of CG steps $k$ required to reach a given desired error tolerance $\epsilon$ is generally head-, sequence- and token-specific due to the context-dependence of the linear systems $H_t + \Lambda$ to be solved. Via utilization of a stopping criterion, the Mesa layer thus exhibits dynamic inference (and potentially training) costs. This dynamic test-time compute feature of the Mesa layer draws both parallels and differences to softmax self-attention: whereas softmax self-attention increases compute (and memory) as a function of sequence length independently of the sequence being processed, the Mesa layer adjusts compute dynamically, according to the incoming data it needs to process. We provide in Section 5 an experimental analysis of this property of the Mesa layer in trained MesaNets.

## 4 MESANET IN A LANGUAGE WORLD

Here we present results obtained on 1B-parameter models trained on 50B tokens from the SlimPajama (Soboleva et al., 2023) dataset, and refer to Section L for an extended analysis, comparing models ranging from 140M, 440M up to 1B parameters, each on 15B and 50B tokens. Furthermore, we report strong results on synthetic environments in Section K, which we omit for brevity here.

**Architecture & baselines.** For the main model backbone, we follow the architecture of common transformers, and employ $N$ stacked residual blocks with 1) a sequence modeling part such as multi-head-attention (MHA) or the Mesa layer and 2) a gated MLP block (see Figure 1). As baselines, we compare to a number of other efficient alternatives to MHA based on linear recurrent layers: Mamba2 (Dao & Gu, 2024), Gated Linear Attention (GLA) (Yang et al., 2024b; Katharopoulos et al., 2020), xLSTM (Beck et al., 2024), (Gated) DeltaNet (Schlag et al., 2021a; Yang et al., 2024c;a) and Hawk (De et al., 2024), see Table 1. The latter differs from the models reviewed in Section 2 by employing a vector-valued state, being closer in spirit to a (now linearized) traditional LSTM (Hochreiter & Schmidhuber, 1997). Furthermore we investigate a recurrent hybrid Hawk-Mesa model alternating between a linear recurrent unit (Hawk) and the Mesa layer which we motivate in the next section.

**Controls.** On top of related work, we train transformer models with Sliding-Window Attention (SWA) (Beltagy et al., 2020) of varying window sizes. These models have constant per-token memory and compute cost. The motivation to study SWA models is based on the assumption that transformers as well as SWA models have near perfect recall capabilities, at least within their attention window. Therefore, they provide a simple and interpretable control to study language modeling, reasoning and in-context recall capabilities of RNNs.

**Setup.** We tokenize the SlimPajama datasets using the byte-level BPE tokenizer introduced in GPT-2 (Radford et al., 2018; Brown et al., 2020a) following Beck et al. (2024) and train all modes on a sequence length of 2048 and a fixed ordering of training data. For each model configuration, we scan over a range of learning rates, and select the model that minimizes perplexity on the holdout validation dataset of SlimPajama. For exact hyperparameters and training specifications for each model, see Appendix G. For all results, unless otherwise specified, we use MesaNets with a fixed amount of 30 CG steps. See Appendix M on varying CG steps during training and Section 5 on using the CG stopping criterion to invoke dynamic test-time compute.

We stress that through sharing the exact same architecture backbone, tokenizer, data and data order across all models, while using the same number of parameters and independently tuned learning

| | SLIM ppl↓ | LMB. ppl↓ | WIKI. ppl↓ | PG19 ppl↓ | GOV. ppl↓ | QASP. ppl↓ | AVG |
|---|---|---|---|---|---|---|---|
| - Hawk | 11,24 | 26,67 | 12,23 | 10,93 | 10,63 | 14,89 | 14.43 |
| - Mamba2 | 11,39 | 28,02 | 12,23 | 11,42 | 10,42 | 14,02 | 14.58 |
| - GLA | 10,99 | 29,77 | 11,77 | 10,95 | 9,99 | 13,52 | 14.03 |
| - xLSTM | 11,01 | 26,93 | 11,81 | 10,94 | 10,00 | 13,55 | 14.03 |
| - DeltaNet | 11,01 | 27,08 | 11,73 | 11,00 | 10,02 | 13,44 | 14.05 |
| - Gated DeltaNet | 10,89 | 26,79 | 11,58 | 10,81 | 9,88 | 13,28 | 13.87 |
| - Mesa | 10,83 | 26,78 | 11,49 | 10,71 | 9,80 | 13,13 | 13.79 |
| - Hawk-Mesa | 10,78 | 26,59 | 11,53 | 10,60 | 9,79 | 13,20 | 13.75 |
| - SWA-4 | 16,46 | 29,93 | 19,42 | 16,42 | 17,86 | 29,15 | 21.54 |
| - SWA-64 | 12,37 | 27,76 | 14,14 | 12,51 | 11,56 | 16,77 | 15.85 |
| - SWA-1024 | 11,00 | 27,22 | 11,78 | 10,92 | 9,79 | 13,11 | 13.97 |
| - Transformer | 10,86 | 27,16 | 11,42 | 10,74 | 9,69 | 12,86 | 13.79 |

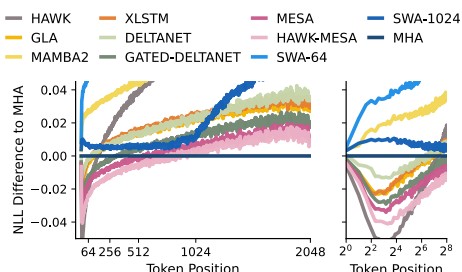

**Table 2: Language Modeling Performance (PPL ↓) of 1B Models (50B Tokens) evaluated on sequence length of 2048).** Mesa and Hawk-Mesa show strong performance on all benchmarks, matching or exceeding a Transformer baseline w.r.t. to avg. per-token PPL. Lambada (LMB.) scores are higher due to significantly shorter sequences (≤ 256) with an average of 78 tokens.

**Figure 3: NLL Difference relative to a Transformer (1B models, 50B tokens) on SlimPajama.** Most recurrent layers show superior language modeling performance in terms of NLL up to the 64'th token. MesaNet and Hawk-Mesa extend the advantage beyond 512 tokens. The advantage early in the sequence is even more apparent in log-scale (right).

rate for all models, we aim to provide a fair 1-1 comparison[1]. This controlled setup should allow to solely assess differences on the sequence mixing layer while reducing noise. Note, however, that this backbone might be a suboptimal choice for RNNs, including the MesaNet. Related work has tuned architectures to their specific sequence layers (Beck et al., 2024; Gu & Dao, 2024). However, these architectural optimizations prevent the integration of Mixture-of-Experts layers, a heavily used building block in current language models. Therefore, we carefully evaluate all sequence layers on the same backbone, based on the widespread decoder-only transformer architecture – here, the Llama2 model (Touvron et al., 2023), including rotary position encodings (RoPE; Su et al., 2024) when using softmax attention layers. This backbone does not fuse MLPs with sequence layers, allowing for a direct comparisons between layers. Furthermore, we did not attempt to optimize the architecture e.g., key size and number of heads for the Mesa layer.

**Comparison to the original mesa layer.** We considered comparing to the original sequential-in-time Mesa layer (von Oswald et al., 2024). However, because this model was already an order of magnitude slower when training at the 400M parameter scale, and suffered a large increase in SlimPajama language modeling perplexity of about 3.2 points (∼23% performance degradation) due to the inability to train with forget gates, we did not pursue these comparisons further. These results directly motivate the new Mesa layer introduced in this paper.

## 4.1 LANGUAGE MODELING (WITHIN AND BEYOND TRAIN SEQUENCE LENGTH)

We measure a model's general language modeling capabilities first by assessing average per-token perplexity (PPL) (Jelinek et al., 1977) on a set of benchmarks. We report PPL on the hold-out validation set of SlimPajama (Soboleva et al., 2023), as well as Lambada (Paperno et al., 2016), Wikitext-2 (Merity et al., 2016), PG19 (Rae et al., 2019), GovReport (Huang et al., 2021), and Qasper (Dasigi et al., 2021) on the train sequence length and beyond. Because uniformly averaging over all tokens might masquerade important differences between models, we additionally investigate average per-token PPL conditional on sequence position. As we see below, this turns out to be a crucial factor when comparing RNNs to transformers.

**MesaNet is a strong language model early in sequences.** When evaluating on the training sequence length of 2048, MesaNet and Hawk-MesaNet outperform all recurrent baselines on all benchmarks on the common metric of average per-token PPL (see Table 2). MesaNet matches on average the performance of the transformer baseline, while Hawk-MesaNet even surpasses it. Notably, a SWA model with a window size of 1024 outperforms the majority of recurrent baselines. However, attaining similar PPL scores does not imply equivalent language modeling abilities at different sequence lengths (Lin et al., 2025). Conditioning on the token position, and assessing the NLL difference relative to a transformer, reveals, surprisingly, that most recurrent layers exhibit superior language modeling performance early in the sequence but fall behind later in the sequence (see Figure 3). Recurrent models show especially strong performance on short sequences up to 64 tokens.

---

[1]Related work such as Yang et al. (2024a), Behrouz et al. (2024) and Behrouz et al. (2025a) use a single learning rate for all models which likely leads to biased and unfair comparisons. Behrouz et al. (2025a) further inherit baseline results from previous work which use a different tokenizer, confounding the comparison further.

| Model | Reasoning Global (Acc ↑) | Reasoning Local (Acc ↑) | In-Context Recall (Acc ↑) | Scramble 100-shot (Acc ↑) | Translation 50-shot (bleu-sb ↑) | Model | Reasoning Global (Acc ↑) | Reasoning Local (Acc ↑) | In-Context Recall (Acc ↑) | Scramble 100-shot (Acc ↑) | Translation 50-shot (bleu-sb ↑) |
|---|---|---|---|---|---|---|---|---|---|---|---|
| Hawk | 37.42 | 50.04 | 21.29 | 4.70 | 3.51 | Hawk | 41.17 | 51.57 | 25.04 | 6.49 | 4.73 |
| Mamba2 | 37.58 | 48.19 | 32.21 | 3.38 | 2.55 | Mamba2 | 41.62 | 50.51 | 37.67 | 4.19 | 4.18 |
| GLA | 39.45 | 48.86 | 36.50 | 5.06 | 2.57 | GLA | 44.34 | 51.44 | 39.64 | 7.29 | 7.58 |
| xLSTM | 38.97 | 48.90 | 34.89 | 5.56 | 2.74 | xLSTM | 42.99 | 51.50 | 39.25 | 7.78 | 7.68 |
| DeltaNet | 39.72 | 48.91 | 35.19 | 5.14 | 2.47 | DeltaNet | 43.86 | 51.58 | 40.46 | 7.93 | 5.37 |
| Gated DeltaNet | 40.19 | 49.10 | 35.96 | 6.17 | 2.98 | Gated DeltaNet | 44.84 | 50.76 | 39.54 | 8.90 | 8.53 |
| Mesa | 40.88 | 49.64 | 39.30 | 6.22 | 3.83 | Mesa | 45.03 | 50.49 | 41.79 | 10.10 | 8.17 |
| Hawk-Mesa | 40.13 | 49.53 | 36.23 | 5.19 | 3.25 | Hawk-Mesa | 44.62 | 51.51 | 39.99 | 8.61 | 8.81 |
| SWA-4 | 28.63 | 48.20 | 9.79 | 0.82 | 0.14 | SWA-4 | 31.20 | 49.62 | 11.87 | 1.66 | 0.51 |
| SWA-64 | 38.17 | 48.76 | 24.84 | 3.66 | 2.59 | SWA-64 | 42.33 | 50.52 | 26.72 | 5.91 | 4.70 |
| SWA-1024 | 41.30 | 48.84 | 38.21 | 5.43 | 5.65 | SWA-1024 | 45.68 | 51.35 | 40.84 | 6.66 | 14.17 |
| Transformer | 42.15 | 48.80 | 49.95 | 6.01 | 5.89 | Transformer | 45.54 | 51.62 | 52.27 | 6.98 | 13.61 |

(a) 400M Params, 50B Tokens    (b) 1B Models, 50B Tokens

**Table 3: Grouped Benchmark Scores (↑) on models trained on 50B Tokens from SlimPajama with a context length of 2048.** We compare the aggregated performance of models with Linearized Recurrent Unit, Gated Linearized Multi-Head Attention, DeltaNet and MESA layers on 5 different subsets of benchmarks. As a reference, we show the performance of Sliding Window-Attention models (SWA) with varying window sizes.

While Hawk exhibits the best performance up to this depth, the model exhibits a sharp performance decline after that. This finding motivated us to introduce and investigate the Hawk-Mesa model, which combines the best short-sequence and long-sequence modeling layers (as measured by negative log-likelihood). Confirming this intuition, the Hawk-Mesa outperforms the remaining recurrent models, with the MesaNet being second best: MesaNet and Hawk-MesaNet not only attain the strongest early-in-the-sequence modeling ability, but also extend the advantage beyond a depth of 512 tokens.

**MesaNet is competitive on length extrapolation with recurrent baselines, but SWA-1024 is a hard-to-beat baseline.** Next, we evaluate the ability to extrapolate to sequences of up to 32k tokens (see Figure 4). While transformer, Mamba2, DeltaNet and HawkMesa fail to extrapolate catastrophically to longer sequences on all evaluated benchmarks, MesaNet exhibits length-extrapolation capabilities superior to Hawk, GLA, xLSTM and on-par with Gated DeltaNet on all evaluated long-sequence benchmarks with respect to PPL scores (aggregated and conditional on token positions). However, these results should be tempered by the fact that a SWA model with an attention window of 1024 attains competitive benchmark scores, even superior at a sequence length of 32k on some benchmarks. This finding is in line with recent criticism that PPL may not distinguish a model's ability to capture local vs. long-range dependencies between tokens (Hu et al., 2024; Fang et al., 2024). We refer to Section L for detailed score breakdown and results on the Needle-in-the-haystack (NIAH) benchmark (Hsieh et al., 2024), where MesaNet shows strong performance.

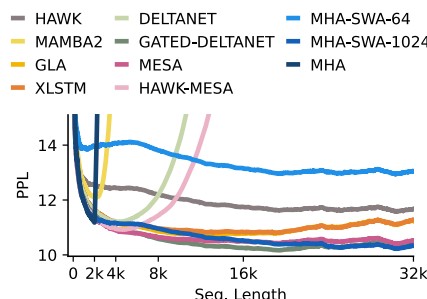

Figure 4: Avg. Mean-so-Far PPL on 3 Long-Context Benchmarks (WIKI, GOV, QASPER).

## 4.2 LANGUAGE BENCHMARKS

We next evaluate MesaNet's capabilities on a comprehensive set of downstream tasks, ranging across zero-shot reasoning, in-context recall and in-context learning tasks. We evaluate on various benchmarks considered in prior work (Gu & Dao, 2024; Yang et al., 2024a; Beck et al., 2024), and complement them with few-shot learning tasks involving token-manipulation and translation. We present the aggregated results of 400M and 1B models trained on 50B tokens in **??**, and report detailed scores in Section L. Across most evaluated benchmarks, the MesaNet matches or exceeds the performance of the evaluated recurrent baselines.

**Zero-Shot Common-Sense Reasoning Performance: Transformers & MesaNet ≥ other RNNs.** Prior work (Gu & Dao, 2024; Yang et al., 2024a; Behrouz et al., 2024; Beck et al., 2024) commonly reports the average performance of a set of common-sense reasoning benchmarks to compare models. However, evaluations of SWA models with different window sizes reveal that competitive, or even superior, scores on many of these frequently reported benchmarks can be attained with attention

window size as short as 4 (see Table 13). This observation strongly indicates that some of these benchmarks are exploitable by short-range language heuristics, and do not require longer-range language modeling capabilities to reach competitive scores, or are simply too hard such that we end up measuring noise. To reduce the potential benchmark noise and deconfound the results, we hence report the zero-shot reasoning benchmarks in two separate splits:

- The **Global Reasoning Benchmark Set** encompasses all benchmarks where we observe a significant performance increase with a growing attention window size. This includes Lambada (Paperno et al., 2016), HellaSwag (Zellers et al., 2019) and RACE-{M,H} (Lai et al., 2017). Within both reported model sizes (400M and 1B), MesaNet outperforms all other recurrent models on average on these benchmarks. However, MesaNet still underperforms the transformer baseline.

- The **Local Reasoning Benchmark Set** includes all benchmarks where we see little to marginal improvement with a growing attention window size. This includes PIQA (Bisk et al., 2020), WinoGrande (Sakaguchi et al., 2021), ARC-{E,C} (Clark et al., 2018), SIQA (Sap et al., 2019), BoolQ (Clark et al., 2019), OpenBookQA (Mihaylov et al., 2018) and StoryCloze (Srinivasan et al., 2018). Unsurprisingly, we observe very similar average scores for all models. Notably, Hawk, the worst performing recurrent model on global reasoning and in-context recall benchmarks, shows excellent performance on this benchmark subset. This observation supports the hypothesis that these subsets of benchmarks are likely to measure different capabilities, and highlights the differences between Hawk to e.g. the MesaNet. These analyses motivate the recurrent hybrid Hawk-Mesa model, which tries to capitalize on the complimentary strengths of the two layers.

**In-Context Recall Performance: Transformers $>$ MesaNet $\geq$ other RNNs.** To gauge the ability to recall in-context information, we follow Arora et al. (2024) and Yang et al. (2024a) and evaluate models on SWDE (Lockard et al., 2019), SQUAD (Rajpurkar et al., 2016), FDA (Arora et al., 2023b), TQA (Kembhavi et al., 2017), NQ (Kwiatkowski et al., 2019) and DROP (Dua et al., 2019). We adopt the minimal-transformed versions of the benchmarks from Arora et al. (2024) that adjust for the evaluation of non-instruction-tuned models. In line with the observations on synthetic benchmarks in Section K, MesaNet outperforms all other recurrent models on these tasks. Moreover, MesaNet exceeds the performance of a SWA-1024, the only recurrent model to do so. However, there remains a gap in performance relative to the transformer baseline with an attention window size of 2048.

**Few-Shot Learning Performance: Transformers & MesaNet $>$ other RNNs.** Finally, we measure the model's ability to learn from few-shot demonstrations. We evaluate on two GPT3 word scrambling tasks (cycle letters in word, anagrams of all but first and last two characters) (Brown et al., 2020b) and three translation tasks (WMT-14 FR-EN (Bojar et al., 2014) , WMT-16 DE-EN and RO-EN (Bojar et al., 2016) ). MesaNet demonstrates strong performance on all few-shot learning tasks. While it exceeds the performance of the Transformer on word scrambling tasks, it fails to do so in translations.

## 5 TEST-TIME COMPUTE ANALYSIS

In the previous section we showed results from models trained and evaluated with 30 CG steps. We study now the effect of using the MesaNet trained on 30 CG steps but evaluate the model when using a dynamic stopping criterion aiming to reducing the CG steps used at inference time. We refer again to Appendix C for a description of the CG method used in this work.

**Mesa objectives differ widely across heads and layers.** When analysing the internals of the Mesa layer on sequences of the SlimPajama validation set, we observe a bimodal distribution of condition numbers of $H_{h,t} + \Lambda_h$ across heads almost in every layer, see Figure 14. In particular, we observe that heads either have 1) large and growing condition number with sequence length, or 2) rather low and constant condition number over the sequence. In every layer, there are roughly 1-2 heads for which the condition number of linsolve$(H_{h,t} + \Lambda_h, q_{h,t})$ (and therefore the number of CG steps) grows with $t$. This motivates dynamic allocation of CG steps in every head.

**MesaNets allocate test-time compute dynamically.** We test 1) reducing the number of CG steps of all layers and heads uniformly, and 2) varying the solver's stopping criterion $\epsilon$ to dynamically allocate test-time compute. As shown in Figure 7, when reducing CG steps uniformly, we observe an increase in negative log-likelihood when comparing to our model evaluated with 30 steps, especially on tokens later in the sequence. This is in line with our findings on the need for higher number of steps as $t$ grows. By contrast, with a dynamic stopping criterion $\epsilon$, increasing $\epsilon$ yields a uniform degradation

over sequence length. A model with a stopping criterion of $\epsilon = 10^{-4}$ performs on-par with the base model using a fixed number of 30 CG steps, while reducing the average CG steps used to $\approx 9$.

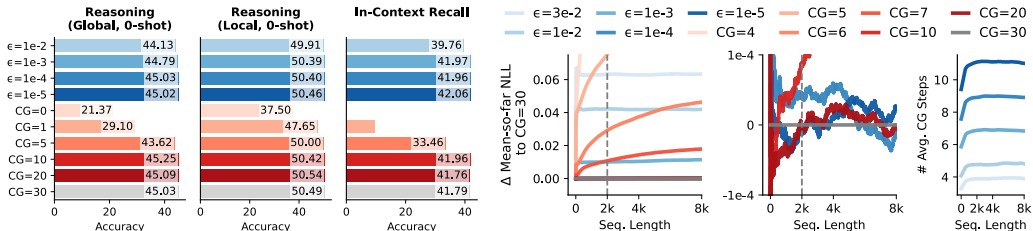

**Figure 5: Effect of Number of Conjugate Gradient (CG) Steps on SlimPajama Perplexity within and beyond train context length.** We show here the effect of reducing the number of CG steps during inference on token perplexity across token position of a 1B MesaNet trained on 50B tokens. We either use a fixed number CG steps uniformly across the model or apply a dynamic stopping criterion $\epsilon > 0$.

## 6 DISCUSSION

We present a chunkwise parallelized, numerically stable version of the Mesa layer (von Oswald et al., 2024), and scale it up to 1B parameter language models. This layer generates a prediction by solving an optimization problem, which yields a linear model that best fits a given sequence. Our Mesa layer can allocate test-time compute dynamically according to the stopping criterion. Complex sequences are then modeled by many of such layers, while interleaving them with MLPs, into MesaNets.

This approach has ties to multiple long-running lines of research. It relates to alternatives to end-to-end differentiation based on stacks of greedy local learners (e.g., Hinton et al., 2006; Nøkland & Eidnes, 2019; Veness et al., 2021), bringing these to the fast inference timescale, and then delegating to nonlocal backpropagation-based learning the role of determining which optimization problems must be solved at inference time. This in turn relates to mesa-optimization (Hubinger et al., 2019), since test-time optimization objectives (though not the optimizers themselves) are discovered by (base) sequence prediction loss optimization. The idea of specifying the output of a neural layer through an optimization problem is an old one (Amos & Kolter, 2017; Gould et al., 2021), with roots at least to energy-based neural models (Hopfield, 1984). Finally, the Mesa layer is perhaps most related to fast weights of Schmidhuber (1992), replacing Hebbian with locally-optimal learning.

The Mesa layer extends state-of-the-art recurrent language models such as Mamba (Gu & Dao, 2024), RWKV (Peng et al., 2023), xLSTM (Beck et al., 2024), and (Gated) DeltaNet (Schlag et al., 2021a; Yang et al., 2024c;a), which can also be motivated by an in-context regression loss, but update their fast weights with a slower GD process. In a new in-depth evaluation, we show that RNNs, in particular MesaNets, outperform transformers significantly early in sequences, while underperforming in next-token prediction and benchmark performance when longer contexts are needed. It should be stressed that it is exactly in the long-context regime, however, that RNNs show advantages over transformers in terms of inference time. In our view, these observations merit further investigation, and may serve as the starting point for novel RNN scaling law analyses.

The biggest shortcoming of the MesaNet in its current form is the increase in test-time compute despite its dynamic nature. One possible way around this may lie on the findings of Figure 14, where we see that heads which require more CG steps often do not forget, i.e. $\gamma \approx 1$ irrespective of the input data. This motivates leveraging the similarity of solutions from neighboring time steps, to warm-start optimization of consecutive steps. Moreover, one could envision a hybrid approach where the chunkwise parallel CG method introduced in this paper is used during training, while then reverting back to using the efficient Sherman-Morrison recursion at inference time, which could work given the almost-no-forgetting $\gamma \approx 1$ condition. We point to additional discussion points in Appendix J and leave investigating these directions for future work.

## ACKNOWLEDGMENTS

The authors would like to thank wholeheartedly Sarthak Mittal for his contributions to the work after the official submission deadline. Further, we would like to thank the anonymous reviewers for their

time and thoughtful reviews that helped to improve this work. We would also like to thank Soham De and Stephen Roller for insightful discussions and advice on model pre-training at the beginning of the project. Moreover, we would like to thank Angelika Steger, Rajai Nasser, Maciej Wołczyk, Eyvind Niklasson, Ahmad Beirami, Dimitris Papailiopoulos and the members of the Paradigms of Intelligence team for fruitful discussions and their support throughout the project.

## Reproducibility statement

We provide pseudocode for the conjugate-gradient implementation of the Mesa layer in Section C and Section D, and provide detailed descriptions regarding numerical precision in Section G.5. All other important aspects for training (e.g. tokenizer, data, context length) are given in Section 4. We will furthermore, upon publication, provide a triton-based open source implementation of the MesaNet and Mesa layer, as well as educational colab notebooks to further ease reproduction and experimentation with our layer and models. Moreover, we focused not only on improving the numbers of our proposed method but scanned hyperparameters of the related works extensively (see Section E). Lastly, we focused on an apples-to-apples comparison between methods by using the exact same backbone while only varying the sequence layer.

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

# A    RELATED WORK

**Linear Attention.**    As already described above, Tsai et al. (2019) demonstrated that the softmax attention mechanism can be linearized by replacing the softmax kernel $\kappa(k, q) = \exp(k^T q)$ with a surrogate kernel $\kappa' = \langle \sigma(k), \sigma(q) \rangle$. The resulting linear attention mechanism iteratively accumulates the outer product of key-value pairs into a recurrent state that is queried at each step, resembling RNNs (Katharopoulos et al., 2020). Since then, numerous works have proposed different designs of the feature map $\sigma(\cdot)$ (Katharopoulos et al., 2020; Choromanski et al., 2021; Schlag et al., 2021a; Peng et al., 2021; Sun et al., 2023; Dao & Gu, 2024) and key-value normalization (Yang et al., 2024c; Schlag et al., 2021a; Sun et al., 2023). Notably, a more general form of (unnormalized) linear attention was introduced in the early '90s as *Fast Weight Programmers* (Schmidhuber, 1992; Schlag et al., 2021a; Ba et al., 2016), connected to Meta-Learning (Schmidhuber, 1987).

**Test-time regression.**    Contrary to softmax attention, linear attention variants are only capable of storing a finite number of key-value associations. Given key dimension $d_{\text{key}}$, there exist at most $d_{\text{key}}$ orthogonal keys, and therefore, retrieval beyond $d_{\text{key}}$ tokens cannot be error-free. Inspired by the error-correcting delta rule (Widrow & Hoff, 1960), Schlag et al. (2021b;a) proposed to interpolate the value with the previously stored association, yielding the DeltaNet. The DeltaNet update rule is equivalent to performing a gradient descent step with respect to the recurrent state $\Phi$ on $||\Phi k_t - v_t||^2$. Yang et al. (2024a) demonstrated that the DeltaNet is parallelizable and achieved strong language modeling performance when embedded into a modern architecture. Motivated this online regression loss, other works derived the same update rule as the DeltaNet. Instead of a parallel implementation, Liu et al. (2025) approximate the update with a diagonal matrix, while Sun et al. (2025) perform the DeltaNet update on a per-chunk basis, implicitly performing batched gradient descent. Building on this, Titans (Behrouz et al., 2024) adds momentum to the batched gradient descent update. Wang et al. (2025); Behrouz et al. (2025b) unify numerous efficient foundation models from the perspective of test-time regression. Extending Titans, concurrent follow-up work Atlas Behrouz et al. (2025a) is effectively a sliding-window variant of the Mesa layer. It is worth highlighting that this line of research is an instance of Dynamic Evaluation (Mikolov et al., 2010; Krause et al., 2018; Clark et al., 2022; Rannen-Triki et al., 2024), where model weights are updated at test time via gradient descent steps on a prediction loss.

**Models with recurrent depth.**    The MesaNet is related to a broader class of models building on fixed point iterations. Universal Transformers (Dehghani et al., 2019) apply transformer blocks iteratively, using Adaptive Computation Time (Graves, 2017) to make the number of recurrent steps token-dependent. Deep Equilibrium Models (DEQs) (Bai et al., 2019) take this idea further by directly solving the corresponding fixed point iteration using quasi Newton methods. More recently, Schöne et al. (2025) introduced an implicit State Space Model that also relies on a fixed-point iteration, which is trainable in parallel utilizing the Phantom Gradient technique (Geng et al., 2021). In contrast to DEQ-style methods, the Mesa layer benefits from the linear structure of fast weight memory, which allows for a more efficient optimization using conjugate gradient steps.

**Linear RNNs with forgetting.**    Forget gates were first introduced by Gers et al. (1999) within the framework of Long Short-Term Memory networks (LSTMs) (Hochreiter & Schmidhuber, 1997), and have since become part of the standard LSTM architecture. Even more, studies on simplified LSTM variants, such as the Gated Recurrent Unit (Cho et al., 2014), have shown the forget gate to be fundamental for the effectiveness of recurrent sequence models (Westhuizen & Lasenby, 2018).

Compared to LSTMs, modern linear attention variants have adopted more coarse grained forgetting mechanisms on the matrix-valued recurrent state. RetNet (Sun et al., 2023) and TransNormerLLM (Qin et al., 2024) both utilize a trainable decay factor on the recurrent matrix. More recent work found that *data-dependent* forgetting improves language modeling performance, although the data dependency is usually limited to the current input, but not the recurrent state, to allow for parallel training. Using an input-dependent decay factor as in this work is the de-facto standard in modern linear attention variants, such as Mamba-2 (Dao & Gu, 2024), xLSTM (Beck et al., 2024), and Gated DeltaNet (Yang et al., 2024a). Gated Linear Attention (Yang et al., 2024b) opts for a data-dependent decay vector, effectively using a separate forget gate for each row of the matrix-valued recurrent state. Similarly, Gated Slot Attention (Zhang et al., 2024) applies separate input-dependent forget gates to each row of both matrices of a fixed size Key-Value cache.

**State Space Models.** State Space Models (SSMs) (Gu et al., 2021; 2022; Fu et al., 2023; Gu & Dao, 2024) build upon first-order differential equations used to describe dynamical systems, which are then discretized for sequence modeling. In linear time-invariant (LTI) SSMs, the recurrent state can be obtained through a fixed linear combination of previous recurrent states, which allows for a parallel mode using convolutions. Gu et al. (2022) identified the computation of the convolution kernel as the primary bottleneck and proposed Structured State Space Models (S4), a parametrization for LTI SSMs that enables efficient computation. Mamba (Gu & Dao, 2024) introduces *selectivity* to State Space Models, making the recurrent state transitions dependent on the input. Since the resulting *time-varying* SSM cannot leverage global convolutions, the authors propose a hardware-efficient parallel scan implementation. Mamba-2 further constrains the transition matrix to scalar times identity, and demonstrates that the resulting State Space Model is equivalent to (gated) linear attention (Dao & Gu, 2024).

## B DERIVATION OF PREVIOUS TEST-TIME TRAINING RULES

For completeness, we discuss in more detail the update rules for a number of closely related previous sequence modeling layers discussed above and in the main text section 2. Like the Mesa layer, the update rules of these models perform some form of test-time learning by optimizing a sequence of objective functions $(L_{t'})_{t'=1}^{t}$. We summarize in Table 4 the update rules and corresponding online objective functions that we cover below.

| Layer | Objective function | Update rule |
|---|---|---|
| GLA | $L_t = -v_t^\top \Phi k_t + \frac{1-\gamma_t}{2\beta_t} \operatorname{Tr}(\Phi\Phi^\top)$ | $\Phi_t = \Phi_{t-1} - \beta_t \nabla_\phi L_t(\Phi_{t-1})$ |
| DeltaNet | $L_t = \frac{1}{2}\|v_t - \Phi k_t\|^2$ | $\Phi_t = \Phi_{t-1} - \beta_t \nabla_\phi L_t(\Phi_{t-1})$ |
| LongHorn | $L_t = \frac{1}{2}(v_t - \Phi k_t)^\top \operatorname{diag}(\beta_t)(v_t - \Phi k_t)$ $+\frac{1}{2}\operatorname{Tr}(\Phi - \Phi_{t-1})^\top(\Phi - \Phi_{t-1})$ | $\Phi_t = \arg\min_\Phi L_t(\Phi)$ |
| Atlas | $L_t = \sum_{t'=t-c+1}^{t} \zeta_{tt'}\|v_{t'} - \mathcal{M}_\Phi(k_{t'})\|^2$ | $\tilde{\Phi}_t = \tilde{\theta}_t \tilde{\Phi}_{t-1} + \nabla_\Phi L_t(\Phi_{t-1})$ $\Phi_t = \gamma_t \Phi_{t-1} - \beta_t \operatorname{NewtonSchulz}_k(\tilde{\Phi}_t)$ |
| Mesa | $L_t = \frac{1}{2}\sum_{t'=1}^{t} \zeta_{tt'}\|v_{t'} - \Phi k_{t'}\|^2$ $+\frac{1}{2}\operatorname{Tr}(\Phi\Lambda_t\Phi^\top)$ | $\Phi_t = \arg\min_\Phi L_t(\Phi)$ |

**Table 4:** Overview of test-time training recurrent layers, whose update rules can be derived from an online learning objective function.

**GLA and DeltaNet update rules.** For convenience, we first restate equation 3 below:

$$L_t(\Phi) = l_t(\Phi) + \frac{1}{2}\operatorname{Tr}(\Phi\Lambda_t\Phi^\top). \tag{11}$$

We show in detail how to obtain the basic GLA and DeltaNet update rules by letting $\Phi_t$ follow an online gradient-based learning dynamics,

$$\Phi_t = \Phi_{t-1} - \beta_t \nabla_\phi L_t(\Phi_{t-1}), \tag{12}$$

where the input gate $\beta_t$ plays the role of a time-dependent step size.

For GLA, we choose $l_t$ to be the quadratic continuous-state Hopfield energy,

$$l_t(\Phi) = l_t^{\text{Hopfield}}(\Phi) := -v_t^\top \Phi k_t,$$

and we set the quadratic regularizer to depend on the forget gate $\gamma_t$ and input gate $\beta_t$ as follows:

$$\Lambda_t = \frac{1-\gamma_t}{\beta_t}I.$$

Now, plugging $l_t$ and $\Lambda_t$ into equation 12 yields

$$\Phi_t = \Phi_{t-1} - \beta_t \nabla_\phi \left[ -v_t^\top \Phi k_t + \frac{1-\gamma_t}{2\beta_t}\operatorname{Tr}(\Phi\Phi^\top) \right]\Bigg|_{\Phi=\Phi_{t-1}} \tag{13}$$

$$= \Phi_{t-1} - (1-\gamma_t)\Phi_{t-1} + \beta_t v_t k_t^\top \tag{14}$$

$$= \gamma_t \Phi_{t-1} + \beta_t v_t k_t^\top, \tag{15}$$

which corresponds to gated linear attention as defined in the main text (equation 2).

To obtain DeltaNet, we choose instead $l_t$ to be the squared error loss,

$$l_t(\Phi) = l_t^{\text{sq-err}}(\Phi) := \frac{1}{2}\|v_t - \Phi k_t\|^2,$$

and we disable the regularizer ($\Lambda_t = 0$), as it was not included in the original DeltaNet model (Schlag et al., 2021a). Performing again the same computation as above, but now with this squared error online loss, yields the DeltaNet update:

$$\Phi_t = \Phi_{t-1} - \beta_t \nabla_\phi \left[\frac{1}{2}\|v_t - \Phi k_t\|^2\right]\bigg|_{\Phi=\Phi_{t-1}} \tag{16}$$

$$= \Phi_{t-1} + \beta_t (v_t - \Phi_{t-1} k_t) k_t^\top. \tag{17}$$

**LongHorn update rule.** Yet another recent method called LongHorn (Liu et al., 2025) can be derived as online learning on a sequence of loss functions ($l_t$). Its update rule can be derived by minimizing an objective function:

$$\Phi_t = \arg\min_\Phi L_t^{\text{LongHorn}} \tag{18}$$

$$= \arg\min_\Phi \frac{1}{2}(v_t - \Phi k_t)^\top \text{diag}(\beta_t)(v_t - \Phi k_t) + \frac{1}{2}\text{Tr}(\Phi - \Phi_{t-1})^\top(\Phi - \Phi_{t-1}), \tag{19}$$

with $\beta_t$ now a vector of the same dimension as $v_t$, instead of a scalar, determining an elementwise squared error precision. The solution can be obtained in closed-form, following the derivation provided in Appendix C of (Liu et al., 2025):

$$\Phi_t = \Phi_{t-1} + \text{diag}(\epsilon_t)(v_t - \Phi_{t-1} k_t) k_t^\top, \tag{20}$$

with $\epsilon_{ti} = \frac{\beta_{ti}}{1+\beta_{ti}k_t^\top k_t}$. This is a variant of DeltaNet with a particular diagonal input-dependent step size that is both a function of $k_t$ and $\beta_t$ (which is chosen to be a vector in this model, as opposed to the scalar gates used in our DeltaNet and in our current MesaNet implementation). For computational efficiency, the actual implementation of LongHorn approximates the update above with a simpler rule that makes use of elementwise multiplications, denoted here by $\odot$:

$$\Phi_t = (\mathbf{1} - \epsilon_t(k_t \odot k_t)^\top) \odot \Phi_{t-1} + (\epsilon_t \odot v_t)k_t^\top, \tag{21}$$

where $\mathbb{1}$ is a matrix of ones. Like the DeltaNet, the LongHorn objective still only takes into account the instantaneous squared error for the current key-value pair, with an additional memory quadratic potential pulling towards the previous solution to avoid forgetting it entirely through the full $\arg\min$. By contrast, the Mesa layer explicitly optimizes the full forget-weighted sum of squared errors from the beginning of the sequence until the present ($t' = 1$ to $t$).

**Omega/Atlas update rule.** Concurrent work by Behrouz et al. (2025a) investigated online learning layers that are intimately related to the Mesa layer. The paper focuses on a sliding window variant of our objective function:

$$L_t^{\text{Omega}} = \sum_{t'=t-c+1}^{t} \zeta_{tt'}\|v_{t'} - \mathcal{M}_\Phi(k_{t'})\|^2, \tag{22}$$

where $c$ is the sliding window length, and $\zeta_{tt'}$ determines the cumulative forget at time step $t$ for the past loss $t'$, as in the Mesa layer objective. The authors further allow $\mathcal{M}_\Phi$ to be a 1-hidden-layer MLP with parameters $\Phi$, similarly to (Sun et al., 2025), and unlike the Mesa layer, which derives a specialized update exploiting the fact that $\mathcal{M}$ is a linear model. Behrouz et al. (2025a) optimize the sequence of loss functions ($\tilde{l}_t$) online using a second-order Muon method (Jordan et al., 2024):

$$\tilde{\Phi}_t = \tilde{\theta}_t \tilde{\Phi}_{t-1} + \nabla_\Phi l_t^{\text{Omega}}(\Phi_{t-1}), \tag{23}$$

$$\Phi_t = \gamma_t \Phi_{t-1} - \beta_t \text{NewtonSchulz}_k(\tilde{\Phi}_t), \tag{24}$$

where $\text{NewtonSchulz}_k$ denotes the execution of $k$ steps of the NewtonSchulz algorithm, $\tilde{\Phi}_t$ is an auxiliary momentum gradient accumulation state variable, and $\tilde{\theta}_t$ is a dynamic (time-dependent) momentum decay factor, which determines the retention of past accumulated gradients.

## C    RANK-ONE UPDATE CONJUGATE GRADIENT METHOD

In the next two sections, we describe how we can use the conjugate gradient method to obtain a solution for $(H_t + \Lambda_t)^{-1}q_t = q_t^*$ for many $t$ in parallel. As we will discuss below, the aim is to show how one can do this without materializing $H_t$ for all time steps as this would lead to unnecessary memory overhead, see Yang et al. (2024b) for a detailed discussion of this problem and a "chunkwise parallel" solution. We therefore aim to show here, as a starting point, how to compute $q_t^*$ without materializing $H_t = H_{t-1}\gamma_t + k_t k_t^T$ and only relying on $H_{t-1}$ as well as on $y_t$ and $k_t$. This will eventually allow us, see the next Appendix section D, to compute and materialize $H_t$ only every $T/C$ steps with train length $T$ and chunksize $C$ times, leading to a drastic decrease in memory usage. We will do this while approximating $Q_c^* = [q_{c+1}^*, \dots, q_{c+C}^*]$ numerically in parallel by only materializing $H_c$ where $c \in \{0, C, 2C, \dots T - C\}$.

We opted to initialize the conjugate gradient method with $x \leftarrow q_t \cdot \text{diag}(H_t + \Lambda_t)^{-1}$ in this work.

---

**Algorithm 1** Rank-One Update Conjugate Gradient Method

---

1: **procedure** RANKONECONJUGATEGRADIENT($H_{t-1}, \gamma_t, k_t, q_t, \epsilon, k_{\max}$)
2:     **Input:** Symmetric positive-definite matrix $H_{t-1} \in \mathbb{R}^{n \times n}$, forget strength $\gamma_t \in (0, 1)$, key $k_t \in \mathbb{R}^n$, query $q_t \in \mathbb{R}^n$, tolerance $\epsilon > 0$, maximum iterations $k_{\max}$.
3:     **Output:** Approximate solution $x$.

4:     $k \leftarrow 0$
5:     $x \leftarrow q_t \cdot \text{diag}(H_{t-1} + \Lambda_t)^{-1}$ ▷ Initial guess $x \in \mathbb{R}^n$
6:     $r \leftarrow q_t - (H_{t-1}\gamma_t + k_t k_t^\top + \Lambda_t)x$ ▷ Initial residual $r$
7:     $p \leftarrow r$ ▷ Initial search direction $p$
8:     $\delta_{old} \leftarrow r^T r$ ▷ Squared norm of the initial residual
9:     $\delta_0 \leftarrow \delta_{old}$ ▷ Store initial squared norm for relative tolerance

10:     **while** $k < k_{\max}$ **do** ▷ Loop until max iterations reached
11:         $q \leftarrow (H_{t-1}\gamma_t + k_t k_t^\top + \Lambda_t)p$ ▷ Matrix-vector product $(H_{t-1}\gamma_t + k_t k_t^\top + \Lambda_t)p$
12:         $\alpha \leftarrow \frac{\delta_{old}}{p^T q}$ ▷ Step length $\alpha$
13:         $x \leftarrow x + \alpha p$ ▷ Update solution $x$
14:         $r \leftarrow r - \alpha q$ ▷ Update residual $r$
15:         $\delta_{new} \leftarrow r^T r$ ▷ Squared norm of the new residual, $\delta_{new}$

16:         **if** $\sqrt{\delta_{new}} \le \epsilon \sqrt{\delta_0}$ **then** ▷ Check relative convergence: $||r_{k+1}|| \le \epsilon ||r_0||$
17:             **break** ▷ Converged
18:         **end if**

19:         $\beta \leftarrow \frac{\delta_{new}}{\delta_{old}}$ ▷ Improvement factor $\beta$
20:         $p \leftarrow r + \beta p$ ▷ Update search direction $p$
21:         $\delta_{old} \leftarrow \delta_{new}$ ▷ Store new norm as old for next iteration
22:         $k \leftarrow k + 1$ ▷ Increment iteration counter
23:     **end while**

24:     **return** $x$ ▷ Return the approximate solution
25: **end procedure**

---

On top of $H_{t-1}p$, all other parts of the $(H_{t-1}\gamma_t + k_t k_t^T + \Lambda_t)p$ computation can be reduced to one vector inner-product $k_t^\top p$ as well as element-wise products and a final addition of the results. One can therefore approxiamte $q_t^*$ numerically without materializing $H_t$, which we will extend in the following to chunks i.e. compute $Q_c^* = [q_{c+1}^*, \dots, q_{c+C}^*]$ in parallel without explicitly materializing $H_t$ with $c < t \le c + C$. This will become obvious after realizing that the computation of $(H_{t-1}\gamma_t + k_t k_t^T + \Lambda_t)p$ is equivalent to GLA, therefore allowing for the chunkwise parrallel computation proposed in Yang et al. (2024b) of GLA.

Note that the most flops during inference are spend in the matrix-vector product $H_t p$ where we apply the CG method simply to $(H_t + \Lambda_t)^{-1} q_t$ (and not do not use the "rank-one" update formulation above) resulting in the $\mathcal{O}(k n_a^2)$ of Table 5.

We refer to Appendix G.5 for further details about numerical precisions considerations within our CG solver.

## D    CHUNKWISE PARALLEL FORM OF GATED LINEAR ATTENTION AND THE MESA LAYER

**Mesa layer forward pass.** The main Mesa recurrence (Equation 7) can be rewritten as follows, considering only one head and assuming without loss of generality that input gates are absorbed in keys and values:

$$
\begin{aligned}
H_t &= H_{t-1}\gamma_t + k_t k_t^\top \\
G_t &= G_{t-1}\gamma_t + v_t k_t^\top \\
q_t^* &= (H_t + \Lambda)^{-1} q_t \\
o_t &= G_t q_t^*
\end{aligned}
\tag{25}
$$

Note that $H_t$ is symmetric, and $\Lambda$ is symmetric positive definite, so $H_t + \Lambda$ is also symmetric. Let's define

$$
\zeta_{ts} = \begin{cases} \prod_{i=s+1}^{t} \gamma_i & \text{if } t \geq s \\ 0 & \text{otherwise} \end{cases}
$$

with which the computation of $o_t$ (unrolling the definition of $G_t$) has the following form:

$$
o_t = \sum_{i=1}^{t} \zeta_{ti} v_i k_i^\top q_t^*.
\tag{26}
$$

To connect to Section 2 where the Mesa layer is defined through a set of optimized linear model fast weights $\Phi$, we note that this is equivalent to minimizing the following objective w.r.t. $\Phi$,

$$
\hat{\Phi}_t^{\text{mesa}} = \arg\min_{\Phi} \frac{1}{2} \sum_{i=1}^{t} \zeta_{ti} \|v_i - \Phi k_i\|^2 + \frac{1}{2}\text{Tr}(\Phi\Lambda\Phi^\top),
\tag{27}
$$

and then computing the output through $o_t = \hat{\Phi}_t^{\text{mesa}} q_t$.

To see why this is the case, let us compute the stationarity condition

$$
\nabla_\Phi \left[ \frac{1}{2} \sum_{i=1}^{t} \zeta_{ti} \|v_i - \Phi k_i\|^2 + \frac{1}{2}\text{Tr}(\Phi\Lambda\Phi^\top) \right] = 0
\tag{28}
$$

$$
\iff \Phi\Lambda - \sum_{i=1}^{t} \zeta_{ti}(v_i - \Phi k_i)k_i^\top = 0
\tag{29}
$$

$$
\iff \Phi\Lambda - \sum_{i=1}^{t} (\zeta_{ti} v_i k_i^\top - \Phi \zeta_{ti} k_i k_i^\top) = 0
\tag{30}
$$

$$
\iff \Phi\Lambda + \Phi \tilde{K}_t \tilde{K}_t^\top = \tilde{V}_t \tilde{K}_t^\top
\tag{31}
$$

$$
\iff \Phi = \tilde{V}_t \tilde{K}_t^\top (\tilde{K}_t \tilde{K}_t^\top + \Lambda)^{-1}
\tag{32}
$$

$$
\iff \Phi = \left( \sum_{i=1}^{t} \zeta_{ti} v_i k_i^\top \right) \left( \sum_{i=1}^{t} \zeta_{ti} k_i k_i^\top + \Lambda \right)^{-1}.
\tag{33}
$$

To simplify the calculation we introduced the auxiliary matrix variables $\tilde{V}_t$ and $\tilde{K}_t$, which absorbed square roots of the cumulative forget factors $\zeta_t$. We denote the above (unique, for $\Lambda > 0$) solution by $\Phi_t^{\text{mesa}}$.

Now, the recurrence relation for the state variable $H_t$ can be solved analytically, yielding

$$H_t = \gamma_t H_{t-1} + k_t k_t^\top = \sum_{i=1}^t \left( \prod_{j=i+1}^t \gamma_j \right) k_i k_i^\top = \sum_{i=1}^t \zeta_{ti} k_i k_i^\top, \tag{34}$$

assuming $H_0 = 0$. The same holds for the other state variable, $G_t = \gamma_t G_{t-1} + v_t k_t^\top = \sum_{i=1}^t \zeta_{ti} v_i k_i^\top$.

Therefore, as claimed, we recover equation 25:

$$o_t = \hat{\Phi}_t^{\text{mesa}} q_t \tag{35}$$

$$= \left( \sum_{i=1}^t \zeta_{ti} v_i k_i^\top \right) \left( \sum_{i=1}^t \zeta_{ti} k_i k_i^\top + \Lambda \right)^{-1} q_t \tag{36}$$

$$= G_t (H_t + \Lambda)^{-1} q_t \tag{37}$$

$$= G_t q_t^*. \tag{38}$$

**Chunkwise form.** We remark that if $q_t^*$ is given, this computation is equivalent to a Gated Linear Attention (GLA) layer Yang et al. (2024b), and thus can be efficiently computed on GPUs and TPUs by splitting the sequence in blocks of opportune sizes $C$ resulting in a "chunkwise parallel" form of the layer. In short, given $G_c$, where $c \in \{0, C, \ldots, T - C\}$ dividing the training sequence length $T$ in $T/C$ chunks of size $C$, we can compute the output at time $c < t \le c + C$ as

$$o_t = (G_c + \sum_{i=c+1}^t \zeta_{ti} v_i k_i^\top) q_t^* = G_c q_t^* + \sum_{i=c+1}^t \zeta_{ti} v_i k_i^\top q_t^* \tag{39}$$

Similar to softmax self-attention, this computation can be done in parallel for $t \in \{c+1, \ldots c+C\}$ which becomes clearer when using matrix notation

$$O_c = G_c Q_c^* + V_c (Z_c \odot (K_c^\top Q_c^*)) \tag{40}$$

where $K_c = [k_c, \ldots, k_{c+C}]$ and $O_c, V_c, Q_c^*$ accordingly. $Z_c$ is a upper triangular matrix of size $C \times C$ with $Z_c[i, j] = \zeta_{c+j,c+i}$. Please see for Triton-based implementation of this chunked parallel formulation of GLA at https://github.com/fla-org/flash-linear-attention.

We differ from GLA as the Mesa layer replaces $q_t$ which is the standard query $q_t = W_q e_t$ by $q_t^* = (H_t + \Lambda)^{-1} q_t$ which, as we alluded to above, can as well be computed equivalently to GLA in chunkwise parallel form. Indeed, as shown in the previous section, the conjugate gradient method relies purely on simple vector additions and multiplications which can be trivially realized in chunkwise parallel form without extensive memory overhead, with the exception of $(H_t + \Lambda)p$. This operation suffers from the same memory problems as a naive GLA layer implementation as storing $H_t$ for all time steps is costly which we therefore wish to circumvent. Fortunately, this can easily be done with the exact same chunkwise parallel trick just discussed, which we now leverage to compute

$$(H_t + \Lambda)p = H_t p + \Lambda \cdot p = \sum_{i=1}^t \zeta_{ti} k_i k_i^\top p + \Lambda \cdot p. \tag{41}$$

which is required in the conjugate gradient algorithm.

Note that the first term $\sum_{i=1}^t \zeta_{ti} k_i k_i^\top p$ is in an equivalent form of GLA (by replacing $v_i$ with $k_i$) for which we just established that a fast chunkwise parallel formulation exist, if we again store only some intermediate states $H_c$. We conclude that the computation of $q_t^* = (H_t + \Lambda)^{-1} q_t$ and therefore the whole Mesa layer can be approximated by repeatedly applying a in chunkwise parallel computation leveraging matrix-matrix accelerators on GPUs or TPUs.

**Mesa layer backward pass**: Let $e_t$ be the error coming from future layers at time $t$ and $L$ be the final loss. Then we have the following:

$$e_t^* = (H_t + \Lambda)^{-1} G_t^\top e_t$$

$$\frac{dL}{dq_t} = \frac{do_t}{dq_t} e_t = e_t^*$$

$$\frac{dL}{d\Lambda_{t,i}} = \frac{do_t}{d\Lambda_{t,i}} e_t = -q_{t,i}^* e_{t,i}^*$$

$$\frac{do_t}{dv_s} e_t = e_t \zeta_{ts} k_s^\top (H_t + \Lambda)^{-1} q_t$$

$$\frac{dL}{dv_s} = \sum_{t \geq s} \zeta_{ts} e_t q_t^{*\top} k_s$$

$$\frac{dL}{dk_s} = \sum_{t \geq s} \zeta_{ts} (q_t^* e_t^{*\top} v_s - e_t^* q_t^{*\top} k_s - q_t^* e_t^{*\top} k_s)$$

$$\frac{dL}{d\gamma_s} = \sum_{t \geq s} \zeta_{ts} (q_t^{*\top} G_{s-1} e_t - e_t^{*\top} H_{s-1} q_t^*)$$

This is a time-reversed version of the formulas to compute the derivatives with respect to $v_s$ and $k_s$. Note that $\frac{dL}{dv_s}$ and $\frac{dL}{dk_s}$ can again be computed in chunkwise parallel manner as they are sums of expressions which are all GLA formulation equivalent. $e_t^*$ is also chunkwise parallel compatible since, as we just established, running conjugate gradient (chunked) parallelized in time is possible.

It remains to see how to quickly compute the derivatives with respect to $\gamma_s$. To that purpose, let us consider the first term in the equation defining the derivative, as the second can be handled similarly; we have that:

$$\sum_{t \geq s} \zeta_{ts} q_t^{*\top} G_{s-1} e_t = \sum_{t \geq s} \text{Tr}[\zeta_{ts} q_t^{*\top} G_{s-1} e_t] =$$

$$= \sum_{t \geq s} \text{Tr}[G_{s-1} \zeta_{ts} e_t q_t^{*\top}] =$$

$$= \text{Tr}\left[ G_{s-1} \sum_{t \geq s} \zeta_{ts} e_t q_t^{*\top} \right]$$

This already gives a way to compute the derivatives that is linear in sequence length (as it is sufficient to accumulate the $t$-dependent part as $s$ decreases). However, for maximum efficiency we would like to also split the computation into blocks and make use of matrix multiplication units for this computation.

Let $F_s = \sum_{t \geq s} \zeta_{ts} e_t q_t^{*\top}$. We now explain how to compute the value above simultaneously for a block of indices $s = \mathcal{L} + 1, \ldots, \mathcal{U} - 1$.

$$G_{s-1} = G_{\mathcal{L}} \zeta_{s-1\mathcal{L}} + \sum_{\mathcal{L} < p < s} \zeta_{s-1p} v_p k_p^\top$$

$$\sum_{t \geq s} \zeta_{ts} e_t q_t^{*\top} = \sum_{s \leq t < \mathcal{U}} \zeta_{ts} e_t q_t^{*\top} + \zeta_{\mathcal{U}s} F_{\mathcal{U}}$$

$$\text{Tr}\left[ G_{s-1} \sum_{t \geq s} \zeta_{ts} e_t q_t^{*\top} \right] = \text{Tr}\left[ \left( G_{\mathcal{L}} \zeta_{s-1\mathcal{L}} + \sum_{\mathcal{L} < p < s} \zeta_{s-1p} v_p k_p^\top \right) \left( \sum_{s \leq t < \mathcal{U}} \zeta_{ts} e_t q_t^{*\top} + \zeta_{\mathcal{U}s} F_{\mathcal{U}} \right) \right] =$$

$$= \text{Tr}[G_{\mathcal{L}} F_{\mathcal{U}} \zeta_{\mathcal{U}s} \zeta_{s-1\mathcal{L}}] + \text{Tr}\left[ G_{\mathcal{L}} \zeta_{s-1\mathcal{L}} \sum_{s \leq t < \mathcal{U}} \zeta_{ts} e_t q_t^{*\top} \right] +$$

$$+ \text{Tr}\left[ F_{\mathcal{U}} \zeta_{\mathcal{U}s} \sum_{\mathcal{L} < p < s} \zeta_{s-1p} v_p k_p^\top \right] + \text{Tr}\left[ \sum_{s \leq t < \mathcal{U}} \zeta_{ts} e_t q_t^{*\top} \sum_{\mathcal{L} < p < s} \zeta_{s-1p} v_p k_p^\top \right] =$$

$$= \text{Tr}\left[G_{\mathcal{L}}F_{\mathcal{U}}\right]\zeta_{\mathcal{U}s}\zeta_{s-1\mathcal{L}} + \sum_{s \leq t < \mathcal{U}}\zeta_{ts}\zeta_{s-1\mathcal{L}}\text{Tr}\left[G_{\mathcal{L}}e_t q_t^{*\top}\right] +$$

$$+ \sum_{\mathcal{L}<p<s}\zeta_{\mathcal{U}s}\zeta_{s-1p}\text{Tr}\left[F_{\mathcal{U}}v_p k_p^\top\right] + \sum_{\mathcal{L}<p<s}\sum_{s \leq t < \mathcal{U}}\zeta_{ts}\zeta_{s-1p}\text{Tr}\left[e_t q_t^{*\top}v_p k_p^\top\right]$$

For computing the last term, we can make use of the fact that $\zeta_{ab} = 0$ if $a < b$ to rewrite it in the equivalent forms

$$\sum_{\mathcal{L}<p<s}\sum_{s \leq t < \mathcal{U}}\zeta_{s-1p}(q_t^{*\top}v_p)(k_p^\top e_t)\zeta_{ts} = \sum_{\mathcal{L}<p<\mathcal{U}}\sum_{\mathcal{L}<t<\mathcal{U}}\zeta_{s-1p}(q_t^{*\top}v_p)(k_p^\top e_t)\zeta_{ts}$$

which can be computed as the product of the three matrices $Z^*, Q, Z$ with $Z_{ij}^* = \zeta_{i-1j}$, $Q_{ij} = (q_j^{*\top}v_i)(k_i^\top e_j)$, $Z_{ij} = \zeta_{ij}$; the requested values appear then as the main diagonal of this matrix.

The second term can be similarly rewritten as

$$\zeta_{s-1\mathcal{L}}\sum_{s \leq t < \mathcal{U}}(q_t^{*\top}G_{\mathcal{L}}e_t)\zeta_{ts} = \zeta_{s-1\mathcal{L}}\sum_{\mathcal{L}<t<\mathcal{U}}(q_t^{*\top}G_{\mathcal{L}}e_t)\zeta_{ts}$$

which can be computed by multiplying the vector $p_t = q_t^{*\top}G_{\mathcal{L}}e_t$ by the $Z$ matrix defined above, and then by doing a point-wise vector multiplication by $\zeta_{s-1\mathcal{L}}$.

Finally, the first term can be computed simply by computing the trace once and then doing a point-wise vector multiplication, and the third term can be computed as the second.

# E    A FULL DESCRIPTION OF THE MESA LAYER, RELATED WORK AND THE MESANET

For completion, we repeat the Mesa layer computation which is described throught the following equations

$$\Delta e_t^{\text{mesa}} = \sum_{h=1}^H P_h \hat{\Phi}_{h,t}^{\text{mesa}} q_{h,t} = \sum_{h=1}^H P_h G_{h,t}\text{linsolve}(H_{h,t} + \Lambda_h, q_{h,t}). \tag{42}$$

The equation above depends on two state variables, $S_{h,t} = \{G_{h,t}, H_{h,t}\}$, which we obtain through the linear recurrence relations:

$$G_{h,t} = G_{h,t-1}\gamma_{h,t} + v_{h,t}k_{h,t}^\top\beta_{h,t}, \qquad H_{h,t} = H_{h,t-1}\gamma_{h,t} + k_{h,t}k_{h,t}^\top\beta_{h,t}, \tag{43}$$

where as before $\gamma_{h,t} \in [0,1]$ is a forget gate and $\beta_{h,t} \in [0,1]$ is a input gate, where we adopt the conjugate gradient method as the solver (Lanczos, 1950; Hestenes et al., 1952). Before the Mesa layer computation, we compute the keys, queries, values as well as input and forget strength in the following way.

First, we normalize the embeddings with an RMS norm $e_i \leftarrow \text{RMSNorm}(e_i)$. After projections $k_t = W_k e_t, q_t = W_q e_t, v_t = W_k v_t$ we convolve them in time with a window size of 4 e.g. $k_t \leftarrow \sum_{i=0}^3 k_{t-i}b_{i+1}$ with learnable parameters $b_1, \dots, b_4$. Furthermore, after applying a SiLU$(x) = x * \sigma(x)$ non-linearity we normalize the keys and queries (but not values) to have L2-norm of 1 i.e. $k_t \leftarrow \text{SiLU}(k_t)/\|\text{SiLU}(k_t)\|$ and $q_t \leftarrow \text{SiLU}(q_t)/\|\text{SiLU}(q_t)\|$.

For the forgetting and input gate we simply squeeze the RMS normed $e_t$ projections through a sigmoid i.e. $\beta_t = \sigma(e_t W_\beta)$ and $\gamma_t = \sigma(e_t W_\gamma)$. After computing the output of every head, we apply a RMS norm i.e. the actual output of the Mesa layer amounts to

$$\Delta e_t^{\text{mesa}} = \sum_{h=1}^H P_h \text{RMSNorm}_h(G_{h,t}\text{linsolve}(H_{h,t} + \Lambda_h, q_{h,t})). \tag{44}$$

The regularization parameters are simply send through a softplus function to ensure positivity i.e. $\Lambda_h \leftarrow \text{softplus}(\Lambda_h)$. We did experiment with a input / time dependent regularization strength but in this work opted for a fixed lambda over time, see Section J

**Comparison to related work**: To ensure a 1-1 comparison with related work, we use the exact same parametrization of the keys, values and queries as well as forget and input strength parametrization for the GLA, Mamba2 and (gated) DeltaNet. Here, only the state update as well as output computation differ depending on the rule, see Table 1 for an overview. The mLSTM layers, which we also compare to, have a different parametrization of the forgetting as well as input strength and keys and quries are not normalized by their L2 norm, see Beck et al. (2024).

| Layer | Output & state update | Memory | Flops output & state update |
|---|---|---|---|
| MHA | $o_t = \sum_{t'=1}^{t} v_{t'} \alpha(K_t^\top q_t)_{t'}$ | $(v_{h,t'}, k_{t'})_{t'=1}^{t} - 2n_a t$ | $\mathcal{O}(n_a t) - \mathcal{O}(1)$ |
| GLMHA | $o_t = \Phi_t q_t$ with 
 $\Phi_t = \Phi_{t-1} \gamma_t + \beta_t v_t k_t^T$ | $\Phi_t - n_a^2$ | $\mathcal{O}(n_a^2) - \mathcal{O}(n_a^2)$ |
| DN | $o_t = \Phi_t q_t$ with 
 $\Phi_t = \Phi_{t-1}(y_t(I - \beta_t k_t^\top k_t)) + \beta_t v_t^\top k_t$ | $\Phi_t - n_a^2$ | $\mathcal{O}(n_a^2) - \mathcal{O}(n_a^2)$ |
| MESA | Equation 7 | $S_t = \{G_t, H_t\} - 2n_a^2$ | $\mathcal{O}(n_a^2) + \mathcal{O}(k n_a^2) - \mathcal{O}(n_a^2)$ |

Table 5: **Flops as well as state size comparison between MHA, gated linearized multi-head-attention (GLMHA) such as xLSTM or Mamba2, (gated) DeltaNet (DN) and the Mesa layer during inference.** All softmax attention alternatives require $\mathcal{O}(n_a^2)$ flops, with key size $n_a$, to compute the output as well as update the state(s). The Mesa layer requires an additional $k$ steps of the CG method which costs $\mathcal{O}(k n_a^2)$. For simplicity we assume $n_v = n_a$.

### E.1 MODEL DESIGN

We give an overview over the network architecture for all models compared in this work in bullet points. The only difference is the way how to do the "sequence" mixing of the keys, valyes and queries (and forget and input gates), with an exception of the LRU layer (De et al., 2024), see Table 1.

- The model consists of an embedding layer of size $n_e$, which is also shared at the end of the model to compute the logits. We do not apply regularization on the parameters of the embedding.

- The model is then followed by $N$ number of blocks consisting of a sequencing layer e.g. MHA, GLMHA, DN or Mesa, described in Section 2, followed by an MLP layer. The input of both the MLP as well as sequencing layer go through a RMSNorm (Zhang & Sennrich, 2019), see Figure 1. After computing the logits, we apply a soft hyperbolic tangent clip with $c = 30$ with logits = $\tanh(\text{logits}/c)c$, again following the open source implementation of De et al. (2024), see `https://github.com/google-deepmind/recurrentgemma/blob/main/recurrentgemma/jax/griffin.py`.

- To compare all different sequencing layers as closely as possible and focus on their ability to incorporate information from the context, MHA, GLA, Mamba2, xLSTM, the (gated) DeltaNet, as well as the Mesa all use the exact same key size and therefore the exact same amount of parameters to compute the queries, keys and values. All RNN layers, for direct comparison, additionally only use per head a one dimensional gate for forgetting as well as writing which we squeeze through a sigmoid function i.e. $\beta_t = \sigma(W_\beta e_t)$ and $\gamma_t = \sigma(W_\gamma e_t)$, except the mLSTM layer. This stands in some contrast to how the models were originally designed e.g. Gated Linear Attention (Yang et al., 2024a) or RWKV (Peng et al., 2023) use higher dimensional forget gates. Furthermore, all RNN layers convolve the keys and queries with a window size of 4. This is by now a standard feature of contemporary RNN/SSM architectures, motivated by earlier analyses (Arora et al., 2023a; Fu et al., 2023). Note that for all models, except from mLSTM which uses a special parameterization and normalization, we apply a SiLU (or swish) non-linearity (Hendrycks & Gimpel, 2023) before we normalize the keys and queries by their L2-norm. The output of each head is independently before the linear projection back to the residual stream send through an additional RMSNorm.

- We define Mamba2 as forget-gated linearized multi-head attention following Yang et al. (2024c), and GLA as its forget- and input-gated counterpart; both methods with $e_t$-dependent gates.

- When using the LRU layer (De et al., 2024), we notice that the layer, in its default hyperparameter configuration, subsumes more parameters than MHA and the other RNN alternatives, as they use exactly the same number of parameters to each other. We therefore decrease the hidden size multiplier which determines the increase of the RNN state when compared to the embedding size, to match parameter count.

- The Hawk-Mesa model simply alternates between blocks that have either a LRU layer or a Mesa layer.

- For the MLP layers we follow again De et al. (2024). We create two branches both with dimension of $n_e \cdot 3$, apply a SiLU non-linearity to one of the branches and merge them by multiplying. We then down project with a simple linear layer into $n_e$ dimension.

- All weights are initialized by sampling them from a normal distribution and in "fan in" mode, while scaling the variance of the weights which project back to the residual stream by $2.0/N$.

## F EXPERIMENTAL DETAILS: MESANET IN SYNTHETIC ENVIRONMENTS

### F.1 MAD BENCHMARK SUITE

We follow the benchmarking procedure detailed in Poli et al. (2024) precisely: For each task in the suite, we evaluate the architectures on subtasks of varying difficulty (i.e. varying sequence length, number of training examples, vocabulary sizes and further, task-specific parameters) and

| Hyper Parameter | Search |
|---|---|
| Embedding dimension | 128 |
| Number of layers | 2 |
| Number of heads | 8 |
| Key size | 16 |
| Epochs | 200 |
| Batch size | 32 |
| Optimizer | AdamW |
|    Learning rate | [3e-3, 1e-3, 5e-4, 1e-4] |
|    Weight decay | [0.01, 0.1] |
|    $\beta$s | (0.9, 0.98) |
| Scheduler | Cosine Scheduler with Warmup |
|    Minimum learning rate | 1e-5 |
|    Warm-up start learning rate | 1e-7 |
|    Warm-up steps | 750 |

**Table 6:** MAD benchmark suite hyper-parameters, taken from Poli et al. (2024).

| Hyper Parameter | Search |
|---|---|
| Embedding dimension | [64, 128, 256, 512, 1024] |
| Number of layers | [1, 2, 4, 8, 12] |
| Number of heads | [1, 2, 4] |
| Epochs | 50 |
| Batch size | 32 |
| Optimizer | AdamW |
|    Learning rate | [1e-4, 2.5e-4, 1e-3] |
|    Weight decay | [0.01, 0.1] |
|    $\beta$s | (0.9, 0.99) |
| Scheduler | Cosine Scheduler with Warmup |
|    Minimum learning rate | 2.5e-5 |
|    Warm-up start learning rate | 1e-7 |
|    Warm-up steps | 25000 |

**Table 7:** RegBench hyper-parameter search-space, taken from Akyürek et al. (2024). For all models, we keep the key size fixed to 128 across combinations of embedding dimension and number of heads.

compute the mean accuracy. We further sweep over varying learning rates and weight decay values for each model and report the maximum average task accuracy. For each architecture, we fix a set of hyper-parameters that can be found in Table 6.

### F.2 REGBENCH IN-CONTEXT LANGUAGE LEARNING BENCHMARK

Following Akyürek et al. (2024), we report the test-accuracy of the configuration obtained from a grid-search over a pre-defined set of shared hyper-parameters for all models, which can be found in Table 7.

## G EXPERIMENTAL DETAILS: MESANET IN A LANGUAGE WORLD

We follow closely the experimental setup of Beck et al. (2024) as well as De et al. (2024).

### G.1 DATA

We train models on SlimPajama Soboleva et al. (2023) and use the GPT-2 tokenizer Radford et al. (2018) which uses a vocab size of 50257, as in Beck et al. (2024). We pre-tokenize the dataset

and fill up sequences with context length shorter than the train length, which is set to 2048, with other randomly sampled sequences until the context train length is full. We separate these separate sequences with a `BOS` token. We follow the same recipe when creating the validation data. Note that this procedure might bias the training as well as evaluation of the model towards shorter sequences.

We train on two dataset sizes: 15 billion and 50 billion tokens.

## G.2 MODEL DESIGN

We give an overview over the network in bullet points.

- The model consists of an embedding layer of size $n_e$, which is also shared at the end of the model to compute the logits. We do not apply regularization on the parameters of the embedding. We follow again De et al. (2024) and initialize the parameters of the embedding matrix in "fan in" mode but scale back the embedding during inference by $\sqrt{n_e}$ leading to a variance of 1 in the residual stream.

- The model is then followed by $N$ number of blocks consisting of a sequencing layer e.g. MHA, GLMHA, DN or Mesa, described in Section 2, followed by an MLP layer. The input of both the MLP as well as sequencing layer go through a RMSNorm (Zhang & Sennrich, 2019), see Figure 1. After computing the logits, we apply a soft hyperbolic tangent clip with $c = 30$ with logits $= \tanh(\text{logits}/c)c$, again following De et al. (2024).

- To compare all different sequencing layers as closely as possible and focus on their ability to incorporate information from the context, MHA, GLA, Mamba2, xLSTM, the (gated) DeltaNet, as well as the Mesa all use the exact same key size and therefore the exact same amount of parameters to compute the queries, keys and values. All RNN layers, for direct comparison, additionally only use per head a one dimensional gate for forgetting as well as writing which we squeeze through a sigmoid function i.e. $\beta_t = \sigma(W_\beta e_t + b_\beta), \gamma_t = \sigma(W_{\gamma_t} e_t + b_{\gamma_t})$, except the mLSTM layer which has a more elaborate parametrization. This stands in some contrast to how the models were originally designed e.g. Gated Linear Attention (Yang et al., 2024a) or RWKV (Peng et al., 2023) use higher dimensional forget gates. Furthermore, all RNN layers convolve the keys and queries with a window size of 4. Note that for all models, except from mLSTM which uses a special parameterization and normalization, we apply a SiLU (or swish) non-linearity (Hendrycks & Gimpel, 2023) before we normalize the keys and queries by their L2-norm. The output of each head is independently before the linear projection back to the residual stream send through an additional RMSNorm.

- We define Mamba2 as non-gated linearized multi-head attention following Yang et al. (2024c) and GLA as its gated counterpart with $e_t$-dependent forget strength $\gamma_t$.

- When using the LRU layer (De et al., 2024), we notice that the layer, in its default hyperparameter configuration, subsumes more parameters than MHA and the other RNN alternatives, as they use exactly the same number of parameters to each other. We therefore decrease the hidden size multiplier which determines the increase of the RNN state when compared to the embedding size, to match parameter count.

- The Hawk-Mesa model simply alternates between blocks that have either a LRU layer or a Mesa layer.

- For the MLP layers we follow again De et al. (2024). We create two branches both with dimension of $3n_e$, apply a SiLU non-linearity to one of the branches and merge them by multiplying. We then down project with a simple linear layer into $n_e$ dimension.

- All weights are initialized by sampling them from a normal distribution and in "fan in" mode, while scaling the variance of the weights which project back to the residual stream by $2.0/N$.

## G.3 TRAINING DETAILS

We train over all the models in this work with batch size of 256, the AdamW optimizer (Loshchilov & Hutter, 2019) with weight decay strength 0.1, $\epsilon = 1 \times 10^{-8}, \beta_1 = 0.9, \beta_2 = 0.98$, and a cosine learning rate scheduler with initial learning rate $1 \times 10^{-6}$, warmup steps of 2000 and a peak learning

| Model size | Train size | Transformer | Mamba2 | GLA | xLSTM | DeltaNet | Gated DeltaNet | Hawk | Hawk-Mesa | Mesa |
|---|---|---|---|---|---|---|---|---|---|---|
| Small | 15 | 0.0025 | 0.003 | 0.002 | 0.0025 | 0.003 | 0.001 | 0.002 | 0.0025 | 0.003 |
| Small | 50 | 0.003 | 0.001 | 0.001 | 0.001 | 0.001 | 0.001 | 0.001 | 0.001 | 0.00095 |
| Medium | 15 | 0.0015 | 0.0025 | 0.0025 | 0.003 | 0.003 | 0.0025 | 0.0025 | 0.002 | 0.0025 |
| Medium | 50 | 0.001 | 0.001 | 0.00095 | 0.0009 | 0.00085 | 0.00095 | 0.0009 | 0.0009 | 0.001 |
| Large | 15 | 0.002 | 0.002 | 0.002 | 0.0015 | 0.0015 | 0.002 | 0.002 | 0.002 | 0.002 |
| Large | 50 | 0.0008 | 0.0009 | 0.00085 | 0.0008 | 0.0008 | 0.0009 | 0.0009 | 0.00085 | 0.00085 |

**Table 8:** Peak learning rate for all models trained for this work determined by a learning rate grid scan.

rate of $l$ which is scanned for each experiment, see below. We cosine decay the learning rate to 10% of the peak learning rate till the end of the training determined by the train set size. We use as loss the classic cross entropy on the next token; we do not compute the loss on the BOS token. We apply gradient norm clipping to norm 1. We apply mixed precision training where the weights are float32 but activations are bfloat16 following Beck et al. (2024). Interestingly, we find that this actually improves next token perplexity slightly compared to using float32 everywhere.

### G.4 HYPERPARAMETER SCANS

We train 3 model sizes: 140 million, 440 million and 940 million parameters following roughly Beck et al. (2024). As already mentioned, all architectures have by construction almost exactly the same number of parameters for the same architectrual dimensions. All recurrent neural network types have the same parameters as multi-head attention but additionally have two parameter vectors of size $n_a$ which produce the two gates per head. The Mesa layer has additionally $n_a$ (fixed in time) parameters for (meta-)learned $\Lambda$ regularization. Since the parametrization of the LRU layer is different by construction, we simply adjust the hidden size scaling to 1.25 to match the parameters of the other RNN layers. The 3 different model sizes use key size $n_a = 128$ and otherwise are setup as follows:

- 140 million — Small: $N = 14$ blocks, $h = 6$ heads, embedding dimension $n_e = 768$.
- 440 million — Medium : $N = 28$ blocks, $h = 8$ heads, embedding dimension $n_e = 1024$.
- 940 million — Large : $N = 28$ blocks, $h = 12$ heads, embedding dimension $n_e = 1536$.

The exact number of parameters and peak learning rate can be found in Table 8. For all models, we scan the same range of learning rates: for models trained for 15 billion tokens we scanned {0.003, 0.0025, 0.002, 0.001, 0.0015}, and for models trained for 50 billion tokens, we observe, similar to Beck et al. (2024), that smaller learning rates were beneficial and thus scan {0.001, 0.00095, 0.0009, 0.0085, 0.0008}. We train all sliding window attention (SWA) models, as they are only reference points, with learning rate 0.001.

### G.5 NOTES ON PRECISION USED IN THE CG-SOLVER, MESA LAYER DESIGN CONSIDERATIONS OR *Why you shouldn't scream at your Mesa layer*

The MesaNet, for the model sizes we consider for the language experiments, solves during training millions of linear systems of equations numerically in one forward pass. Somewhat surprisingly, we did not encounter many training instabilities when setting some crucial hyperparameters and architectural details accordingly. First, we follow related work and normalize keys and queries - this is a crucial first step to stabilize the Mesa layer. Second, the most important hyperparameter for the Mesa layer, which strongly influences the conditioning number and therefore the number of CG steps needed to solve the linear systems, is the regularization strength $\Lambda$. Due to experimentation when training small models, we initialized $\Lambda = I$ but restricted its values to be lower-bounded by 0.25. We hypothesize that this lower bound is important to, implicitly, upper bound the condition number. We determined the $\Lambda$ lower bound by a grid scan when training the medium sized model on 15B tokens. See Figure O for some $\Lambda$ values of a trained model. We parameterize $\Lambda$ through a softplus function i.e. $\Lambda = 0.25 + \text{softplus}(\Lambda)$ and adjusted the initialization of the $\Lambda$ parameter accordingly.

When training on SlimPajama and using the GPT-2 tokenizer, we noticed that the dataset, especially the sequences which contain code, contains sequences which consist of many repeated tokens such as the empty token " ". We call this "screaming at your language model". These kind of inputs to the

Mesa layer lead to a matrix $H_{h,t} = K_{h,t}K_{h,t}^T$ which contains sums of the same vector outer product which we analysed leads to instabilities when $\gamma_t \approx 1$. We therefore upper-bound $\gamma_t$ by $b_\gamma = 0.9975$ (which might be train length specific) and adjust its value depending the input strength $\beta_t$: when training on SlimPajama, we use $\gamma_t = \gamma_t s_{\gamma_t}$ with $s_{\gamma_t} = (1 - (1 - b_\gamma)\beta_t^2)$. Note that other tokenizers which merge repeated " " should solve this problem partially. This correction improves perplexity in scans on small models and so we adopted it throughout our experiments.

**A final comment on the precision of the CG solver**: The opted to use FP32 matrix multiplication precision inside the CG solver solely within our Pallas kernel. Note that we used BF16 everywhere else to compare other RNN and transformer models with our MesaNet fairly. This reduces memory loading times as we only load data with BF16 precision, compute $q*$ in our solver with FP32 precision, and cast it down in our solver to FB16.

Although we did not investigate in depth FP16 or BF16 precision within the CG solver for which convergence problems are well known, we found the training times when using FP32 acceptable. We leave this important investigation for future work.

We end here with a note of caution when using these lower precisions on GPUs as more work might be needed to ensure stable convergence to the approximate solution of the linear solver.

### G.6    EXPERIMENTS COMPUTE RESOURCES

We provide here an estimate of the compute resources used for a single run of a 1B model. We note that transformers, MesaNets and other RNNs were of somewhat comparable speed on average and so estimate compute by averaging and not differentiating costs across models. We mostly relied on TPUv5 to conduct our experiment. Here we used multi-pod TPUv5s which fit the whole models, without model sharding, and therefore were able to rely solely on batch sharding. For the 1B models, one training run, with sparse intermediate evaluation, when training on 50B tokens lasted around 36 hours on average. When training on smaller models, the train time significantly dropped. All Mesa layer investigations were done on the 400million scale when training on 15B tokens resulting in train runs which last 3-12 hours depending on the amount of CG steps used and data parallelization applied.

Running our evaluation pipeline for all downstream benchmarks took on average 3 hours on the same hardware, although we note that we did not optimize this pipeline for run time.

### G.7    TOKEN THROUGHPUT COMPARISONS OF RECURRENT MODELS FOR 1B MODELS

We report in Figure 6 the throughput (in tokens / second) of the 1B MesaNet (for different fixed CG steps), Gated DeltaNet, Gated Linear Attention, as well as standard (global softmax-attention) Transformers. The MesaNet performs competitively, especially with a fixed number of 10 CG steps. We note that 10 CG steps are sufficient to obtain the superior MesaNet results reported in the main text. Gated linear attention, due to the limited flops and matrix multiplications needed to perform a forward pass, reaches significantly higher throughput than all other models. As expected, transformer throughput degrades with increasing sequence length.

## H    THE ORIGINAL RECURSIVE LEAST-SQUARES MESA LAYER

We now review the original version of the Mesa layer (Von Oswald et al., 2024), where $\hat{\Phi}_t^{\mathrm{mesa}}$ was determined through the classical recursive least-squares algorithm. The key observation is that $\hat{\Phi}_t^{\mathrm{mesa}} = V_t K_t^\top R_t^{-1} = \sum_{t'=1}^t v_{t'} k_{t'}^\top \left(\sum_{t'=1}^t k_{t'} k_{t'}^\top + \Lambda\right)^{-1}$, and that one can calculate the inverse $R_t^{-1}$ recursively through the Sherman-Morrison formula (Sherman & Morrison, 1950; Gauss, 1821), $R_t^{-1} = R_{t-1}^{-1} - \frac{R_{t-1}^{-1} k_t k_t^\top R_{t-1}^{-1}}{1 + k_t^\top R_{t-1}^{-1} k_t}$, with $R_0^{-1} = \Lambda^{-1}$. While efficient for sequential inference, this solution is problematic for two reasons. First, when introducing time-dependent forget gates $\gamma_t \in [0, 1]$ which scale the previous state, i.e., $(R_{t-1}\gamma_t + k_t k_t^\top)^{-1}$, the matrix inversion for small $\gamma_t$ can introduce numerical instabilities as $R_{t-1}^{-1} \frac{1}{\gamma_t}$ can grow unbounded. Moreover, note that this Mesa layer version forgets the regularization term $\Lambda$ exponentially fast, as it only enters through the initial

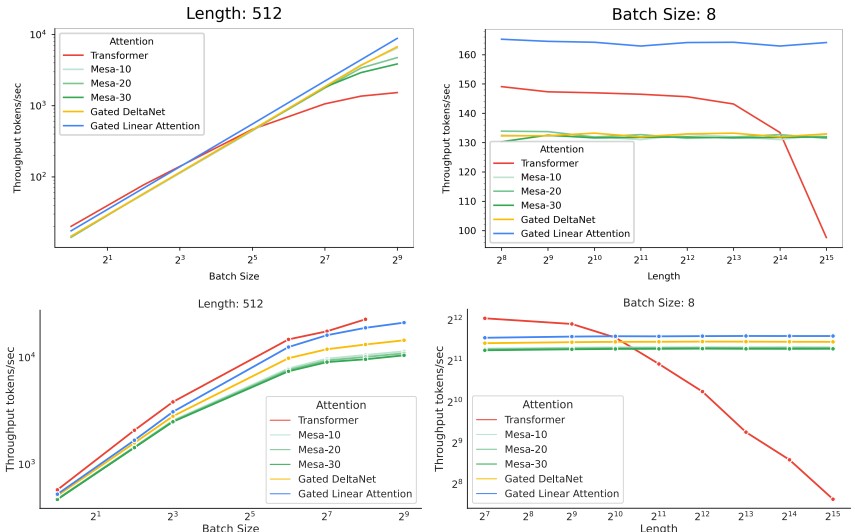

**Figure 6: 1B model throughput (tokens / sec) with bfloat16 activation and float32 weight precision on a H100 GPU (top row) using the open source framework of `https://github.com/fla-org/flash-linear-attention` or our custom TPUv5 implementation (bottom row).** We show the effect of scaling the batch size (left), while fixing the generation length, or scaling the generation length, while fixing the batch size on the token throughput / sec. For this experiment, we averaged over 5 iterations to reduce noise. On both hardware systems, we see that 1) MesaNet and Gated DeltaNet perform competitive despite MesaNet consuming significantly more flops, 2) Gated Linear Attention outperforming other layers significantly as well as 3) the throughput of the Transformer degrading with larger batchsize and especially sequence length. We chose sequence length for left panels and batch size for the right panels small enough, such that the (global softmax) Transformer does not run out of memory for the H100. On the TPUv5 and the left configuration, the Transformer is running out of memory for the largest batchsize.

.

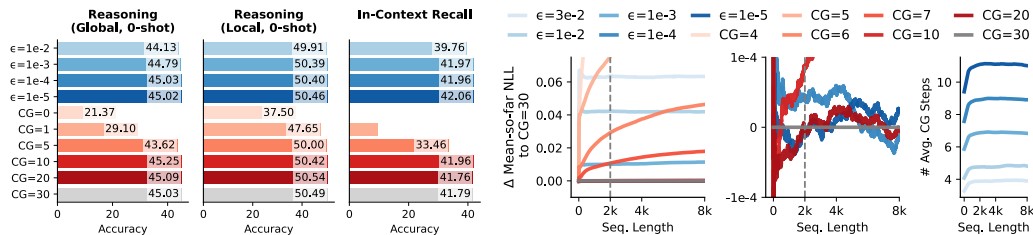

**Figure 7: Effect of Number of Conjugate Gradient (CG) Steps on SlimPajama Perplexity within and beyond train context length.** We show here the effect of reducing the number of CG steps during inference on token perplexity across token position of a 1B MesaNet trained on 50B tokens. We either use a fixed number CG steps uniformly across the model or apply a dynamic stopping criterion $\epsilon > 0$.

state $R_0^{-1}$. Second, we are not aware of a way of computing $R_t^{-1}$ in a parallel-in-time fashion. This precludes efficient parallel training at scale in current hardware.

**The Mesa layer as a second-order in-context learning method.** As reviewed in Sections 2 and A, the closely related DeltaNet model (Schlag et al., 2021a) updates a matrix-valued state variable $\Phi \in \mathbb{R}^{n_v \times n_a}$ following online gradient descent on a squared error loss. Omitting the head index, the dynamics of this layer reads

$$\Phi_t = \Phi_{t-1} - \beta_t \nabla l_t^{\text{sq-err}}(\Phi_{t-1}) = \Phi_{t-1} + \beta_t(v_t - \Phi_{t-1}k_t)k_t^\top. \tag{45}$$

To make comparison with this layer easier, we now express the Mesa layer (equation 4) in a similar recurrent form. We assume that we are in the case where the Sherman-Morrison recursion explained above holds, so that we can write $H_t^{-1}$ as a function of $H_{t-1}^{-1}$. This requires that forgetting is disabled ($\forall_t \gamma_t = 1$), or that the regularizer $\Lambda$ decays exponentially with time. For simplicity, we assume in what follows that there is no forgetting. Then, using the convention that $H_0 = \Lambda$, we have that

$$\Phi_t = G_t H_t^{-1} \tag{46}$$

$$= (G_{t-1} + v_t k_t^\top)H_t^{-1} \tag{47}$$

$$= (\Phi_{t-1}H_{t-1} + v_t k_t^\top)H_t^{-1} \tag{48}$$

$$= \Phi_{t-1}\left(H_t - k_t k_t^\top\right)H_t^{-1} + v_t k_t^\top H_t^{-1} \tag{49}$$

$$= \Phi_{t-1} - \Phi_{t-1}k_t k_t^\top H_t^{-1} + v_t k_t^\top H_t^{-1} \tag{50}$$

$$= \Phi_{t-1} - (\Phi_{t-1}k_t - v_t)k_t^\top H_t^{-1} \tag{51}$$

$$= \Phi_{t-1} - \nabla_{\phi\phi}^2 \mathcal{L}_t(\Phi_{t-1})^{-1}\nabla_\Phi l_t^{\text{sq-err}}(\Phi_{t-1}), \tag{52}$$

recalling that $\mathcal{L}_t$ is the cumulative regularized loss (equation 4) and $l_t^{\text{sq-err}} = \|v_t^2 - \Phi k_t\|^2$. To go from equations 47 to 48, we used the fact that $\Phi_{t-1} = G_{t-1}H_{t-1}^{-1}$. From equations 48 to 49, we used the identity $H_{t-1}H_t^{-1} = (H_t - k_t k_t^\top)H_t^{-1}$.

Thus, while the DeltaNet and related layers perform (first-order) online gradient descent on a squared error loss, the Mesa layer implements instead an online (second-order) Newton descent algorithm.

# I   A PRELIMINARY INVESTIGATION INTO STATE TRACKING WITH THE MESA LAYER

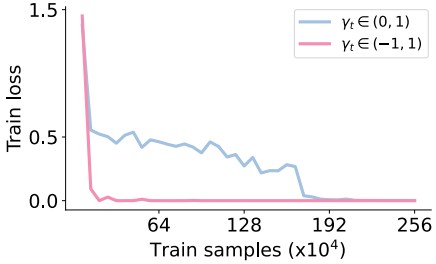
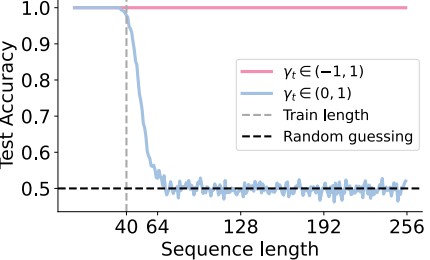

**Figure 8: Negative $\gamma_t$ and high $\Lambda$ allow MesaNets to solve parity**: When using $\gamma_t \in (-1, 1)$ as well as enforce high $\Lambda$, we enforce the MesaNet into functionality close to GLA as $q_t^* = q_t$ which allows us to use MesaNet with $\gamma_t \in (-1, 1)$ which naive applied does not lead to a well-defined mesa-optimization problem.

Recent work has investigated the (missing) state-tracking ability of transformers, modern state space models and linearized transformer RNN models, see e.g. (Merrill et al., 2025). It remains an active research direction to study under which circumstances these in-time parallelizable RNN models can better track state than transformers (Merrill & Sabharwal, 2024; Li et al., 2024).

One simple architecture change proposed in Sarrof et al. (2024); Grazzi et al. (2025) which allows layers such as Mamba, GLA or gated DeltaNet to solve certain state tracking tasks is to use forget strength $\gamma_t \in (-1, 1)$ instead of $\gamma_t \in (0, 1)$. We highlight that this change naively is not possible

to incorporate into the Mesa layer. Indeed, $\gamma_t \in (-1, 1)$ could violate the positive definiteness of $(H_{t-1}\gamma_t + k_t k_t^\top + \Lambda)$ leading to a potentially ill-defined linear system of equations problem. The Mesa layer is equivalent to GLA if $q_t^* = q_t$ which can be enforced by setting $\Lambda$ to very large values such that $(H_t + \Lambda)^{-1} \approx \Lambda^{-1}$ and rescaling $q_t$ by $\Lambda$. Although undesirably from an online learning perspective, high $\Lambda$ should lead to $(H_{t-1}\gamma_t + k_t k_t^\top + \Lambda)$ rendering positive definite even if $\gamma_t \in (-1, 1)$ leading to state tracking capabilities as observed in Grazzi et al. (2025) for models such as Mamba or DeltaNet with $\gamma_t \in (-1, 1)$. We show first state tracking results for MesaNets with $\gamma_t \in (-1, 1)$ or $\gamma_t \in (0, 1)$ while initializing $\Lambda = 50 \cdot I$ and restricting its lower value to 49. These values are chosen by hand, generally a wide range of (large) $\Lambda$ actually gave us the same results. When now learning parity, see Figure 8, MesaNets, as hypothesized, start solving parity with perfect accuracy when endowed with $\gamma_t \in (-1, 1)$, similar to results presented for Mamba and gated DeltaNet in Grazzi et al. (2025) when using $\gamma_t \in (-1, 1)$. Although this parametrization showcases the flexibility of the Mesa layer encompassing the capacity of GLA (and similar layers such as Mamba and mLSTM) by enforcing high regularization, we stress that this solution is in our opinion rather a bug than a feature. This is because we actually wish to utilize the extra flops spend to compute $q_t^*$. We leave investigating how the MesaNet could track state while not falling back to GLA functionally for future work.

**Experimental details**. We train a MesaNet with 2 layers, an embedding dimension of 128, and 4 heads per sequence mixing module (each head with dimension 128) amounting to roughly 1M parameters. For training we sample bitstrings on the fly and compute the respective ground truth parity scores at each sequence position. We then train the model to predict the parity score at each position in the sequence. During training bitstrings are restricted to a length of 40. In a final evaluation, we test the trained model on sequences up to length 256. We train on a batch size of 256 and train in the infinite data regime sampling a total of 10000 batches. We use a weight decay of 0.03 and a learning rate of 0.001. To obtain the results displayed in Figure 8 we initialize $\Lambda = 50$ and lower bound it to 49 and train once with positive eigenvalues only ($\gamma_t \in (0, 1)$) and once allowing for negative eigenvalues ($\gamma_t \in (-1, 1)$).

## J FURTHER DISCUSSION POINTS

We list here some additional discussion points which we couldn't place in the main text because of space constraints

- **Backpropagation through the conjugate gradient method**: Currently, we are computing the gradient through the Mesa layer assuming that we have approximated $q_t^*$ numerically well. We believe this current version is a shortcoming of the Mesa layer and speculate that it is actually feasible to train the MesaNet to cope better with fewer steps (and not approximate $q_t^*$ as well). For this we would use a stochastic number of CG steps during training, ranging for example from 0 to 30, and backpropagate through the unrolled process, potentially obtaining a model which is trained to be behave "optimally" given a certain number of CG steps. This would allow for an even better dynamic test-time compute allocation of the MesaNet during inference as users could flexibly decide to spend more compute for a better model. Interestingly, one could additionally condition (e.g. with a set of BOS token indicating the number of CG steps used during the forward pass) the models forward computation and therefore allow the model to learn to adjust its representation at every layer dependent on the CG steps used in the Mesa layers. We speculate that we therefore would obtain a MesaNet which behaves on par with e.g. GLA, Mamba or xLSTM with 0 CG steps and outperforms these RNNs when allocating more CG steps.

- **Architecture considerations**: We decided to benchmark related work while using the common transformer backbone allowing for a direct 1-1 comparison between all models. This architecture is extremely widespread and has the advantage to allow for a direct usage of Mixture-of-Experts Shazeer et al. (2017) layers. xLSTM and Mamba, see e.g. Beck et al. (2024), use a different backbone which notably merges the MLP layer and the RNN layer in one while matching parameter count. This architecture change leads to overall better perplexity but question if the particular RNN layer or the architecture change, or its combination offers better results. We leave an investigation of a fair comparison of the Mesa layer and other related work when changing the architecture backbone for future

work. Generally, we acknowledge that it is unclear if these architecture changes address the shortcomings of RNNs, which we show in the evaluation section, namely to incorporate sequential long range information. We are excited to study the influence of the backbone when optimizing for incorporating long-range understanding and not perplexity.

- **Learning fast matrix inversion algorithms from data**: To obtain $(H_t + \Lambda)^{-1}q_t$ we decided to use the well known and powerful conjugate gradient method. While this algorithm is widespread, we hypothesis that learning a neural network to solve $(H_t + \Lambda)^{-1}q_t$ directly or adjusting the CG method by learned parameterization, could lead to significant speed ups. We generally find extending well-known algorithms with the help of deep learning or using them as building blocks of deep neural networks an exciting research direction (von Oswald et al., 2023; 2025; Vladymyrov et al., 2024).

- **Mesa layer to model sequences outside the language domain**: We speculate that the Mesa layer is a promising layer for sequence modeling of continuous data, where in-context generalization and not memory is the driving factor of improving next token prediction. Therefore the Mesa layer might excel in domains which require some form of in-context (control or reinforcement learning) algorithm distillation (Laskin et al., 2023).

- **The fundamental limit of RNNs with finite memory**: (Modern) RNNs do have a finite amount of state which they can use to save information for future access. This has two interconnected, intermediate shortcomings when comparing to softmax: The interpretation and the relevance of certain information in a sentence can drastically change even at the last token. Since softmax stores all information of the past (all input text and its representations in all layers), it can recall information relevant to the current query (for example, a particular question about the text. RNNs need to anticipate when processing information which needs saving such that it can be accessed later on.

## K    MesaNet Trained in Synthetic Environments

We evaluate the token manipulation and in-context learning capabilities by training and evaluating MesaNets on two purely synthetic benchmarks: (i) Mechanistic Architecture Design (MAD) (Poli et al., 2024) and (ii) RegBench (Akyürek et al., 2024). For MAD, we train 2-layer models and sweep over a range of optimization hyperparameters for each task. For RegBench, we follow Akyürek et al. (2024) and sweep over a larger grid of hyperparameters for each task, including number of layers and heads, see Appendix F.

**MesaNet excels at the MAD benchmark.** MAD comprises a suite of recall, memorization, compression, and copying tasks. As shown in Table 9, the MesaNet achieves the highest average performance, outperforming all linear recurrent architectures and matching the performance of transformers. These strong results demonstrate the MesaNet's efficacy in managing its fixed-size recurrent state to store and retrieve necessary information across diverse manipulation challenges.

**MesaNet and Transformers perform on par on the RegBench.** This benchmark requires models to infer the underlying grammar of pseudo-languages, defined by probabilistic finite automata (PFAs), solely from context sequences. At test time, this in-context learning capability is tested on token sequences generated with held-out PFAs. Again, the MesaNet surpasses other RNN models and matches transformers, demonstrating its capability to infer rules at test time (Figure 9).

| | IC & Noisy Recall | Fuzzy Recall | Memorize Train Data | Selective Copy | Compress | Avg. |
|---|---|---|---|---|---|---|
| Mamba2 | 100 | 51.2 | 42.0 | 95.4 | 41.3 | 66.0 |
| GLA | 100 | 39.0 | 82.5 | 96.1 | 42.3 | 72.0 |
| xLSTM | 100 | 47.6 | 79.8 | 95.4 | 43.4 | 73.2 |
| DeltaNet | 100 | 55.5 | 40.8 | 98.8 | 43.3 | 67.7 |
| Gated DeltaNet | 100 | 32.7 | 81.7 | 95.7 | 45.0 | 71.0 |
| Hawk | 93.0 | 13.6 | 91.3 | 77.0 | 47.7 | 64.5 |
| MesaNet | 100 | 58.5 | 77.2 | 99.2 | 45.4 | 76.1 |
| Hawk-MesaNet | 100 | 30.2 | 85.6 | 99.6 | 52.3 | 73.5 |
| Transformer | 100 | 48.6 | 84.7 | 96.0 | 49.5 | 75.8 |

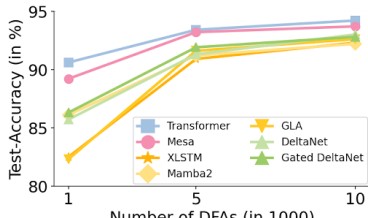

**Table 9: Performance (% Accuracy ↑) on the MAD benchmark (Poli et al., 2024).** The MesaNet performs strongly compared to other RNNs and matches the transformer.

**Figure 9: Performance on RegBench (Akyürek et al., 2024).** MesaNet outperforms other linear architectures and closes the gap to transformers.

## L    Extended Results in Language Environment

### L.1    Language Modelling / Perplexity Analyses

The common approach to measure language modeling performance on a set of sequences $S = \{s_1, \ldots, s_N\}$ is perplexity (PPL), which is defined as the exponential of the average negative log-likelihood per token (Jelinek et al., 1977; Brown et al., 2020b; Biderman et al., 2024):

$$\mathtt{NLL} = -\frac{1}{\sum_{j=1}^{|S|} |s_j|} \sum_{j=1}^{|S|} \sum_{i=1}^{|s_j|} \log P(s_{j,i}|s_{j,1}, \ldots, s_{j,i-1}) \tag{53}$$

$$\mathtt{PPL} = \exp\left[\mathtt{NLL}\right]$$

where $|S|$ is the number of sequences, $s_j$ is the j'th sequence in $S$ and $s_{j,i}$ is the i'th token in the sequence $s_j$. However, all tokens are weighted equally in these metrics, independent of their token position. This is especially critical, as the magnitudes of the log-likelihood scores tend to be quite different for early and late tokens in a sequence. As a consequence, interesting differences between models might be masked in these aggregated metrics, especially when comparing different model families with different inductive biases. Therefore, one needs to condition on the sequence position to pinpoint qualitative model differences in a quantitative manner.

**Mean-so-far {NLL, PPL}.** To investigate whether models exhibit different language modelling capabilities at different sequence depths $k$, we therefore assess mean-so-far NLL and PPL:

$$\mathtt{Mean\text{-}so\text{-}far\text{-}NLL}_{:k} = -\frac{1}{\sum_{j=1}^{|S|} \min(|s_j|, k)} \sum_{j=1}^{|S|} \sum_{i=1}^{\min(|s_j|, k)} \log P(s_{j,i}|s_{j,1}, \ldots, s_{j,i-1}) \tag{54}$$

$$\mathtt{Mean\text{-}so\text{-}far\text{-}PPL}_{:k} = \exp\left[\mathtt{Mean\text{-}so\text{-}far\text{-}NLL}_{:k}\right]$$

Intuitively, these metrics can be interpreted as **how well are sequences modeled up to length** $k$. While these metrics give a more granular picture of the loss behavior dependent on sequence length, they still mask important transition points due to the cumulative aggregation up to position $k$. For instance, the mean-so-far NLL could still be decreasing for higher $k$ (decreasing slope), despite the token-position-dependent NLL may have already plateaued or increased (Lin et al., 2025).

**Token-Position-Dependent NLL.** Consequently, we follow (Lin et al., 2025) and assess the average negative log-likelihood conditional on the token-position $k$ (for which only sum over sequences with $|s_j| \geq k$):

$$\texttt{NLL}_k = -\frac{1}{|S|} \sum_{j=1}^{|S|} \log P(s_{j,\mathbf{k}}|s_{j,1}, \dots, s_{j,k-1}). \tag{55}$$

**Difference in Token-Position-Dependent NLL Relative to a multi-head-attention transformer.** As the field's main interest is to improve upon the current state-of-the-art transformer architecture, we investigate the difference in token-position-dependent NLL with respect to a transformer (MHA):

$$\Delta\texttt{NLL}_k^{\text{model}} = \texttt{NLL}_k^{\text{model}} - \texttt{NLL}_k^{\text{MHA}}, \tag{56}$$

where a negative $\Delta\texttt{NLL}_k^{\text{model}}$ means superior language modelling ability at position $k$ relative to a transformer as the model's loss is lower. The same difference can be formulated for the mean-so-far metrics. Certainly, such a relative metric requires a well-tuned transformer baseline.

### L.1.1 WITHIN TRAIN CONTEXT-LENGTH

Here, we expand upon the results shown in Section 4.1 and present within-train-context-length language modelling evaluations on all evaluated pairs of model sizes (i.e., 145M, 400M and 1B parameters) and number of training tokens (15B and 50B tokens).

**PPL.** We present the PPL scores on the five evaluated datasets in Table 10. Across all model sizes and number of training tokens, Hawk-MesaNet exhibits the best PPL performance on the majority of benchmarks among the recurrent models, closely followed by MesaNet. Notably, Hawk-Mesa and Mesa match or exceed the transformer baseline with respect to PPL on the majority of benchmarks on all model sizes. Furthermore, one can clearly observe the impact of the attention window size on PPL based on our SWA baselines. PPL is decreasing with an increasing window size in all settings. Notably, SWA-1024 reaches competitive performance with the majority of recurrent models, i.e. Hawk, Mamba2, GLA, xLSTM and DeltaNet.

**Conditioning on the Sequence Position.** As indicated in the metrics description, and shown in Section 4.1, uniformly averaging over all tokens in the PPL computation, independent of a token's depth in a sequence, may masquerade important qualitative difference between models. Therefore, we condition on the token position and investigate the difference in token-position-dependent NLL relative to a multi-head-attention transformer $\texttt{NLL}_k^{\text{model}}$. As shown in Figure 10, most recurrent models demonstrate superior language modelling abilities early in a sequence relative to the transformer baseline. However, beyond a certain token position, transformers surpass the performance of all recurrent models.

- **Which model performs strongest early in the sequence?** Notably, MesaNet and Hawk-MesaNet exhibit the strong performance early-in-the-sequence tokens except Hawk. However, while Hawk exhibits the best performance up to a certain depth, the model exhibits a sharp performance decline after that and falls behind most models. See Figure 11 for a clearer visualization (equivalent to Figure 10, but token-position in log-scale).

- **Which model offers superior performance to a transformer "for the longest"?** While Hawk losses its advantage the earliest, Hawk-MesaNet extends the performance advantage to the largest token depths, closely followed by MesaNet.

For completeness, we also show the mean-so-far NLL difference $\Delta\texttt{Mean-so-far-NLL}_{:k}^{\text{model}}$ relative to a Transformer in Figure 12. However, as indicated, the cummulative aggregation in the metric skews the important token depth transition point where a transformer surpasses the recurrent models in terms of language modeling.

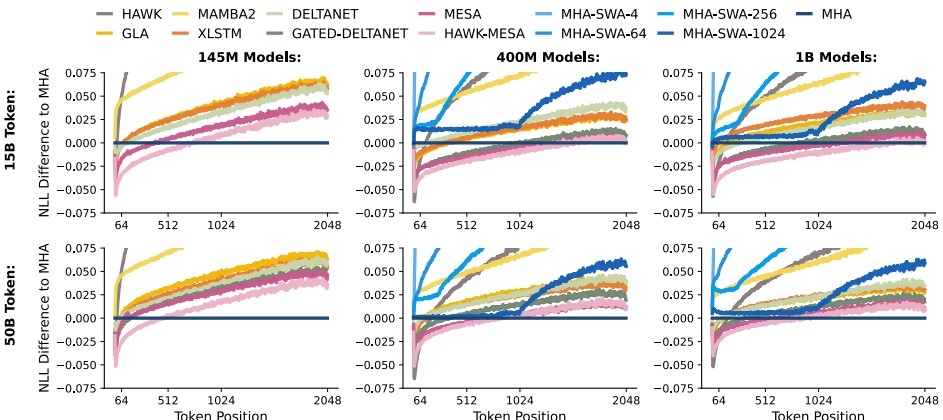

**Figure 10: NLL Difference (per token-position) $\Delta\text{NLL}_k^{\text{model}}$ relative to a Transformer on SlimPajama Validaton Dataset.** Most recurrent models demonstrate superior language modelling abilities early in a sequence relative to the transformer baseline, across all settings. However, beyond a certain token position, transformers surpass the performance of all recurrent models.

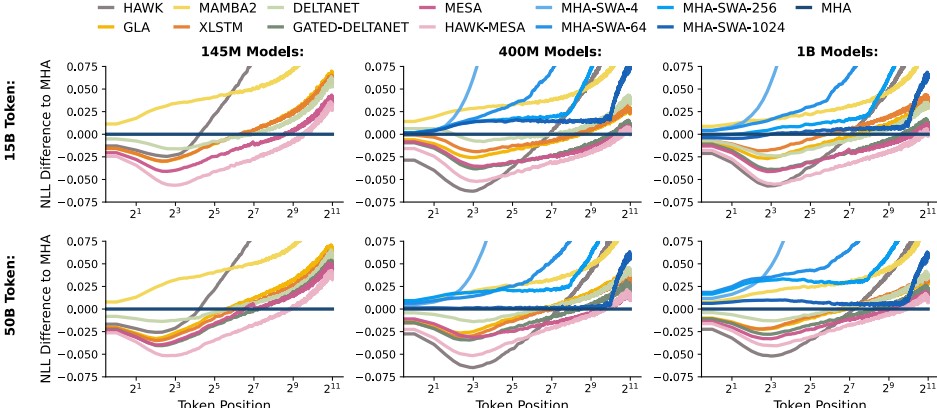

**Figure 11: NLL Difference (per token-position) $\Delta\text{NLL}_k^{\text{model}}$ relative to a Transformer on SlimPajama Validaton Dataset in log-scale.** MesaNet and Hawk-MesaNet exhibit the strong language modeling performance early-in-the-sequence tokens except Hawk. While Hawk exhibits the best performance up to a certain depth, the model exhibits a sharp performance decline relatively early in the seq. depth.

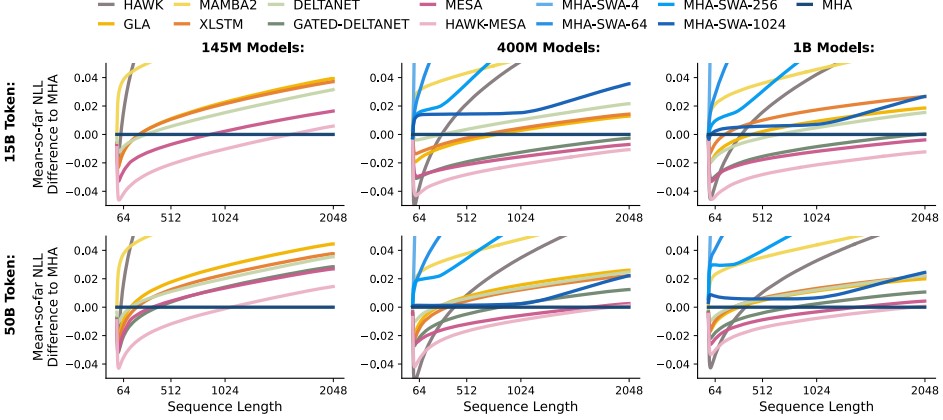

**Figure 12: Mean-so-far NLL Difference $\Delta\text{Mean-so-far-NLL}_{:k}^{\text{model}}$ relative to a Transformer on SlimPajama Validaton Dataset.** The cummulative aggregation in the mean-so-far metric skews the important token depth transition point where a transformer surpasses the recurrent models in terms of language modeling.

| | | 15B Tokens | | | | | | | 50B Tokens | | | | | | |
|---|---|---|---|---|---|---|---|---|---|---|---|---|---|---|---|
| | | SLIM ppl↓ | LMB. ppl↓ | WIKI. ppl↓ | PG19 ppl↓ | GOV. ppl↓ | QASP. ppl↓ | AVG ppl↓ | SLIM ppl↓ | LMB. ppl↓ | WIKI. ppl↓ | PG19 ppl↓ | GOV. ppl↓ | QASP. ppl↓ | AVG ppl↓ |
| **145M** | - Hawk | 19.73 | 38.94 | 23.06 | 19.87 | 19.23 | 29.66 | 25.08 | 18.34 | 37.43 | 21.25 | 18.49 | 18.17 | 27.83 | 23.59 |
| | - Mamba2 | 18.29 | 40.34 | 20.86 | 19.17 | 17.03 | 23.71 | 23.23 | 17.05 | 38.22 | 19.24 | 17.87 | 15.90 | 22.10 | 21.73 |
| | - GLA | 17.37 | 37.96 | 19.57 | 18.11 | 15.86 | 22.37 | 21.87 | 16.30 | 36.20 | 18.43 | 16.90 | 15.02 | 20.91 | 20.62 |
| | - xLSTM | 17.35 | 37.97 | 19.57 | 18.12 | 15.88 | 22.50 | 21.90 | 16.20 | 36.19 | 18.31 | 16.97 | 14.91 | 20.85 | 20.57 |
| | - DeltaNet | 17.26 | 38.18 | 19.29 | 17.93 | 15.67 | 21.75 | 21.68 | 16.17 | 36.55 | 18.08 | 16.78 | 14.81 | 20.53 | 20.49 |
| | - Gated-DeltaNet | 17.12 | 37.62 | 19.18 | 17.77 | 15.55 | 22.13 | 21.56 | 16.05 | 35.80 | 18.04 | 16.79 | 14.77 | 20.67 | 20.35 |
| | - Mesa | 17.02 | 37.64 | 19.10 | 17.72 | 15.44 | 21.87 | 21.47 | 16.05 | 36.17 | 17.96 | 16.60 | 14.72 | 20.57 | 20.34 |
| | - Hawk-Mesa | 16.81 | 37.20 | 18.87 | 17.14 | 15.29 | 21.62 | 21.15 | 15.82 | 35.51 | 17.70 | 16.19 | 14.55 | 20.38 | 20.02 |
| | - Transformer | 16.95 | 38.69 | 18.65 | 17.47 | 15.00 | 20.80 | 21.26 | 15.81 | 36.54 | 17.35 | 16.25 | 14.04 | 19.33 | 19.89 |
| **400M** | - Hawk | 14.40 | 31.54 | 16.12 | 14.23 | 13.67 | 19.85 | 18.30 | 12.87 | 29.44 | 14.30 | 12.71 | 12.24 | 17.54 | 16.52 |
| | - Mamba2 | 14.45 | 33.38 | 15.99 | 14.80 | 13.27 | 18.36 | 18.37 | 13.07 | 31.05 | 14.28 | 13.28 | 12.10 | 16.37 | 16.69 |
| | - GLA | 13.69 | 31.64 | 15.01 | 13.89 | 12.36 | 17.08 | 17.28 | 12.61 | 29.93 | 13.73 | 12.75 | 11.52 | 15.77 | 16.05 |
| | - xLSTM | 13.71 | 31.70 | 14.95 | 13.88 | 12.28 | 17.10 | 17.27 | 12.56 | 29.79 | 13.60 | 12.72 | 11.49 | 15.72 | 15.98 |
| | - DeltaNet | 13.80 | 31.98 | 15.07 | 14.01 | 12.51 | 17.20 | 17.43 | 12.59 | 30.00 | 13.68 | 12.70 | 11.49 | 15.57 | 16.00 |
| | - Gated-DeltaNet | 13.48 | 31.40 | 14.71 | 13.59 | 12.16 | 16.64 | 17.00 | 12.44 | 29.57 | 13.45 | 12.52 | 11.31 | 15.42 | 15.79 |
| | - Mesa | 13.44 | 31.38 | 14.65 | 13.51 | 12.02 | 16.56 | 16.93 | 12.34 | 29.57 | 13.36 | 12.40 | 11.15 | 15.19 | 15.67 |
| | - Hawk-Mesa | 13.37 | 31.10 | 14.55 | 13.32 | 12.07 | 16.68 | 16.85 | 12.30 | 29.38 | 13.33 | 12.30 | 11.28 | 15.32 | 15.65 |
| | - SWA-4 | 23.36 | 38.65 | 29.29 | 23.51 | 26.94 | 48.24 | 31.66 | 19.32 | 33.76 | 23.43 | 19.35 | 21.50 | 35.41 | 25.46 |
| | - SWA-64 | 15.98 | 32.97 | 18.89 | 16.31 | 15.20 | 23.08 | 20.40 | 14.04 | 30.51 | 16.35 | 14.19 | 13.25 | 19.37 | 17.95 |
| | - SWA-256 | 14.69 | 32.64 | 16.99 | 15.04 | 13.42 | 19.36 | 18.69 | 13.23 | 30.36 | 14.94 | 13.38 | 12.08 | 17.09 | 16.85 |
| | - SWA-1024 | 13.95 | 32.63 | 15.40 | 14.09 | 12.36 | 17.58 | 17.58 | 12.52 | 30.13 | 13.71 | 12.56 | 11.12 | 15.26 | 15.88 |
| | - Transformer | 13.64 | 32.25 | 14.71 | 13.73 | 12.06 | 16.51 | 17.15 | 12.40 | 30.10 | 13.23 | 12.42 | 10.96 | 14.84 | 15.66 |
| **1B** | - Hawk | 12.71 | 28.72 | 13.95 | 12.44 | 11.90 | 17.30 | 16.17 | 11.24 | 26.67 | 12.23 | 10.93 | 10.63 | 14.89 | 14.43 |
| | - Mamba2 | 12.78 | 30.30 | 13.97 | 12.92 | 11.68 | 15.97 | 16.27 | 11.39 | 28.02 | 12.23 | 11.42 | 10.42 | 14.02 | 14.58 |
| | - GLA | 12.28 | 29.13 | 13.29 | 12.35 | 11.08 | 15.20 | 15.55 | 10.99 | 26.98 | 11.77 | 10.95 | 9.99 | 13.52 | 14.03 |
| | - xLSTM | 12.38 | 29.21 | 13.43 | 12.40 | 11.16 | 15.33 | 15.65 | 11.01 | 26.93 | 11.81 | 10.94 | 10.00 | 13.55 | 14.04 |
| | - DeltaNet | 12.23 | 29.13 | 13.20 | 12.28 | 11.04 | 15.11 | 15.50 | 11.01 | 27.08 | 11.73 | 11.00 | 10.02 | 13.44 | 14.05 |
| | - Gated-DeltaNet | 12.06 | 28.67 | 13.00 | 12.05 | 10.85 | 14.86 | 15.25 | 10.89 | 26.79 | 11.58 | 10.81 | 9.88 | 13.28 | 13.87 |
| | - Mesa | 12.02 | 28.57 | 12.92 | 11.96 | 10.76 | 14.76 | 15.17 | 10.83 | 26.78 | 11.49 | 10.71 | 9.80 | 13.13 | 13.79 |
| | - Hawk-Mesa | 11.91 | 28.45 | 12.79 | 11.83 | 10.72 | 14.60 | 15.05 | 10.78 | 26.59 | 11.53 | 10.60 | 9.79 | 13.20 | 13.75 |
| | - SWA-4 | 20.27 | 34.66 | 24.56 | 20.33 | 22.98 | 40.37 | 27.20 | 16.46 | 29.93 | 19.42 | 16.42 | 17.86 | 29.15 | 21.54 |
| | - SWA-64 | 14.08 | 30.01 | 16.47 | 14.33 | 13.34 | 19.78 | 18.00 | 12.37 | 27.76 | 14.14 | 12.51 | 11.56 | 16.77 | 15.85 |
| | - SWA-256 | 12.98 | 29.63 | 14.76 | 13.18 | 11.82 | 16.82 | 16.53 | 11.60 | 27.39 | 12.89 | 11.71 | 10.58 | 14.69 | 14.81 |
| | - SWA-1024 | 12.33 | 29.65 | 13.47 | 12.35 | 10.92 | 14.93 | 15.61 | 11.00 | 27.22 | 11.78 | 10.92 | 9.79 | 13.11 | 13.97 |
| | - Transformer | 12.16 | 29.55 | 12.90 | 12.10 | 10.68 | 14.47 | 15.31 | 10.86 | 27.16 | 11.42 | 10.74 | 9.69 | 12.86 | 13.79 |

**Table 10: PPL at a Maximum Sequence Length of 2048.** The score of the best recurrent model with respect to PPL on each dataset is highlighted, and PPL scores from SWA and the transformer baseline are shown as reference. Across all model sizes and number of training tokens, Hawk-Mesa exhibits the best PPL performance on most benchmarks, closely followed by Mesa.

### L.1.2 BEYOND TRAIN CONTEXT-LENGTH

**PPL.** We present the PPL scores for context lengths of 4k (see Table 11) and 32k (see Table 12) respectively on all model sizes and number of training tokens.

| | | 15B Tokens | | | | | 50B Tokens | | | | |
|---|---|---|---|---|---|---|---|---|---|---|---|
| | | WIKI. ppl↓ | PG19 ppl↓ | GOV. ppl↓ | QASP. ppl↓ | AVG ppl↓ | WIKI. ppl↓ | PG19 ppl↓ | GOV. ppl↓ | QASP. ppl↓ | AVG ppl↓ |
| 145M | - Hawk | 23.80 | 24.23 | 19.64 | 30.09 | 24.44 | 21.90 | 22.63 | 18.54 | 28.10 | 22.79 |
| | - Mamba2 | 24.28 | 27.31 | 20.07 | 27.51 | 24.79 | 24.13 | 27.85 | 22.56 | 29.17 | 25.93 |
| | - GLA | 20.07 | 22.14 | 15.68 | 21.38 | 19.82 | 18.83 | 20.70 | 14.73 | 19.95 | 18.55 |
| | - xLSTM | 20.04 | 22.13 | 15.56 | 21.43 | 19.79 | 18.68 | 20.67 | 14.61 | 19.89 | 18.46 |
| | - DeltaNet | 19.85 | 22.05 | 15.47 | 20.85 | 19.55 | 18.66 | 20.64 | 14.64 | 19.76 | 18.42 |
| | - Gated-DeltaNet | 19.64 | 21.75 | 15.23 | 21.03 | 19.41 | 18.46 | 20.47 | 14.45 | 19.63 | 18.25 |
| | - Mesa | 19.52 | 21.60 | 15.10 | 20.78 | 19.25 | 18.38 | 20.25 | 14.42 | 19.52 | 18.14 |
| | - Hawk-Mesa | 19.33 | 20.86 | 15.03 | 20.69 | 18.98 | 18.15 | 19.72 | 14.31 | 19.48 | 17.91 |
| | - Transformer | 27.68 | 34.18 | 23.59 | 30.77 | 29.06 | 52.12 | 65.58 | 47.93 | 59.37 | 56.25 |
| 400M | - Hawk | 16.61 | 17.35 | 13.80 | 19.73 | 16.87 | 14.70 | 15.45 | 12.33 | 17.35 | 14.96 |
| | - Mamba2 | 18.31 | 20.59 | 15.33 | 20.59 | 18.70 | 17.94 | 20.75 | 16.07 | 20.48 | 18.81 |
| | - GLA | 15.31 | 16.84 | 12.08 | 16.20 | 15.11 | 14.05 | 15.43 | 11.26 | 14.95 | 13.92 |
| | - xLSTM | 15.31 | 16.82 | 11.98 | 16.18 | 15.07 | 13.90 | 15.39 | 11.22 | 14.87 | 13.85 |
| | - DeltaNet | 15.49 | 17.07 | 12.27 | 16.37 | 15.30 | 14.09 | 15.50 | 11.35 | 14.86 | 13.95 |
| | - Gated-DeltaNet | 14.99 | 16.46 | 11.84 | 15.73 | 14.76 | 13.75 | 15.13 | 11.04 | 14.60 | 13.63 |
| | - Mesa | 15.02 | 16.41 | 11.73 | 15.72 | 14.72 | 13.67 | 14.98 | 10.87 | 14.36 | 13.47 |
| | - Hawk-Mesa | 14.90 | 16.15 | 11.82 | 15.86 | 14.68 | 13.67 | 14.83 | 11.05 | 14.54 | 13.52 |
| | - SWA-4 | 30.09 | 29.68 | 28.80 | 50.69 | 34.82 | 24.31 | 24.55 | 22.88 | 37.16 | 27.23 |
| | - SWA-64 | 19.58 | 20.23 | 15.65 | 23.38 | 19.71 | 16.93 | 17.48 | 13.55 | 19.44 | 16.85 |
| | - SWA-256 | 17.54 | 18.41 | 13.59 | 19.29 | 17.21 | 15.47 | 16.44 | 12.19 | 16.88 | 15.25 |
| | - SWA-1024 | 15.90 | 17.28 | 12.32 | 16.58 | 15.52 | 14.22 | 15.41 | 11.27 | 14.92 | 13.95 |
| | - Transformer | 33.17 | 46.81 | 34.34 | 41.51 | 38.96 | 74.74 | 130.23 | 122.52 | 142.67 | 117.54 |
| 1B | - Hawk | 14.37 | 15.11 | 12.01 | 17.10 | 14.65 | 12.59 | 13.25 | 10.67 | 14.68 | 12.80 |
| | - Mamba2 | 15.90 | 18.03 | 13.33 | 17.85 | 16.28 | 17.56 | 20.90 | 16.28 | 19.98 | 18.68 |
| | - GLA | 13.56 | 14.90 | 10.81 | 14.37 | 13.41 | 12.05 | 13.15 | 9.77 | 12.80 | 11.94 |
| | - xLSTM | 13.71 | 14.98 | 10.88 | 14.54 | 13.53 | 12.11 | 13.15 | 9.79 | 12.86 | 11.98 |
| | - DeltaNet | 13.55 | 14.90 | 10.82 | 14.30 | 13.39 | 12.11 | 13.32 | 9.84 | 12.79 | 12.02 |
| | - Gated DelaNet | 13.26 | 14.50 | 10.56 | 14.01 | 13.08 | 11.86 | 12.98 | 9.62 | 12.54 | 11.75 |
| | - Mesa | 13.21 | 14.43 | 10.50 | 13.93 | 13.02 | 11.78 | 12.90 | 9.57 | 12.43 | 11.67 |
| | - Hawk-Mesa | 13.08 | 14.27 | 10.49 | 13.85 | 12.92 | 11.81 | 12.72 | 9.60 | 12.53 | 11.66 |
| | - SWA-4 | 25.40 | 25.64 | 24.58 | 42.51 | 29.53 | 20.17 | 20.71 | 18.99 | 30.44 | 22.58 |
| | - SWA-64 | 17.05 | 17.70 | 13.74 | 20.02 | 17.13 | 14.66 | 15.34 | 11.81 | 16.84 | 14.66 |
| | - SWA-256 | 15.25 | 16.11 | 11.98 | 16.71 | 15.01 | 13.33 | 14.24 | 10.65 | 14.49 | 13.18 |
| | - SWA-1024 | 13.89 | 15.03 | 10.84 | 14.45 | 13.56 | 12.20 | 13.27 | 9.75 | 12.71 | 11.98 |
| | - Transformer | 24.40 | 31.60 | 24.06 | 30.51 | 27.64 | 46.14 | 64.04 | 57.04 | 74.80 | 60.50 |

**Table 11: PPL at a Maximum Sequence Length of 4k.**

| | | 15B Tokens | | | | | 50B Tokens | | | | |
|---|---|---|---|---|---|---|---|---|---|---|---|
| | | WIKI. ppl↓ | PG19 ppl↓ | GOV. ppl↓ | QASP. ppl↓ | AVG ppl↓ | WIKI. ppl↓ | PG19 ppl↓ | GOV. ppl↓ | QASP. ppl↓ | AVG ppl↓ |
| 145M | - Hawk | 23.93 | 29.50 | 20.16 | 30.73 | 26.08 | 21.98 | 27.62 | 19.01 | 28.78 | 24.35 |
| | - Mamba2 | 37.56 | 96.96 | 44.95 | 38.47 | 54.48 | 49.51 | 174.03 | 106.47 | 50.52 | 95.13 |
| | - GLA | 20.28 | 27.32 | 16.21 | 21.40 | 21.30 | 18.96 | 26.30 | 15.23 | 20.09 | 20.15 |
| | - xLSTM | 20.30 | 28.02 | 15.91 | 21.61 | 21.46 | 18.78 | 26.25 | 15.11 | 20.02 | 20.04 |
| | - DeltaNet | 25.11 | 979.34 | 43.10 | 24.93 | 268.12 | 26.79 | 883.32 | 52.20 | 26.31 | 247.16 |
| | - Gated-DeltaNet | 19.73 | 27.03 | 15.46 | 21.05 | 20.82 | 18.59 | 27.27 | 14.77 | 19.77 | 20.10 |
| | - Mesa | 19.70 | 26.67 | 15.26 | 20.79 | 20.61 | 18.58 | 25.72 | 14.65 | 19.62 | 19.64 |
| | - Hawk-Mesa | 19.72 | 26.79 | 15.69 | 20.97 | 20.79 | 18.44 | 26.09 | 14.69 | 19.99 | 19.80 |
| | - Transformer | 42.42 | 72.04 | 43.19 | 41.64 | 49.82 | 528.05 | 4436.78 | 2029.43 | 324.84 | 1829.77 |
| 400M | - Hawk | 16.65 | 21.10 | 14.04 | 20.10 | 17.97 | 14.72 | 18.82 | 12.53 | 17.64 | 15.93 |
| | - Mamba2 | 26.64 | 65.40 | 34.37 | 28.00 | 38.60 | 53.90 | 919.97 | 172.39 | 41.73 | 297.00 |
| | - GLA | 15.43 | 23.08 | 12.76 | 16.33 | 16.90 | 14.25 | 20.36 | 11.74 | 15.08 | 15.36 |
| | - xLSTM | 15.34 | 20.86 | 12.02 | 16.20 | 16.11 | 14.00 | 20.21 | 11.29 | 14.97 | 15.12 |
| | - DeltaNet | 18.59 | 487.01 | 28.09 | 19.28 | 138.24 | 19.13 | 359.90 | 31.71 | 17.98 | 107.18 |
| | - Gated-DeltaNet | 15.16 | 21.19 | 12.27 | 15.85 | 16.12 | 13.82 | 20.72 | 11.37 | 14.67 | 15.14 |
| | - Mesa | 15.40 | 21.94 | 12.31 | 15.98 | 16.40 | 13.83 | 19.55 | 11.17 | 14.51 | 14.77 |
| | - Hawk-Mesa | 15.43 | 22.70 | 12.98 | 16.40 | 16.88 | 14.04 | 21.61 | 12.27 | 15.04 | 18.24 |
| | - SWA-4 | 30.07 | 37.94 | 29.66 | 52.16 | 37.46 | 24.29 | 31.49 | 23.59 | 38.40 | 29.44 |
| | - SWA-64 | 19.69 | 25.07 | 16.01 | 23.90 | 21.17 | 16.98 | 21.53 | 13.81 | 19.83 | 18.04 |
| | - SWA-256 | 17.63 | 22.43 | 13.82 | 19.62 | 18.38 | 15.59 | 20.07 | 12.37 | 17.17 | 16.30 |
| | - SWA-1024 | 16.01 | 21.02 | 12.40 | 16.73 | 16.54 | 14.48 | 19.01 | 11.89 | 15.26 | 15.16 |
| | - Transformer | 118.84 | 538.89 | 188.16 | 94.22 | 235.03 | 428.15 | 4312.79 | 2013.32 | 473.55 | 1806.95 |
| 1B | - Hawk | 14.40 | 18.44 | 12.20 | 17.42 | 15.61 | 12.62 | 16.07 | 10.84 | 14.95 | 13.62 |
| | - Mamba2 | 21.43 | 48.14 | 23.28 | 23.01 | 28.96 | 47.30 | 240.81 | 101.96 | 39.52 | 107.40 |
| | - GLA | 13.61 | 18.72 | 10.96 | 14.44 | 14.43 | 12.11 | 16.85 | 9.98 | 12.89 | 12.96 |
| | - xLSTM | 13.74 | 18.38 | 10.91 | 14.58 | 14.40 | 12.20 | 16.95 | 10.02 | 13.03 | 13.05 |
| | - DeltaNet | 14.75 | 145.22 | 17.33 | 15.54 | 48.21 | 14.65 | 150.90 | 21.92 | 14.95 | 50.60 |
| | - Gated DeltaNet | 13.25 | 17.75 | 10.55 | 13.97 | 13.88 | 11.87 | 15.77 | 9.60 | 12.53 | 12.44 |
| | - Mesa | 13.35 | 18.17 | 10.80 | 14.04 | 14.09 | 11.92 | 16.29 | 9.71 | 12.58 | 12.63 |
| | - Hawk-Mesa | 13.57 | 139.08 | 19.41 | 14.55 | 46.65 | 12.31 | 17.50 | 17.51 | 13.03 | 15.09 |
| | - SWA-4 | 25.35 | 32.78 | 25.33 | 43.92 | 31.85 | 20.15 | 26.44 | 19.55 | 31.49 | 24.41 |
| | - SWA-64 | 17.10 | 21.83 | 14.05 | 20.49 | 18.37 | 14.68 | 18.83 | 12.03 | 17.21 | 15.69 |
| | - SWA-256 | 15.31 | 19.61 | 12.17 | 17.00 | 16.02 | 13.39 | 17.28 | 10.78 | 14.71 | 14.04 |
| | - SWA-1024 | 13.93 | 18.15 | 10.84 | 14.58 | 14.38 | 12.27 | 16.04 | 9.80 | 12.87 | 12.75 |
| | - Transformer | 48.41 | 119.56 | 56.09 | 53.95 | 69.50 | 228.12 | 1326.59 | 563.97 | 234.95 | 588.41 |

**Table 12: PPL at a Maximum Sequence Length of 32k.**

| | Global Subset | | | | | Local Subset | | | | | | | | |
|---|---|---|---|---|---|---|---|---|---|---|---|---|---|---|
| | LMB. acc ↑ | Hella. acc ↑ | RACE-M acc ↑ | RACE-H acc ↑ | AVG | PIQA acc ↑ | Wino acc ↑ | ARC-E acc ↑ | ARC-C acc ↑ | SIQA acc ↑ | BOOLQ acc ↑ | OBQA acc ↑ | SC. acc ↑ | AVG |
| **400M Parameters / 15B Tokens** | | | | | | | | | | | | | | |
| - SWA-4 | 4,62 | 34,97 | 25,97 | 25,93 | 22,87 | 66,81 | 49,33 | 43,81 | 24,23 | 39,82 | 57,31 | 30,00 | 63,78 | 46,89 |
| - SWA-16 | 27,11 | 37,20 | 28,18 | 28,04 | 30,13 | 67,63 | 52,64 | 43,52 | 23,81 | 39,71 | 54,89 | 27,60 | 65,82 | 46,95 |
| - SWA-64 | 38,54 | 39,35 | 32,87 | 30,24 | 35,25 | 68,93 | 52,17 | 44,40 | 22,87 | 39,76 | 58,56 | 29,20 | 64,99 | 47,61 |
| - SWA-256 | 40,52 | 40,44 | 34,25 | 31,48 | 36,67 | 69,21 | 50,67 | 43,35 | 24,91 | 40,89 | 56,82 | 30,20 | 66,90 | 47,87 |
| - SWA-1024 | 41,43 | 40,90 | 37,57 | 34,26 | 38,54 | 67,90 | 52,80 | 44,49 | 22,61 | 40,58 | 60,37 | 30,20 | 66,58 | 48,19 |
| - Transformer | 41,12 | 41,27 | 37,29 | 34,45 | 38,53 | 68,23 | 51,07 | 44,28 | 24,57 | 40,23 | 58,10 | 28,40 | 66,58 | 47,68 |
| **400M Parameters / 50B Tokens** | | | | | | | | | | | | | | |
| - SWA-4 | 18,28 | 39,02 | 29,56 | 27,66 | 28,63 | 67,85 | 51,93 | 44,49 | 24,83 | 39,71 | 58,23 | 32,40 | 66,14 | 48,20 |
| - SWA-16 | 35,03 | 41,52 | 29,01 | 28,33 | 33,47 | 68,99 | 52,72 | 45,88 | 24,32 | 39,56 | 57,40 | 33,00 | 67,54 | 48,68 |
| - SWA-64 | 42,34 | 44,14 | 34,53 | 31,67 | 38,17 | 69,53 | 53,75 | 45,24 | 24,74 | 40,28 | 56,45 | 31,60 | 68,49 | 48,76 |
| - SWA-256 | 43,86 | 45,31 | 36,46 | 35,79 | 40,36 | 70,24 | 52,33 | 45,79 | 23,98 | 40,23 | 57,00 | 32,40 | 68,94 | 48,86 |
| - SWA-1024 | 45,08 | 46,43 | 38,95 | 34,74 | 41,30 | 69,64 | 52,25 | 45,71 | 25,00 | 40,07 | 57,92 | 32,20 | 67,92 | 48,84 |
| - Transformer | 44,96 | 46,30 | 41,44 | 35,89 | 42,15 | 69,91 | 52,64 | 45,96 | 24,06 | 40,48 | 57,31 | 30,40 | 69,64 | 48,80 |
| **1B Parameters / 15B Tokens** | | | | | | | | | | | | | | |
| - SWA-4 | 8,46 | 38,56 | 27,62 | 27,18 | 25,46 | 67,95 | 51,30 | 46,72 | 23,72 | 40,17 | 56,73 | 30,40 | 65,50 | 47,81 |
| - SWA-16 | 33,81 | 41,52 | 28,73 | 27,66 | 32,93 | 68,77 | 52,64 | 47,26 | 24,32 | 40,28 | 55,26 | 33,40 | 67,41 | 48,67 |
| - SWA-64 | 42,60 | 44,04 | 31,49 | 30,72 | 37,21 | 69,91 | 51,30 | 46,72 | 24,66 | 41,10 | 58,56 | 33,20 | 67,98 | 49,18 |
| - SWA-256 | 45,82 | 45,64 | 35,91 | 34,35 | 40,43 | 69,86 | 52,09 | 47,26 | 25,26 | 41,91 | 58,53 | 31,40 | 69,06 | 49,42 |
| - SWA-1024 | 45,06 | 46,23 | 39,50 | 34,74 | 41,38 | 70,29 | 53,99 | 47,39 | 24,15 | 40,94 | 59,54 | 30,60 | 69,00 | 49,49 |
| - Transformer | 45,31 | 46,65 | 41,16 | 35,79 | 42,23 | 70,78 | 52,25 | 48,19 | 23,55 | 40,28 | 52,91 | 31,40 | 67,98 | 48,42 |
| **1B Parameters / 50B Tokens** | | | | | | | | | | | | | | |
| - SWA-4 | 24,63 | 44,90 | 28,18 | 27,08 | 31,20 | 70,35 | 52,49 | 48,19 | 24,83 | 39,56 | 60,15 | 32,80 | 68,56 | 49,62 |
| - SWA-16 | 39,03 | 48,10 | 28,73 | 29,47 | 36,33 | 72,09 | 53,04 | 48,99 | 25,43 | 41,15 | 53,39 | 32,80 | 70,78 | 49,71 |
| - SWA-64 | 46,11 | 51,30 | 38,40 | 33,49 | 42,33 | 71,87 | 53,35 | 49,62 | 26,71 | 40,74 | 56,70 | 33,40 | 71,74 | 50,52 |
| - SWA-256 | 50,28 | 52,08 | 40,88 | 35,69 | 44,74 | 72,20 | 52,64 | 49,37 | 27,05 | 40,84 | 58,35 | 32,80 | 73,01 | 50,78 |
| - SWA-1024 | 50,38 | 53,69 | 41,44 | 37,22 | 45,68 | 72,47 | 53,35 | 49,41 | 27,13 | 41,61 | 62,20 | 32,60 | 72,06 | 51,35 |
| - Transformer | 48,92 | 53,63 | 42,27 | 37,32 | 45,54 | 72,31 | 54,62 | 49,41 | 28,24 | 40,17 | 60,73 | 35,20 | 72,25 | 51,62 |
| - Random | ≈ 0 | 25,00 | 25,00 | 25,00 | - | 50.00 | 50.00 | 25.00 | 25.00 | 33.33 | 50.00 | 25.00 | 50.00 | - |

**Table 13: Reference Scores of Sliding Window Attention (`SWA`) Models on Common-Sense Reasoning Benchmarks.** On LAMBADA, HellaSwag and RACE-M and RACE-H, we observe significant performance increases with a growing attention window. On the remaining benchmarks, we only observe marginal performance differences between a Transformer with a sliding window-size of 4 (`SWA-4`) and a full-window attention Transformer (attention window of 2048). We highlight the scores of the first short-range `SWA` model (window sizes = {4,16,64}) that matches or exceeds the Transformer performance.

## L.2 DOWNSTREAM BENCHMARKS

To evaluate the performance of the investigated models on downstream task, we investigate three classes of benchmarks:

- **Zero-Shot Common-Sense Reasoning Benchmarks (Section L.2.1)**

- **In-Context Recall Benchmarks ( Section L.2.2)**

- **Few-Shot Learning Benchmarks (Section L.2.3)**

Within each benchmark section, we report all raw numbers on all model sizes and number of training tokens, and complement them with reference scores of Sliding-Window Attention models with varying attention-window sizes.

### L.2.1 ZERO-SHOT COMMON-SENSE REASONING BENCHMARKS

When tracking the performance of "many models" on "many benchmarks", it is common to resort to aggregated benchmark scores. However, aggregated scores tend to masquerade important sub-trends and limit our understanding (Burnell et al., 2023). For instance, prior work (Gu & Dao, 2024; Yang et al., 2024a; Beck et al., 2024) averages over a set of common-sense reasoning benchmarks. However, evaluations with 400M and 1B Sliding-Window Attention (SWA) models with different attention-window sizes reveal that competitive, or even superior, scores on a subset of these benchmarks can be attained with an attention windows as short as $4, 16$ or $64$ (see Table 13). This observation strongly indicates that a subset of these benchmarks are either exploitable by short-range language heuristics, and do not require longer-range language modeling capabilities to reach competitive scores, or are simply too hard such that we end up measuring noise.

**Splitting Reasoing Benchmark into Two Groups.** To reduce the potential benchmark noise and deconfound the results, we aim to split the benchmark into two subsets. Therefore, we employ the following benchmark splitting protocol:

1. **Reference Scores.** Run every selected benchmark on SWA-{4,16,64} models and a transformer model (attention window of size 2048) on 400M and 1B parameters trained on 15B or 50B tokens each.

2. **Splitting Conditions.** We then assess the following splitting conditions:
   - **Condition 1:** Analyze for every benchmark whether benchmark scores increase with increasing attention windows (from SWA-4 to SWA-64).
   - **Condition 2:** Verify whether no short-range SWA model (window sizes = 4, 16 and 64) outperforms the transformer baseline with an attention windows of 2048.

3. **Benchmark Grouping.** Finally, we split the benchmark into two subsets:
   - **Local Reasoning Benchmark Set:** One of the above conditions is violated.
   - **Global Reasoning Benchmark Set:** None of the above conditions is violated.

We refer to Table 13 for a detailed score breakdown, including two additional SWA reference models (SWA-256 and SWA-1024). Additionally, we want to highlight that these findings, and the benchmark splitting, are based on experiments 400M and 1B models trained on SlimPajama (Soboleva et al., 2023). The benchmark splitting is likely to change slightly when training with bigger model sizes or on different datasets.

**Results on all Model Configurations.** We report the full set of benchmark scores on all model configuration (model sizes and number of training tokens) in Table 14. Across all settings, we observe similar trends – MesaNet and Hawk-MesaNet show strong performance especially on the global reasoning benchmark set. Among the remaining recurrent models, only Gated DeltaNet reaches competitive scores with MesaNet on this benchmark subset. In contrast, we do not observe much score variation on the local reasoning benchmark set. Hawk, the worst performing model on the global set, reaches competitive or even close-to-best scores within this set on average. This confirms the hypothesis that this set of benchmark are likely to measure different aspects of language modeling, or are potentially noisy, or are not suited for our models as they might be still too challenging.

| Model | Global Subset | | | | | Local Subset | | | | | | | | |
|---|---|---|---|---|---|---|---|---|---|---|---|---|---|---|
| | LMB. acc ↑ | Hella. acc ↑ | RACE-M acc ↑ | RACE-H acc ↑ | AVG | PIQA acc ↑ | Wino acc ↑ | ARC-E acc ↑ | ARC-C acc ↑ | SIQA acc ↑ | BOOLQ acc ↑ | OBQA acc ↑ | SC. acc ↑ | AVG |
| **145M Models / 15B T.** | | | | | | | | | | | | | | |
| - Hawk | 21,87 | 33,54 | 29,01 | 28,52 | 28,23 | 64,64 | 50,83 | 40,24 | 21,93 | 39,41 | 59,11 | 27,80 | 62,25 | 45,78 |
| - Mamba2 | 27,83 | 33,21 | 32,04 | 30,53 | 30,90 | 64,47 | 50,36 | 39,27 | 22,27 | 39,00 | 51,44 | 26,40 | 62,13 | 44,42 |
| - GLA | 31,05 | 34,20 | 33,43 | 28,71 | 31,84 | 63,66 | 52,09 | 41,41 | 21,76 | 38,89 | 56,85 | 28,80 | 63,97 | 45,93 |
| - xLSTM | 31,19 | 34,41 | 30,94 | 29,47 | 31,50 | 65,13 | 52,17 | 40,78 | 21,76 | 38,79 | 56,64 | 27,40 | 63,40 | 45,76 |
| - DeltaNet | 32,02 | 33,89 | 32,04 | 30,43 | 32,10 | 65,45 | 50,91 | 40,82 | 21,42 | 39,15 | 60,89 | 28,00 | 63,97 | 46,33 |
| - Gated DeltaNet | 31,65 | 34,53 | 33,98 | 29,09 | 32,31 | 64,53 | 51,07 | 41,62 | 21,59 | 39,05 | 60,03 | 28,40 | 63,21 | 46,19 |
| - Mesa | 31,65 | 34,49 | 32,87 | 30,43 | 32,36 | 66,43 | 51,85 | 40,03 | 22,27 | 38,43 | 56,73 | 27,40 | 63,34 | 45,81 |
| - Hawk-Mesa | 32,14 | 34,99 | 32,87 | 31,96 | 32,99 | 65,40 | 52,96 | 41,16 | 23,55 | 39,05 | 55,26 | 28,00 | 62,89 | 46,03 |
| - Transformer | 33,84 | 33,91 | 35,91 | 30,62 | 33,57 | 65,34 | 52,49 | 39,27 | 22,44 | 39,10 | 59,63 | 28,40 | 63,78 | 46,31 |
| **145M Models / 50B T.** | | | | | | | | | | | | | | |
| - Hawk | 22,14 | 35,09 | 28,18 | 30,33 | 28,94 | 65,94 | 51,62 | 41,33 | 22,87 | 39,46 | 59,45 | 28,20 | 63,97 | 46,60 |
| - Mamba2 | 29,23 | 34,24 | 33,15 | 29,86 | 31,62 | 65,78 | 51,46 | 41,08 | 21,67 | 39,82 | 59,30 | 28,00 | 61,74 | 46,11 |
| - GLA | 32,16 | 35,57 | 32,04 | 29,86 | 32,41 | 65,56 | 51,07 | 43,18 | 23,81 | 39,82 | 52,23 | 29,40 | 63,72 | 46,10 |
| - xLSTM | 32,74 | 35,89 | 32,87 | 30,14 | 32,91 | 66,59 | 51,54 | 41,67 | 23,12 | 39,15 | 58,65 | 27,00 | 64,23 | 46,49 |
| - DeltaNet | 32,89 | 35,39 | 32,32 | 31,67 | 33,07 | 66,10 | 51,93 | 40,53 | 22,78 | 38,74 | 57,46 | 29,00 | 64,29 | 46,36 |
| - Gated DeltaNet | 32,85 | 36,15 | 33,15 | 31,96 | 33,53 | 66,76 | 51,22 | 41,92 | 23,55 | 38,38 | 60,43 | 29,00 | 64,10 | 46,92 |
| - Mesa | 32,33 | 36,24 | 34,53 | 30,24 | 33,33 | 65,40 | 51,70 | 41,62 | 22,61 | 38,89 | 54,65 | 28,80 | 63,53 | 45,90 |
| - Hawk-Mesa | 34,31 | 36,40 | 32,04 | 31,20 | 33,49 | 66,21 | 51,93 | 41,54 | 22,53 | 38,54 | 55,57 | 30,00 | 64,74 | 46,38 |
| - Transformer | 35,40 | 36,03 | 35,08 | 31,10 | 34,40 | 64,58 | 52,09 | 41,41 | 22,01 | 40,12 | 59,79 | 30,20 | 64,23 | 46,80 |
| **400M Models / 15B T.** | | | | | | | | | | | | | | |
| - Hawk | 32,97 | 42,33 | 33,15 | 32,06 | 35,13 | 68,66 | 50,99 | 44,53 | 25,00 | 39,66 | 59,69 | 30,80 | 67,09 | 48,30 |
| - Mamba2 | 35,92 | 39,95 | 33,70 | 32,25 | 35,46 | 68,44 | 51,70 | 43,31 | 23,46 | 39,71 | 59,54 | 30,40 | 66,45 | 47,88 |
| - GLA | 40,09 | 42,49 | 34,53 | 32,54 | 37,41 | 68,61 | 51,78 | 44,99 | 24,91 | 39,61 | 60,40 | 28,40 | 68,30 | 48,37 |
| - xLSTM | 39,67 | 41,99 | 35,08 | 33,11 | 37,46 | 68,50 | 52,25 | 45,12 | 23,46 | 39,87 | 59,72 | 31,60 | 68,17 | 48,59 |
| - DeltaNet | 39,28 | 41,49 | 36,46 | 32,34 | 37,39 | 69,26 | 51,70 | 46,00 | 23,81 | 39,76 | 52,51 | 31,20 | 67,47 | 47,71 |
| - Gated DeltaNet | 39,98 | 42,55 | 32,87 | 33,68 | 37,27 | 69,59 | 52,33 | 45,20 | 25,17 | 40,02 | 59,14 | 29,40 | 67,60 | 48,56 |
| - Mesa | 40,17 | 42,71 | 34,53 | 33,21 | 37,65 | 67,79 | 50,51 | 45,12 | 22,87 | 39,10 | 52,42 | 29,80 | 68,43 | 47,00 |
| - Hawk-Mesa | 39,84 | 43,15 | 34,81 | 31,67 | 37,37 | 69,64 | 52,17 | 45,33 | 22,27 | 40,23 | 58,04 | 29,80 | 67,41 | 48,11 |
| - SWA-4 | 4,62 | 34,97 | 25,97 | 25,93 | 22,87 | 66,81 | 49,33 | 43,81 | 24,23 | 39,82 | 57,31 | 30,00 | 63,78 | 46,89 |
| - SWA-64 | 38,54 | 39,35 | 32,87 | 30,24 | 35,25 | 68,93 | 52,17 | 44,40 | 22,87 | 39,76 | 58,56 | 29,20 | 64,99 | 47,61 |
| - SWA-1024 | 41,43 | 40,90 | 37,57 | 34,26 | 38,54 | 67,90 | 52,80 | 44,49 | 22,61 | 40,58 | 60,37 | 30,20 | 66,58 | 48,19 |
| - Transformer | 41,12 | 41,27 | 37,29 | 34,45 | 38,53 | 68,23 | 51,07 | 44,28 | 24,57 | 40,23 | 58,10 | 28,40 | 66,58 | 47,68 |
| **400M Models / 50B T.** | | | | | | | | | | | | | | |
| - Hawk | 36,70 | 47,02 | 33,43 | 32,54 | 37,42 | 71,93 | 52,25 | 47,26 | 24,06 | 40,89 | 59,91 | 34,20 | 69,83 | 50,04 |
| - Mamba2 | 38,23 | 44,22 | 35,64 | 32,25 | 37,58 | 68,72 | 52,17 | 45,33 | 23,98 | 40,74 | 54,31 | 31,80 | 68,49 | 48,19 |
| - GLA | 41,98 | 46,00 | 35,08 | 34,74 | 39,45 | 69,86 | 54,14 | 46,46 | 23,98 | 40,07 | 56,57 | 29,80 | 69,96 | 48,86 |
| - xLSTM | 41,82 | 46,22 | 34,53 | 33,30 | 38,97 | 68,99 | 53,35 | 46,00 | 23,46 | 41,61 | 57,43 | 31,00 | 69,32 | 48,90 |
| - DeltaNet | 42,25 | 45,92 | 37,02 | 33,68 | 39,72 | 70,18 | 52,72 | 45,24 | 24,23 | 40,48 | 57,37 | 32,20 | 68,87 | 48,91 |
| - Gated DeltaNet | 43,99 | 46,57 | 35,36 | 34,83 | 40,19 | 70,18 | 51,85 | 46,38 | 25,77 | 40,58 | 54,89 | 32,60 | 70,53 | 49,10 |
| - Mesa | 43,39 | 46,93 | 38,95 | 34,26 | 40,88 | 70,73 | 54,46 | 46,21 | 24,91 | 41,10 | 57,89 | 32,40 | 69,38 | 49,64 |
| - Hawk-Mesa | 41,94 | 46,96 | 38,12 | 33,49 | 40,13 | 70,46 | 54,78 | 46,46 | 25,51 | 40,74 | 57,80 | 30,00 | 70,46 | 49,53 |
| - SWA-4 | 18,28 | 39,02 | 29,56 | 27,66 | 28,63 | 67,85 | 51,93 | 44,49 | 24,83 | 39,71 | 58,23 | 32,40 | 66,14 | 48,20 |
| - SWA-64 | 42,34 | 44,14 | 34,53 | 31,67 | 38,17 | 69,53 | 53,75 | 45,24 | 24,74 | 40,28 | 56,45 | 31,60 | 68,49 | 48,76 |
| - SWA-1024 | 45,08 | 46,43 | 38,95 | 34,74 | 41,30 | 69,64 | 52,25 | 45,71 | 25,00 | 40,07 | 57,92 | 32,20 | 67,92 | 48,84 |
| - Transformer | 44,96 | 46,30 | 41,44 | 35,89 | 42,15 | 69,91 | 52,64 | 45,96 | 24,06 | 40,48 | 57,31 | 30,40 | 69,64 | 48,80 |
| **1B Models / 15B T.** | | | | | | | | | | | | | | |
| - Hawk | 37,98 | 47,71 | 35,08 | 32,25 | 38,25 | 71,93 | 50,43 | 48,61 | 25,43 | 41,50 | 58,53 | 31,80 | 70,59 | 49,85 |
| - Mamba2 | 39,63 | 45,06 | 36,74 | 34,35 | 38,95 | 70,13 | 52,33 | 46,97 | 25,43 | 39,41 | 57,34 | 31,80 | 70,34 | 49,22 |
| - GLA | 43,24 | 47,20 | 33,43 | 33,68 | 39,39 | 70,95 | 52,41 | 46,97 | 25,00 | 41,15 | 58,59 | 33,00 | 70,34 | 49,80 |
| - xLSTM | 44,05 | 46,10 | 35,91 | 33,40 | 39,86 | 70,73 | 54,30 | 47,14 | 25,00 | 40,63 | 59,27 | 32,40 | 69,64 | 49,89 |
| - DeltaNet | 43,45 | 47,47 | 36,46 | 33,30 | 40,17 | 70,78 | 52,80 | 48,48 | 25,09 | 39,92 | 60,46 | 31,20 | 69,00 | 49,72 |
| - Gated DeltaNet | 45,37 | 48,49 | 35,36 | 34,07 | 40,82 | 71,60 | 53,99 | 48,57 | 24,83 | 40,07 | 53,76 | 32,40 | 70,46 | 49,46 |
| - Mesa | 44,21 | 47,70 | 37,02 | 33,49 | 40,60 | 70,89 | 54,46 | 47,56 | 25,26 | 41,04 | 56,06 | 32,20 | 70,21 | 49,71 |
| - Hawk-Mesa | 44,05 | 48,70 | 39,23 | 33,40 | 41,34 | 71,22 | 53,20 | 49,54 | 24,74 | 40,89 | 51,93 | 32,00 | 70,78 | 49,29 |
| - SWA-4 | 8,46 | 38,56 | 27,62 | 27,18 | 25,46 | 67,95 | 51,30 | 46,72 | 23,72 | 40,17 | 56,73 | 30,40 | 65,50 | 47,81 |
| - SWA-64 | 42,60 | 44,04 | 31,49 | 30,72 | 37,21 | 69,91 | 51,30 | 46,72 | 24,66 | 41,10 | 58,56 | 33,20 | 67,98 | 49,18 |
| - SWA-1024 | 45,06 | 46,23 | 39,50 | 34,74 | 41,38 | 70,29 | 53,99 | 47,39 | 24,15 | 40,94 | 59,54 | 30,60 | 69,00 | 49,49 |
| - Transformer | 45,31 | 46,65 | 41,16 | 35,79 | 42,23 | 70,78 | 52,25 | 48,19 | 23,55 | 40,28 | 52,91 | 31,40 | 67,98 | 48,42 |
| **1B Models / 50B T.** | | | | | | | | | | | | | | |
| - Hawk | 41,80 | 54,25 | 34,25 | 34,35 | 41,17 | 72,91 | 52,33 | 51,52 | 28,75 | 40,84 | 56,51 | 35,00 | 74,67 | 51,57 |
| - Mamba2 | 42,13 | 51,46 | 37,85 | 35,02 | 41,62 | 71,76 | 53,35 | 48,95 | 26,54 | 40,58 | 55,90 | 33,60 | 73,39 | 50,51 |
| - GLA | 47,27 | 53,05 | 41,44 | 35,60 | 44,34 | 72,25 | 54,14 | 50,46 | 27,56 | 41,25 | 56,85 | 35,00 | 74,03 | 51,44 |
| - xLSTM | 46,57 | 53,08 | 37,57 | 34,74 | 42,99 | 72,52 | 54,62 | 49,45 | 27,05 | 41,76 | 58,78 | 35,80 | 72,06 | 51,50 |
| - DeltaNet | 47,08 | 53,21 | 40,33 | 34,83 | 43,86 | 72,20 | 54,30 | 48,19 | 27,90 | 40,84 | 60,49 | 34,40 | 74,28 | 51,58 |
| - Gated DeltaNet | 49,19 | 54,10 | 39,78 | 36,27 | 44,84 | 71,93 | 54,06 | 51,22 | 26,88 | 41,35 | 53,27 | 34,20 | 73,14 | 50,76 |
| - Mesa | 48,83 | 53,58 | 40,88 | 36,84 | 45,03 | 71,71 | 53,59 | 49,37 | 25,68 | 40,58 | 53,30 | 35,60 | 74,09 | 50,49 |
| - Hawk-Mesa | 47,02 | 54,47 | 40,61 | 36,36 | 44,62 | 72,52 | 56,04 | 50,80 | 26,88 | 40,17 | 56,02 | 35,60 | 74,03 | 51,51 |
| - SWA-4 | 24,63 | 44,90 | 28,18 | 27,08 | 31,20 | 70,35 | 52,49 | 48,19 | 24,83 | 39,56 | 60,15 | 32,80 | 68,56 | 49,62 |
| - SWA-64 | 46,11 | 51,30 | 38,40 | 33,49 | 42,33 | 71,87 | 53,35 | 49,62 | 26,71 | 40,74 | 56,70 | 33,40 | 71,74 | 50,52 |
| - SWA-1024 | 50,38 | 53,69 | 41,44 | 37,22 | 45,68 | 72,47 | 53,35 | 49,41 | 27,13 | 41,61 | 62,20 | 32,60 | 72,06 | 51,35 |
| - Transformer | 48,92 | 53,63 | 42,27 | 37,32 | 45,54 | 72,31 | 54,62 | 49,41 | 28,24 | 40,17 | 60,73 | 35,20 | 72,25 | 51,62 |

**Table 14: Benchmark Scores on Common Reasoning Benchmarks on all model configurations.** Best scores among the recurrent models are highlighted for each training setting.

### L.2.2 IN-CONTEXT RECALL BENCHMARKS

To evaluate in-context recall, we adopted the minimal-transformed version of the benchmarks from Arora et al. (2024) to allow evaluation of non-instruction-tuned models. We truncate inputs to 2000 tokens, and sample greedily until either 48 tokens or a new-line delimiter is generated. We then parsed whether the target was contained in the generation (non-case-sensitive), as in Arora et al. (2024) .

**Sliding-Window Attention Controls.** As expected, we observe consistent score increases with a growing attention window size (see Table 15). However, we observe that the `SWA-1024` is consistently better on SQUAD than the transformer baseline with an attention window of 2048. Closer inspection of the SQUAD benchmarks reveals that the tokens-to-recall are most frequently located in the last 1k tokens of the sequence. Similarly for FDA, most tokens-to-recall are located at the very beginning of the sequence with an average of length 2000. Hence, we observe a significant performance increase from `SWA-1024` to the transformer baseline with an attention window of 2048.

**Results on all Model Settings.** MesaNet consistently attains best, or in few cases second-best, performance scores on average across all evaluated model settings (see Table 16). Moreover, we observe that our insights from the PPL analysis in L.1 directly translate to the observed results in here, e.g., Hawk attaining the worst in-context recall performance.

| | | 15B Tokens | | | | | | | 50B Tokens | | | | | | |
|---|---|---|---|---|---|---|---|---|---|---|---|---|---|---|---|
| | | SWDE acc ↑ | SQUAD acc ↑ | FDA acc ↑ | TQA acc ↑ | NQ acc ↑ | DROP acc ↑ | AVG acc ↑ | SWDE acc ↑ | SQUAD acc ↑ | FDA acc ↑ | TQA acc ↑ | NQ acc ↑ | DROP acc ↑ | AVG acc ↑ |
| **400M Models:** | - SWA-4 | 7,38 | 5,60 | 0,18 | 14,51 | 3,52 | 9,15 | 6,72 | 10,98 | 7,77 | 0,45 | 21,27 | 5,16 | 13,13 | 9,79 |
| | - SWA-16 | 9,63 | 10,82 | 0,27 | 24,88 | 4,88 | 15,33 | 10,97 | 13,05 | 18,30 | 1,09 | 33,35 | 6,59 | 17,35 | 14,95 |
| | - SWA-64 | 13,14 | 26,74 | 10,07 | 39,34 | 5,23 | 19,12 | 18,94 | 19,17 | 38,44 | 11,43 | 48,76 | 7,25 | 23,96 | 24,84 |
| | - SWA-256 | 21,69 | 40,92 | 12,25 | 50,95 | 6,87 | 23,67 | 26,06 | 30,96 | 42,19 | 14,70 | 56,16 | 10,10 | 24,20 | 29,72 |
| | - SWA-1024 | 54,91 | 43,06 | 17,79 | 52,67 | 10,86 | 26,45 | 34,29 | 60,04 | 46,82 | 22,60 | 58,06 | 13,84 | 27,89 | 38,21 |
| | - Transformer | 77,50 | 37,13 | 79,13 | 53,08 | 16,57 | 26,59 | 48,33 | 79,66 | 36,93 | 75,86 | 58,95 | 18,94 | 29,37 | 49,95 |
| **1B Models:** | - SWA-4 | 9,00 | 6,53 | 0,27 | 17,06 | 4,40 | 11,60 | 8,14 | 13,05 | 10,66 | 0,27 | 26,54 | 7,10 | 13,61 | 11,87 |
| | - SWA-16 | 9,54 | 15,25 | 0,27 | 29,15 | 6,46 | 16,44 | 12,85 | 16,74 | 23,76 | 2,09 | 39,28 | 8,46 | 18,59 | 18,15 |
| | - SWA-64 | 16,74 | 30,56 | 16,61 | 44,55 | 7,19 | 20,46 | 22,69 | 22,32 | 39,85 | 12,70 | 51,90 | 9,63 | 23,91 | 26,72 |
| | - SWA-256 | 25,74 | 45,34 | 17,79 | 56,10 | 8,81 | 26,45 | 30,04 | 35,82 | 46,45 | 17,33 | 59,77 | 12,54 | 27,46 | 33,23 |
| | - SWA-1024 | 60,76 | 40,65 | 24,23 | 56,99 | 11,88 | 27,65 | 37,03 | 63,73 | 47,65 | 26,68 | 61,43 | 15,52 | 30,04 | 40,84 |
| | - Transformer | 79,21 | 42,76 | 77,04 | 56,99 | 18,69 | 29,47 | 50,69 | 83,35 | 46,92 | 70,96 | 63,21 | 21,79 | 27,41 | 52,27 |
| | - Random | ≈ 0 | ≈ 0 | ≈ 0 | ≈ 0 | ≈ 0 | ≈ 0 | ≈ 0 | ≈ 0 | ≈ 0 | ≈ 0 | ≈ 0 | ≈ 0 | ≈ 0 | ≈ 0 |

**Table 15: Reference Scores of `SWA` Models on In-Context Recall Benchmarks.** The pattern of best scores (highlighted red) is very consistent across the evaluated settings. As expected, we see increasing performance with increasing sizes of attention windows. Except on SQUAD, the transformer commonly attains the best scores.

| | | 15B Tokens | | | | | | | 50B Tokens | | | | | | |
|---|---|---|---|---|---|---|---|---|---|---|---|---|---|---|---|
| | | SWDE | SQUAD | FDA | TQA | NQ | DROP | AVG | SWDE | SQUAD | FDA | TQA | NQ | DROP | AVG |
| | | acc ↑ | acc ↑ | acc ↑ | acc ↑ | acc ↑ | acc ↑ | acc ↑ | acc ↑ | acc ↑ | acc ↑ | acc ↑ | acc ↑ | acc ↑ | |
| **145M Models:** | - Hawk | 11,43 | 11,09 | 0,27 | 30,39 | 4,09 | 14,18 | 11,91 | 10,08 | 14,08 | 0,36 | 35,25 | 5,38 | 14,85 | 13,33 |
| | - Mamba2 | 29,52 | 24,83 | 14,34 | 40,17 | 7,57 | 20,89 | 22,89 | 37,62 | 26,34 | 14,70 | 44,43 | 7,67 | 20,27 | 25,17 |
| | - GLA | 37,08 | 38,20 | 14,07 | 44,73 | 8,58 | 23,38 | 27,67 | 39,69 | 30,46 | 15,88 | 48,16 | 10,80 | 23,86 | 28,14 |
| | - xLSTM | 33,39 | 25,00 | 11,34 | 44,79 | 10,45 | 25,44 | 25,07 | 34,65 | 36,03 | 19,96 | 48,76 | 11,43 | 23,53 | 29,06 |
| | - DeltaNet | 33,57 | 29,69 | 15,61 | 46,27 | 9,66 | 23,48 | 26,38 | 39,24 | 31,60 | 18,06 | 46,39 | 11,40 | 20,27 | 27,83 |
| | - Gated DeltaNet | 32,31 | 30,83 | 16,42 | 46,68 | 10,48 | 23,43 | 26,69 | 38,07 | 32,44 | 15,79 | 48,34 | 10,74 | 21,23 | 27,77 |
| | - Mesa | 36,90 | 34,35 | 14,88 | 47,22 | 10,20 | 25,68 | 28,21 | 40,50 | 29,99 | 15,79 | 47,04 | 11,97 | 23,77 | 28,18 |
| | - Hawk-Mesa | 34,65 | 30,33 | 13,61 | 46,33 | 9,79 | 22,86 | 26,26 | 34,38 | 36,03 | 9,89 | 46,86 | 11,31 | 21,80 | 26,71 |
| | - Transformer | 63,73 | 23,89 | 54,63 | 46,50 | 12,01 | 25,59 | 37,72 | 67,78 | 30,97 | 70,87 | 50,30 | 14,70 | 23,62 | 43,04 |
| **400M Models:** | - Hawk | 16,47 | 23,86 | 1,09 | 42,42 | 8,01 | 19,65 | 18,58 | 22,05 | 23,86 | 1,48 | 48,93 | 10,83 | 20,60 | 21,29 |
| | - Mamba2 | 43,11 | 29,86 | 20,42 | 47,04 | 11,47 | 22,81 | 29,12 | 51,04 | 29,76 | 22,23 | 52,90 | 12,58 | 24,77 | 32,21 |
| | - GLA | 52,30 | 39,04 | 20,96 | 50,12 | 14,16 | 28,41 | 34,17 | 54,10 | 41,59 | 26,23 | 55,04 | 16,00 | 26,07 | 36,50 |
| | - xLSTM | 51,67 | 38,94 | 23,32 | 51,13 | 14,76 | 23,48 | 33,88 | 50,86 | 38,87 | 25,23 | 53,67 | 16,09 | 24,63 | 34,89 |
| | - DeltaNet | 50,23 | 35,62 | 27,40 | 50,00 | 14,38 | 25,16 | 33,80 | 55,90 | 35,59 | 27,40 | 53,50 | 15,11 | 23,67 | 35,19 |
| | - Gated DeltaNet | 53,20 | 35,15 | 27,04 | 51,72 | 15,96 | 24,82 | 34,65 | 56,53 | 37,23 | 29,49 | 53,55 | 15,01 | 23,96 | 35,96 |
| | - Mesa | 53,11 | 38,54 | 28,58 | 52,13 | 14,29 | 27,02 | 35,61 | 59,05 | 47,05 | 28,95 | 57,17 | 17,29 | 26,31 | 39,30 |
| | - Hawk-Mesa | 52,66 | 39,95 | 23,05 | 52,78 | 13,62 | 26,26 | 34,72 | 53,65 | 39,95 | 25,14 | 55,51 | 15,62 | 27,55 | 36,23 |
| | - Transformer | 77,50 | 37,13 | 79,13 | 53,08 | 16,57 | 26,59 | 48,33 | 79,66 | 36,93 | 75,86 | 58,95 | 18,94 | 29,37 | 49,95 |
| **1B Models:** | - Hawk | 20,25 | 15,72 | 2,09 | 48,34 | 10,42 | 21,61 | 19,74 | 26,73 | 29,96 | 3,27 | 52,96 | 14,63 | 22,66 | 25,04 |
| | - Mamba2 | 54,10 | 33,68 | 26,41 | 51,66 | 13,97 | 25,11 | 34,15 | 59,68 | 37,84 | 31,13 | 56,64 | 15,39 | 25,35 | 37,67 |
| | - GLA | 59,68 | 41,29 | 29,67 | 55,04 | 16,25 | 25,97 | 37,98 | 60,58 | 43,67 | 30,40 | 59,24 | 18,69 | 25,25 | 39,64 |
| | - xLSTM | 57,61 | 39,11 | 24,50 | 54,50 | 15,17 | 26,64 | 36,26 | 63,37 | 38,91 | 31,58 | 58,00 | 18,06 | 25,59 | 39,25 |
| | - DeltaNet | 58,15 | 37,60 | 36,84 | 55,15 | 16,63 | 25,35 | 38,29 | 62,56 | 39,01 | 38,29 | 59,54 | 17,96 | 25,40 | 40,46 |
| | - Gated DeltaNet | 59,59 | 39,48 | 37,30 | 55,86 | 17,39 | 25,87 | 39,25 | 60,22 | 39,81 | 32,12 | 59,54 | 18,56 | 26,98 | 39,54 |
| | - Mesa | 60,40 | 49,06 | 22,50 | 54,38 | 17,55 | 27,46 | 38,56 | 63,10 | 46,25 | 32,67 | 61,37 | 19,64 | 27,74 | 41,79 |
| | - Hawk-Mesa | 61,03 | 41,55 | 27,77 | 54,74 | 15,33 | 25,68 | 37,68 | 60,31 | 45,51 | 28,68 | 60,13 | 17,61 | 27,70 | 39,99 |
| | - Transformer | 79,21 | 42,76 | 77,04 | 56,99 | 18,69 | 29,47 | 50,69 | 83,35 | 46,92 | 70,96 | 63,21 | 21,79 | 27,41 | 52,27 |

**Table 16: Benchmark Scores for In-Context Recall Benchmarks on all Model Settings.** MesaNet consistently attains the best or second-best score on average across all evaluated model settings.

### L.2.3 FEW-SHOT LEARNING BENCHMARKS

To evaluate the few-shot learning ability, we tested two distinct types of few-shot tasks, (i) word scrambling tasks introduced in (Brown et al., 2020b) and (ii) a couple of language-to-language translation tasks.

**Word Scrambling Tasks.** We report the few-shot performances in Table 17 for 0-,1-,10- and 100-shot settings. As few-shot evaluation tend to be sensitive to the selection and ordering of few-shot examples (Lu et al., 2021), we report the mean performance over 10 randomly drawn few-shot prefixes. We observe consistent improvements with an increasing number of fewshots for all models except for `SWA-4`. MesaNet attains the strongest performance scores in most settings, and outperforms the transformer baseline significantly.

While we evaluate on all five word scrambling tasks introduce in Brown et al. (2020b), we observe only observe signal (performance above 1%) for models in the ranges 145M to 1B on two tasks: `gpt3/cycle_letters_in_word` and `gpt3/mid_word_2_anagrams`. On the three remaining tasks, we observe performance score close to 0%, in line with the results of Brown et al. (2020b), and hence omit the scores here.

| | | gpt3/cycle_letters_in_word | | | | gpt3/mid_word_2_anagrams | | | |
| --- | --- | --- | --- | --- | --- | --- | --- | --- | --- |
| | | 0-shot | 1-shot | 10-shot | 100-shot | 0-shot | 1-shot | 10-shot | 100-shot |
| **145M Models** | - Hawk | 0.2 | 0.4±0.2 | 1.3±0.5 | 1.7±0.5 | 0.2 | 0.4±0.1 | 0.8±0.2 | 0.7±0.2 |
| | - Mamba2 | 0.0 | 0.2±0.2 | 1.7±0.4 | 1.4±0.3 | 0.0 | 0.2±0.3 | 0.6±0.2 | 0.3±0.1 |
| | - GLA | 0.1 | 0.2±0.3 | 2.4±0.7 | 3.0±0.4 | 0.2 | 0.1±0.1 | 1.0±0.4 | 1.5±0.1 |
| | - xLSTM | 0.1 | 0.4±0.5 | 2.8±0.6 | 3.8±0.5 | 0.3 | 0.1±0.2 | 0.9±0.3 | 1.6±0.1 |
| | - DeltaNet | 0.1 | 0.5±0.4 | 2.6±0.9 | 3.2±0.6 | 0.1 | 0.2±0.1 | 1.2±0.3 | 1.1±0.2 |
| | - Gated DeltaNet | 0.1 | 0.8±0.6 | 2.5±0.7 | 3.4±0.6 | 0.0 | 0.4±0.4 | 1.4±0.2 | 1.7±0.2 |
| | - Mesa | 0.1 | 0.2±0.3 | 2.2±0.5 | 3.3±0.5 | 0.1 | 0.2±0.2 | 1.1±0.3 | 1.7±0.1 |
| | - Hawk-Mesa | 0.0 | 0.3±0.2 | 1.7±0.5 | 2.4±0.6 | 0.2 | 0.2±0.3 | 0.9±0.3 | 1.4±0.2 |
| | - Transformer | 0.1 | 0.5±0.4 | 2.6±0.5 | 3.7±0.3 | 0.1 | 0.2±0.2 | 1.2±0.3 | 1.7±0.2 |
| **400M Models** | - Hawk | 0.1 | 1.7±1.2 | 5.3±1.2 | 6.6±0.4 | 0.1 | 0.9±0.7 | 2.4±0.1 | 2.8±0.2 |
| | - Mamba2 | 0.4 | 2.0±1.4 | 4.5±0.6 | 5.1±0.5 | 0.4 | 0.9±0.5 | 1.6±0.3 | 1.6±0.1 |
| | - GLA | 0.0 | 1.7±1.1 | 5.2±1.0 | 7.6±0.3 | 0.4 | 1.0±0.7 | 2.4±0.2 | 2.6±0.2 |
| | - xLSTM | 0.0 | 2.3±1.3 | 5.7±1.3 | 8.2±0.5 | 0.2 | 1.1±0.5 | 2.5±0.3 | 2.9±0.3 |
| | - DeltaNet | 0.1 | 1.5±1.0 | 5.7±1.3 | 7.6±0.6 | 0.0 | 1.1±0.5 | 2.4±0.3 | 2.6±0.3 |
| | - Gated DeltaNet | 0.1 | 2.1±1.7 | 6.5±1.0 | 9.0±0.8 | 0.1 | 0.9±0.5 | 2.6±0.3 | 3.4±0.2 |
| | - Mesa | 0.4 | 2.2±1.2 | 6.6±1.0 | 9.2±0.6 | 0.6 | 1.1±0.5 | 2.6±0.3 | 3.2±0.2 |
| | - Hawk-Mesa | 0.0 | 1.3±0.9 | 4.0±1.4 | 7.3±0.5 | 0.1 | 0.9±0.7 | 2.6±0.3 | 3.1±0.1 |
| | - SWA-4 | 0.0 | 0.4±0.3 | 0.8±0.3 | 0.8±0.2 | 0.0 | 0.3±0.3 | 0.9±0.3 | 0.9±0.3 |
| | - SWA-64 | 0.1 | 2.5±1.5 | 4.6±1.1 | 4.7±0.9 | 0.1 | 1.2±0.5 | 2.7±0.2 | 2.7±0.1 |
| | - SWA-1024 | 0.3 | 2.5±1.6 | 6.1±0.9 | 7.7±0.5 | 0.8 | 1.2±0.8 | 2.9±0.4 | 3.1±0.3 |
| | - Transformer | 0.4 | 2.4±1.8 | 6.7±1.2 | 8.5±0.4 | 0.5 | 1.4±0.7 | 3.3±0.4 | 3.6±0.2 |
| **1B Models** | - Hawk | 0.2 | 1.5±1.0 | 6.8±1.5 | 9.2±0.6 | 0.1 | 0.9±0.8 | 3.5±0.4 | 3.8±0.2 |
| | - Mamba2 | 0.8 | 3.7±1.7 | 6.3±0.8 | 6.4±0.7 | 1.1 | 1.8±0.3 | 2.4±0.3 | 2.0±0.4 |
| | - GLA | 0.3 | 4.1±2.1 | 8.4±1.2 | 10.3±0.5 | 0.5 | 2.3±0.7 | 3.9±0.5 | 4.2±0.2 |
| | - xLSTM | 0.0 | 2.2±1.3 | 7.7±1.8 | 11.0±0.4 | 0.3 | 1.8±0.5 | 3.9±0.3 | 4.6±0.3 |
| | - DeltaNet | 0.0 | 2.9±1.9 | 8.7±1.3 | 11.7±0.8 | 0.1 | 1.6±0.8 | 3.7±0.5 | 4.1±0.3 |
| | - Gated DeltaNet | 0.3 | 4.0±1.8 | 8.9±1.4 | 11.8±0.7 | 0.5 | 2.5±0.9 | 4.7±0.6 | 6.1±0.4 |
| | - Mesa | 0.5 | 3.3±2.0 | 9.7±1.3 | 14.0±0.5 | 1.1 | 2.1±1.1 | 4.7±0.6 | 6.2±0.4 |
| | - Hawk-Mesa | 0.4 | 2.1±1.5 | 7.2±1.5 | 11.4±0.5 | 0.6 | 2.0±0.9 | 4.4±0.4 | 5.8±0.3 |
| | - SWA-4 | 0.1 | 1.1±0.9 | 1.5±0.7 | 2.0±0.8 | 0.2 | 0.6±0.5 | 1.4±0.3 | 1.4±0.3 |
| | - SWA-64 | 1.3 | 3.5±1.8 | 6.3±1.3 | 7.8±0.6 | 1.0 | 2.4±0.7 | 3.8±0.3 | 4.0±0.3 |
| | - SWA-1024 | 0.1 | 3.4±1.8 | 7.5±1.3 | 9.0±0.5 | 0.1 | 1.9±0.9 | 4.3±0.4 | 4.3±0.2 |
| | - Transformer | 0.0 | 3.0±2.2 | 6.8±1.7 | 9.2±0.6 | 0.1 | 2.4±0.6 | 4.2±0.4 | 4.7±0.2 |

**Table 17: Few-Shot Performance (Accuracy ± Std.) on GPT-3 Word Scrambling Tasks (Brown et al., 2020b) of Models Trained on 50B Tokens.** Best 50-shot scores are highlighted, and standard deviation is reported over 10 random drawn few-shot selections. MesaNet attains the strongest scores in most settings, and outperforms the transformer baseline significantly.

**Language-to-Language Translation.** We evaluated a model's capability to translate from three different languages to English: (i) French to English (Bojar et al., 2014), (ii) German to English (Bojar et al., 2016) and (iii) Romanian to English (Bojar et al., 2016). We follow the exact prompt setup of Brown et al. (2020b) evaluate with {0,1,5,10} -and 50-shots, and report the performance in Table 18 with respect to BLEU-sb (Post, 2018) for models trained on 50B tokens.

We observe scores of different performance magnitudes across the three languages, which is most likely caused by the multi-lingual distribution of the training data corpus and French being more prevalent than German and Romanian. MesaNet attains superior scores among the recurrent models. However, MesaNet, and more general all recurrent models, fail to match the transformer performance by a relatively big margin, especially at the scale of 1B models. This finding is non-surprising given the impact of the attention mechanism on the field of machine translation (Bahdanau et al., 2014), indicating that pure model- and data-scaling based on recurrent models will not be enough to match the performance of attention-based architecture (Rodchenko et al., 2025).

| | | WMT14 FR-EN | | | | | WMT16 DE-EN | | | | | WMT16 RO-EN | | | | |
|---|---|---|---|---|---|---|---|---|---|---|---|---|---|---|---|---|
| | | 0 | 1 | 5 | 10 | 50 | 0 | 1 | 5 | 10 | 50 | 0 | 1 | 5 | 10 | 50 |
| **145M Models:** | - Hawk | 0,61 | 0,31 | 0,25 | 0,08 | 0,11 | 0,49 | 0,16 | 0,20 | 0,19 | 0,19 | 0,44 | 0,16 | 0,06 | 0,19 | 0,29 |
| | - Mamba2 | 1,68 | 0,56 | 0,73 | 0,73 | 0,19 | 2,13 | 0,40 | 0,28 | 0,51 | 0,37 | 1,68 | 0,32 | 0,44 | 0,50 | 0,46 |
| | - GLA | 1,47 | 0,21 | 0,69 | 0,66 | 0,63 | 1,78 | 0,52 | 0,35 | 0,52 | 0,44 | 1,52 | 0,24 | 0,12 | 0,50 | 0,51 |
| | - xLSTM | 1,64 | 0,07 | 0,73 | 0,87 | 0,67 | 2,09 | 0,63 | 0,33 | 0,81 | 0,77 | 1,68 | 0,22 | 0,34 | 0,50 | 0,85 |
| | - DeltaNet | 1,57 | 0,20 | 0,78 | 0,90 | 0,59 | 1,68 | 0,49 | 0,32 | 0,73 | 0,81 | 1,56 | 0,80 | 0,55 | 0,58 | 0,61 |
| | - Gated DeltaNet | 1,31 | 0,25 | 0,28 | 0,35 | 0,89 | 1,64 | 0,42 | 0,35 | 0,68 | 0,64 | 0,80 | 0,67 | 0,49 | 0,51 | 0,35 |
| | - Mesa | 1,26 | 0,66 | 0,33 | 1,10 | 1,06 | 1,53 | 0,49 | 0,58 | 0,46 | 0,62 | 1,56 | 0,32 | 0,51 | 0,45 | 0,52 |
| | - Hawk-Mesa | 1,62 | 0,19 | 0,80 | 0,77 | 0,94 | 2,03 | 0,51 | 0,67 | 0,47 | 0,79 | 1,77 | 0,24 | 0,54 | 0,49 | 0,90 |
| | - Transformer | 1,55 | 0,05 | 0,59 | 0,70 | 0,87 | 1,90 | 0,39 | 0,86 | 0,61 | 0,71 | 1,75 | 0,28 | 0,30 | 1,41 | 0,50 |
| **400M Models:** | - Hawk | 1,54 | 2,28 | 3,95 | 4,25 | 4,97 | 1,34 | 1,36 | 3,24 | 3,87 | 3,67 | 0,91 | 1,29 | 2,03 | 1,52 | 1,89 |
| | - Mamba2 | 2,15 | 4,05 | 6,07 | 4,55 | 3,49 | 2,17 | 1,43 | 3,13 | 3,19 | 2,52 | 1,68 | 0,86 | 1,36 | 1,94 | 1,64 |
| | - GLA | 1,83 | 3,20 | 2,74 | 4,83 | 4,23 | 2,15 | 2,60 | 1,88 | 2,04 | 2,19 | 1,72 | 0,62 | 1,42 | 1,96 | 1,30 |
| | - xLSTM | 2,14 | 3,08 | 3,48 | 3,66 | 3,28 | 2,29 | 2,06 | 2,68 | 2,79 | 2,77 | 1,63 | 1,10 | 1,37 | 2,20 | 2,15 |
| | - DeltaNet | 1,72 | 3,09 | 4,43 | 3,89 | 3,49 | 1,84 | 1,53 | 3,52 | 2,83 | 2,47 | 1,79 | 1,67 | 1,63 | 1,29 | 1,45 |
| | - Gated DeltaNet | 1,87 | 3,92 | 3,86 | 4,16 | 3,77 | 2,00 | 0,85 | 3,35 | 3,18 | 2,94 | 1,80 | 1,05 | 2,56 | 2,13 | 2,22 |
| | - Mesa | 2,23 | 2,75 | 4,33 | 5,05 | 5,33 | 2,06 | 0,80 | 2,62 | 3,11 | 3,70 | 1,75 | 0,68 | 2,09 | 1,63 | 2,47 |
| | - Hawk-Mesa | 1,90 | 2,83 | 3,89 | 4,54 | 4,27 | 2,00 | 2,55 | 3,66 | 3,26 | 3,20 | 1,74 | 0,68 | 1,71 | 1,71 | 2,28 |
| | - SWA-4 | 0,34 | 0,13 | 0,14 | 0,13 | 0,12 | 0,25 | 0,19 | 0,26 | 0,21 | 0,26 | 0,29 | 0,10 | 0,06 | 0,07 | 0,05 |
| | - SWA-64 | 1,35 | 3,82 | 4,46 | 4,94 | 4,92 | 1,45 | 2,17 | 1,66 | 2,09 | 1,57 | 1,18 | 1,10 | 1,38 | 0,88 | 1,29 |
| | - SWA-1024 | 4,09 | 4,55 | 8,49 | 7,77 | 9,16 | 3,09 | 3,66 | 4,57 | 5,14 | 5,11 | 1,96 | 0,55 | 1,82 | 2,99 | 2,67 |
| | - Transformer | 2,61 | 8,27 | 8,77 | 8,92 | 9,63 | 2,04 | 3,13 | 5,73 | 5,34 | 5,49 | 1,94 | 1,02 | 1,29 | 2,23 | 2,56 |
| **1B Models:** | - Hawk | 3,72 | 5,88 | 8,56 | 7,15 | 4,17 | 3,33 | 3,79 | 3,77 | 5,20 | 5,86 | 2,37 | 2,69 | 4,39 | 4,22 | 4,17 |
| | - Mamba2 | 4,20 | 11,81 | 11,90 | 11,28 | 5,83 | 3,07 | 3,62 | 6,79 | 8,18 | 3,35 | 2,04 | 4,27 | 6,83 | 4,75 | 3,38 |
| | - GLA | 3,15 | 10,60 | 11,87 | 10,90 | 10,31 | 2,58 | 7,90 | 9,41 | 7,77 | 7,46 | 2,15 | 2,59 | 6,60 | 4,30 | 4,95 |
| | - xLSTM | 4,96 | 5,11 | 11,71 | 10,32 | 10,56 | 4,13 | 5,52 | 9,17 | 8,99 | 8,59 | 2,60 | 2,33 | 4,74 | 3,81 | 3,90 |
| | - DeltaNet | 5,24 | 8,34 | 10,79 | 10,08 | 7,88 | 4,02 | 6,91 | 8,72 | 6,01 | 5,66 | 2,29 | 1,01 | 4,32 | 3,39 | 2,58 |
| | - Gated DeltaNet | 4,71 | 8,24 | 10,03 | 11,25 | 11,31 | 4,31 | 7,59 | 9,07 | 8,60 | 8,76 | 2,45 | 4,63 | 5,67 | 5,33 | 5,51 |
| | - Mesa | 3,58 | 11,80 | 12,44 | 11,57 | 11,64 | 3,10 | 6,98 | 10,20 | 8,49 | 7,81 | 1,88 | 5,05 | 2,96 | 6,05 | 5,07 |
| | - Hawk-Mesa | 3,68 | 7,99 | 10,58 | 13,16 | 12,01 | 2,92 | 8,03 | 10,50 | 8,67 | 8,43 | 2,36 | 4,73 | 4,91 | 5,81 | 5,99 |
| | - SWA-4 | 0,54 | 0,72 | 0,72 | 0,72 | 0,74 | 0,49 | 0,75 | 0,89 | 0,87 | 0,72 | 0,22 | 0,12 | 0,14 | 0,11 | 0,06 |
| | - SWA-64 | 5,58 | 6,69 | 2,92 | 8,43 | 7,61 | 4,09 | 5,27 | 4,69 | 4,05 | 3,45 | 2,26 | 1,68 | 1,85 | 3,12 | 3,05 |
| | - SWA-1024 | 8,75 | 16,65 | 18,09 | 18,70 | 19,83 | 5,99 | 10,85 | 14,58 | 14,91 | 14,30 | 3,36 | 4,19 | 10,14 | 10,05 | 8,38 |
| | - Transformer | 8,30 | 18,49 | 17,81 | 17,70 | 19,14 | 6,10 | 13,06 | 11,99 | 13,99 | 13,85 | 3,54 | 5,92 | 7,11 | 7,35 | 7,82 |

Table 18: **Performance Scores (in BLEU-sb) on three Translation Tasks on Models Trained on 50B Tokens.** Best 50-shot scores among recurrent models are highlighted, as well as Transformer reference scores. While MesaNet attains the best-score among the recurrent models in most settings, it under-performs transformer by relative big margin.

### L.3 NEEDLE IN THE HAYSTACK (NIAH) RESULTS

**Setup.** We conducted a sweep of experiments on single-needle tasks (NIAH) from the RULER benchmark (Hsieh et al., 2024) suite for 1B models trained on 50B tokens. We ran experiments for both haystack types (noise and essays) for all key/value combinations (both can be in the form of: words, numbers or uuids) on context lengths 2048 and 4096.

**Results.** As scores are quite sensitive to the chosen key and values types, we report mean±std percent accuracy over all 9 key/value combinations, with 1000 evaluation samples for each setting. On the "noise" haystack, MesaNet demonstrates strong scores with very low fluctuations across key/value combinations. On the "essay" haystack, we observe relatively high score fluctuations across key/value combinations for all models which makes it hard to form conclusions. However, we would still like to highlight the strong performance of Hawk-Mesa on the essay haystack.

|  | **NIAH Noise** | | **NIAH-Essay** | |
|---|---|---|---|---|
|  | L=2048 | L=4096 | L=2048 | L=4096 |
| - Hawk | $4.0 \pm 5.9$ | $1.7 \pm 2.9$ | $3.0 \pm 2.2$ | $2.1 \pm 1.6$ |
| - Mamba2 | $79.7 \pm 17.9$ | $0.7 \pm 1.0$ | $51.3 \pm 22.3$ | $0.0 \pm 0.0$ |
| - GLA | $96.2 \pm 4.2$ | $68.5 \pm 18.9$ | $73.5 \pm 34.7$ | $41.4 \pm 26.9$ |
| - xLSTM | $94.8 \pm 5.0$ | $80.4 \pm 14.9$ | $69.1 \pm 20.5$ | $24.3 \pm 9.9$ |
| - DeltaNet | $99.3 \pm 1.0$ | $96.5 \pm 6.3$ | $68.9 \pm 32.3$ | $27.9 \pm 15.3$ |
| - Gated-DeltaNet | $98.3 \pm 4.1$ | $96.3 \pm 8.1$ | $52.1 \pm 33.7$ | $11.0 \pm 9.4$ |
| - MesaNet | $99.5 \pm 0.5$ | $95.1 \pm 3.9$ | $66.8 \pm 28.9$ | $17.9 \pm 9.0$ |
| - Hawk-Mesa | $97.6 \pm 3.5$ | $65.3 \pm 21.6$ | $90.9 \pm 10.5$ | $55.5 \pm 28.5$ |
| - SWA-1024 | $51.8 \pm 0.9$ | $24.3 \pm 1.3$ | $47.5 \pm 11.8$ | $21.6 \pm 7.2$ |
| - MHA | $99.7 \pm 0.3$ | $0.0 \pm 0.0$ | $98.2 \pm 2.5$ | $0.0 \pm 0.0$ |

**Table 19:** NIAH Benchmark results for 1B models trained on 50B tokens.

## M  VARYING THE NUMBER OF CONJUGATE GRADIENT STEPS WHEN TRAINING MESANETS

Here we present the effect when training the MesaNet on less than 30 steps. We opted for training with 30 steps, as we were not optimizing for training flops but first investigate a fully converge Mesa layer, and because of early experiments on our 400million model which indicated little improvement after 30 steps.

As shown in Figure 13, we see a small, interestingly, uniform increase of training loss across the sequence length when comparing to a model which is trained on 30 steps. Only when dropping the number of CG steps below 10, we see a more drastic jump in loss increase. As we have show in section C, the backward pass also relies on running the CG method to solve linear systems of equations and we leave investigating for future work varying the number of steps in the forward and backward pass.

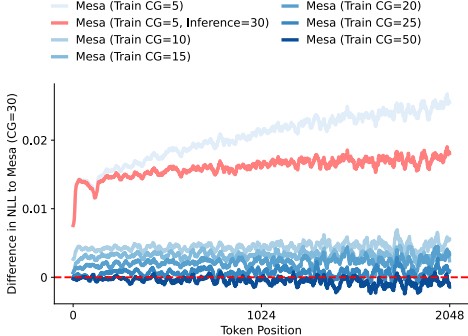

**Figure 13:** We compare the validation loss across the sequence of 400 million parameter MesaNets trained on 15B tokens, when varying the number of conjugate gradient steps during training. We observe a slight uniform increase of validation loss across the sequence length when comparing to a model which is trained on 30 steps. Only when dropping the CG steps drastically to 5 we see a substantial increase in loss.

## N  EVALUATION METHODOLOGY

**Mulitple Choice Tasks:** For a given question $x$, we assess for all possible options $y$ the loss $\text{NLL}(y|x)$ of the option conditional on the question, and then normalize by the number of tokens of $y$. In contrast to related work (Gu & Dao, 2024; Yang et al., 2024a; Beck et al., 2024), we do not heuristically choose between byte-normalized and non-normalized scoring schemes as we have a fixed tokenizer across all models.

**Greedy Matching Tasks.** For a given input $x$ and an expected target sequence $y$ (e.g., one or multiple tokens), we check whether $t$ would be matched under greedy sampling. This is done by obtaining the logits for the concatenated input of $x + y$, and checking whether all tokens belonging to $y$ are matched by taking the argmax over the logits.

**In-Context Recall Tasks.** We follow closely the setup of (Arora et al., 2023b). For a given input $x$, we sample greedily a completion from the model until either $48$ tokens or a new-line character is sampled. We then check whether the target $y$ is contained in the output (non-case-sensitive).

# O  AN INTERNAL ANALYSIS OF THE MESANET

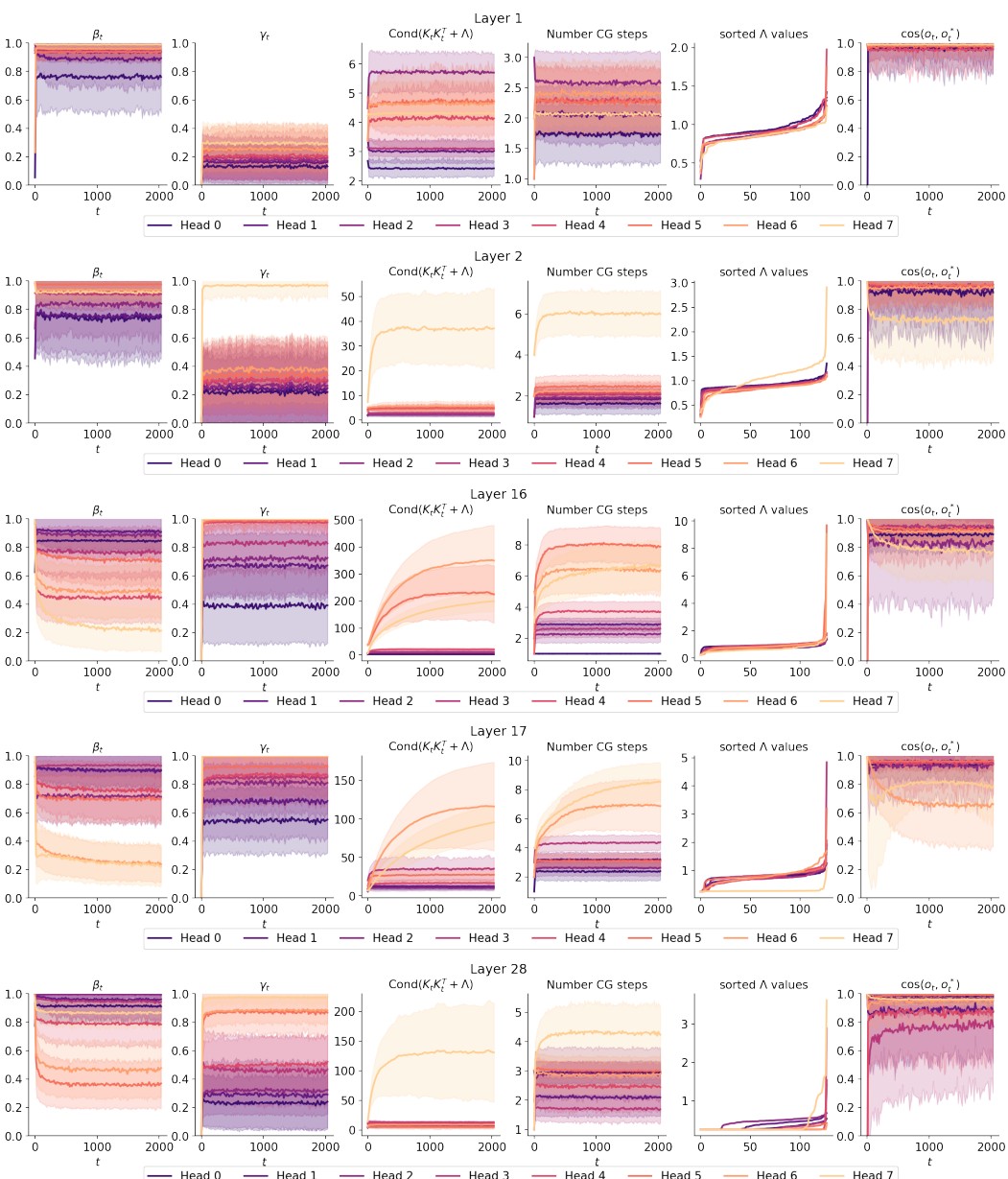

Figure 14: **input strength $\beta$, forget strength $\gamma$, regularization strengths $\Lambda$ as well as other internal statistics of a 400M parameter MesaNet trained on 50B tokens - averaged over 500 sequences from the SlimPajama validation set.** We observe that high $\gamma_t \approx 1$ values usually lead to the condition number of the to be inverted matrix $K_t K_t^T + \Lambda$ increase over time, which in turn leads to more CG steps required to obtain an output for the mesa. We also observe (outer right plot) that usually these heads lead to higher cosine similarity (cos) between $o_t$, the output of the layer if no CG steps are applied which corresponds to gated linear attention, compared to the Mesa output $o_t^*$. We compute the number of conjugate gradient steps are computed by measuring the steps of the conjugate gradient method to reach an error of 0.001. We sort the heads for plotting purposes according to their average gamma values.

