# OpenReview forum: "MesaNet: Sequence Modeling by Locally Optimal Test-Time Training"
_ICLR.cc/2026/Conference — ICLR 2026 Poster_

### Official Review · Reviewer_mKy1 · 2025-10-27

**Soundness:** 3
**Presentation:** 2
**Contribution:** 3
**Rating:** 4
**Confidence:** 4

**Summary:**

This paper introduces Mesanet, which is a linear transformer based on the Mesa optimal regression update rule. The model is inspired by test-time training and the fact that the state update of linear transformers can be seen as an online learning objective. The model is sound, and the chunkwise parallel training makes the model efficient to train; however, the training is more time-consuming than other linear transformers such as GLA and Deltanet due to the solver update. The empirical results are extensive, and the model is evaluated on many different downstream language modeling tasks and length generalization.

**Strengths:**

The paper have several strengths:

- **Model design**: The model design of Mesanet and the connection to test time training is elegant.

- **Experimental setup:** The experimental setup for pre-training in language modeling tasks are sound and followed by the stablished setups such as [1,2].

- **Fairness of comparison:** The paper tunes the learning rate for all different models which isa significant plus point as studies such as Gated Deltanet [3] specifically use the same learning rate for all models, which can indeed be not fair comparison between baselines.

- **Chunk-wise parallel support for training:** In general the training paradigm and chunk wise form is designed nicely and helps parallel training.











------

###  References

[1] Gated Linear Attention Transformers with Hardware-Efficient Training: Songlin Yang, Bailin Wang, Yikang Shen, Rameswar Panda, Yoon Kim

[2] Parallelizing Linear Transformers with the Delta Rule over Sequence Length: Songlin Yang, Bailin Wang, Yu Zhang, Yikang Shen, Yoon Kim

[3] Gated Delta Networks: Improving Mamba2 with Delta Rule: Songlin Yang, Jan Kautz, Ali Hatamizadeh

**Weaknesses:**

The main weakness of the paper lies in its presentation, which significantly reduces clarity regarding the approach and design. These issues can be summarized as follows:

**1)** **Mesanet's online learning objective** is significantly under-explained, starting from Equation 4:

$$
\mathcal{L}(\Phi) = \frac{1}{2} \sum^t_{\tau=1} |v_\tau - \Phi k_\tau|^2 + \frac{\text{Tr} (\Phi^\top \Lambda \Phi)}{2}
$$

By taking the gradient and setting it to zero, we arrive at:

$$
\sum^t_{\tau=1} (v_\tau - \Phi k_\tau)k^\top_\tau + \Lambda \Phi = 0, \quad
\sum^t_{\tau=1} v_\tau k^\top_\tau - \Phi \sum^t_{\tau=1} k_\tau k^\top_\tau + \Lambda \Phi = 0 \quad \text{(Eq. 1)}
$$

The above result corresponds to the optimal $\Phi$. Since in an autoregressive regime the summations above can be expressed recursively, one can write the following recurrences:

$$
G_\tau = G_{\tau-1} + v_\tau k^\top_\tau, \quad
H_\tau = H_{\tau-1} + k_\tau k^\top_\tau
$$

From here, one can substitute the states $G_t$ and $H_t$ into Equation (1) to obtain the optimal Mesa update rule:
$\Phi = G_t(H_t + \Lambda_t)^{-1}$.

Moreover, by including forget gates in the recurrences above, the online learning objective will also change, which is inconsistent with the paper’s notation. I suggest reformulating the online learning loss (Equation 4 in the paper) to explicitly include these forget gates, as also described in *Gated DeltaNet’s Table 1* [1]. Moreover, is $\Lambda_t$ time-dependet or not since in equation 4 of paper is suggested as not but exactly on paragraph bellow it is depending on forget gates.

----

**2) The full recurrence of MesaNet is not shown and not compared with other baselines.**
I recommend moving the recurrence equation of MesaNet (currently on page 21) into the main body of the paper, along with Table 2 for clarity and comparison with other methods.

---

**3) Motivation behind the Mesa layer.**
Since the main motivation is the connection to test-time training (TTT), I wonder why one should use MesaNet instead of DeltaNet or Gated DeltaNet. My main question is: given that DeltaNet and Gated DeltaNet leverage TTT to improve recall abilities, and indeed show significantly better recall, does MesaNet outperform them in this aspect? MesaNet introduces additional complexity in both architecture and training. As Table 14 suggests, if the motivation for this design is to enhance recall ability, this should be clearly stated and justified in the paper.

---

**4) Hawk-Mesa underperforms significantly compared to many baselines and even MesaNet itself in recall tasks (Table 14).**
If the main motivation behind building a model based on the Mesa layer is to introduce an optimal way to structure memory (as suggested by the TTT framework), why does Hawk-Mesa perform substantially worse than MesaNet in Table 14? Why should one construct such a model if it does not improve recall? I ask this because I believe recall ability is one of the primary motivations behind the design of MesaNet and the TTT framework in general.

---

**5) Tables and plots are not clearly visible.**
Many numerical results presented as bar plots are difficult to read. I highly recommend replacing them with tables, particularly Figure 5, to improve clarity.

---

**6) Missing baselines: Atlas and Titans.**
I believe the baselines **Atlas** and **Titans** are missing from the paper’s comparisons. These models address a very similar problem and propose closely related solutions, especially Titans, which should be included for a fair evaluation.


---

### References

[1] Gated Delta Networks: Improving Mamba2 with Delta Rule: Songlin Yang, Jan Kautz, Ali Hatamizadeh

**Questions:**

My most concerns are mentioned above and I mainly have 3 small questions:

**1)** What is the effect of convoulution on q,k,v in Mesanet as it is a cruical component for Deltanet and Gated Deltanet and Mamba2. Also, since it has conections to the TTT paradigm as stated in: https://kexue.fm/archives/11320

**2)** Why does $\Lambda_t = \frac{(1-\gamma_t)}{\beta_t}$ structured as so?

**3)** How are forget gates $\beta_t$ and $\gamma_t$ defined and extracted? Specifically, what nonlinearities and feature mappings are used for them? Are they similar to those in Mamba-2, or do they use a temperatured sigmoid as in GLA or GSA?

Lastly, I am happy to increase my score if above presentation issues and clarifications regarding the paper are resolved.

---

> ### Author Response · Authors · 2025-11-21
> **Rebuttal - Part 1**
>
> Thank you for your thoughtful review, especially your numerous comments on the presentation that helped us to greatly improve the manuscript. We are glad to hear that the reviewer finds the model design elegant and appreciates the experimental setup and fair comparison in a controlled setup.
>
> In the meantime, we have revised our manuscript (highlighted the made changes in red) and hope that we could resolve the raised concerns. Below, we reply point-by-point to your questions and concerns:
>
>
> > The main weakness of the paper lies in its presentation, which significantly reduces clarity regarding the approach and design. [...]
>
> We have rewritten Section 2 following your suggestion. Additionally, we added more details on forgetting, and now present the minimizer of our loss up front (equations 5-6). We also added a step-by-step derivation of the layer forward pass to Appendix D.
>
> >  I suggest reformulating the online learning loss (Equation 4 in the paper) to explicitly include these forget gates, as also described in Gated DeltaNet’s Table 1.
>
> Thank you for raising this point – we completely agree with the reviewer. We updated the manuscript and included the forget gates in the loss.
>
> > The full recurrence of MesaNet is not shown and not compared with other baselines. I recommend moving the recurrence equation of MesaNet (currently on page 21) into the main body of the paper, along with Table 2 for clarity and comparison with other methods.
>
> We would like to highlight that equations 7-10 present the layer recurrence and chunkwise form. Following your suggestion, we moved table 2 into the main text to improve clarity and comparison.
>
> > Motivation behind the Mesa layer. Since the main motivation is the connection to test-time training (TTT), I wonder why one should use MesaNet instead of DeltaNet or Gated DeltaNet. My main question is: given that DeltaNet and Gated DeltaNet leverage TTT to improve recall abilities, and indeed show significantly better recall, does MesaNet outperform them in this aspect? MesaNet introduces additional complexity in both architecture and training. As Table 14 suggests, if the motivation for this design is to enhance recall ability, this should be clearly stated and justified in the paper.
>
> The Mesa layer exploits the fact that cumulative squared error loss functions can be efficiently optimized. As such, it is arguably an interesting next research step, building on DeltaNet and its variants, which use first-order online gradient descent on the current-token squared error loss. We would like to highlight that the 1B-parameter MesaNet outperforms other models not just on in-context recall (Table 14), but in fact also on reasoning benchmarks and negative log-likelihood, the standard sequence model evaluation metric (Figure 3).
>
> > Hawk-Mesa underperforms significantly compared to many baselines and even MesaNet itself in recall tasks (Table 14). If the main motivation behind building a model based on the Mesa layer is to introduce an optimal way to structure memory (as suggested by the TTT framework), why does Hawk-Mesa perform substantially worse than MesaNet in Table 14? Why should one construct such a model if it does not improve recall? I ask this because I believe recall ability is one of the primary motivations behind the design of MesaNet and the TTT framework in general.
>
> Thank you for pointing this out – your comment made us realize that the exploration of HawkMesa was not properly motivated in the initial draft. We did not intend to sell HawkMesa as the preferred model to use – it was simply an analysis exploration to understand the complementary benefits of the different layers. Throughout our careful analysis of the different layers (see figure 3 on NLL difference over loss position), we realized that different layers reduce loss quite differently. Hawk layers are excellent early-in-sequence, but are then vastly outperformed by the other layers as sequence length increases. This observation led us to investigate a combination of Hawk and Mesa, the two best layers in these two regimes, early and late, resp. As expected, HawkMesa exhibits better early-in-the-sequence NLL performance than Mesa, and better later-in-the-sequence NLL performance than Hawk, indicating that hybridization combines the strengths of the two layers. However, HawkMesa fails early in length-extrapolation experiments (see Figure 4), and does not lead to superior in-context recall (see Table 3) – indicating that NLL within the train sequence length does not provide a full picture on the downstream capabilities of the model. We added further context and discussion in the revised manuscript in section 4.1 (highlighted in red).

---

> > ### Author Response · Authors · 2025-11-21
> > **Rebuttal - Part 2**
> >
> > > Tables and plots are not clearly visible. Many numerical results presented as bar plots are difficult to read. I highly recommend replacing them with tables, particularly Figure 5, to improve clarity.
> >
> > Thank you for pointing this out – we agree with the reviewer that the numerical result presentation using bar plots was suboptimal in the initial version. We replaced Figure 5 with an easier-to-access table to improve the presentation and clarity.
> >
> > > Missing baselines: Atlas and Titans. I believe the baselines Atlas and Titans are missing from the paper’s comparisons. These models address a very similar problem and propose closely related solutions, especially Titans, which should be included for a fair evaluation.
> >
> > We would like to highlight that both papers depart from the standard Llama2-type architecture that we adopt in this work, and explore new hybrid model backbones. This deviation makes it hard for us to compare the proposed layers in a fair and controlled manner, a crucial aspect when comparing sequence-mixing layers (as noticed by you and multiple other reviewers). Moreover, the authors of Titans and Atlas do not provide a public implementation of the work, which makes it hard for us to make a sound and fair comparison.
> >
> >
> > > What is the effect of convoulution on q,k,v in Mesanet as it is a cruical component for Deltanet and Gated Deltanet and Mamba2. Also, since it has conections to the TTT paradigm as stated in: https://kexue.fm/archives/11320
> >
> > While we did not ablate it specifically for the MesaNet architecture, the convolution is likely an important architectural element, following e.g. the findings in the studies of Arora et al. (2023) and Fu et al. (2023) for other similar models. We added a comment to Appendix E.1 citing these papers.
> >
> > >  Why does $Lambda_t = \frac{1-\gamma_t}{\beta_t}$ structured as so?
> >
> > This construction of the quadratic regularizer yields the forget and input gates of GLA upon differentiation. This is now detailed in a new Appendix B, where the differentiation is carried out step-by-step.
> >
> > > How are forget gates and defined and extracted? Specifically, what nonlinearities and feature mappings are used for them? Are they similar to those in Mamba-2, or do they use a temperatured sigmoid as in GLA or GSA?
> >
> > No additional tricks were employed apart from what is described in Appendix E.1
> >
> >
> > ---
> >
> > We would like to thank the reviewer again for their helpful comments and suggestions that we happily incorporated in the revised and updated PDF. We hope that we could resolve and address the open concerns and questions of the reviewer. Please let us know if you have further concerns and questions on which we can iterate together throughout the remainder of this rebuttal period – we remain motivated to keep improving our work. In consideration of the overall positive review of the reviewer, and our efforts to clarify the raised concerns, we would be happy if the reviewer would consider increasing their score.

---

> > > ### Comment · Reviewer_mKy1 · 2025-11-23
> > >
> > > Thanks for the improvements specially in terms of the presentation of the paper I believe now the paper is more understandable. As my concerns are addressed I have increased my score. Moreover, I agree with the authors regarding Titans and Atlas that they do not have a public code released.

---

> > > > ### Author Response · Authors · 2025-11-26
> > > >
> > > > Thanks a lot for acknowledging the improved presentation of the paper and raising your score. We are glad to hear that the paper is now more understandable and that your concerns have been addressed. Given this, we wanted to politely ask which reasons are causing the remaining reservation leading you to not mind if the paper is rejected (according to the score description).

---

### Official Review · Reviewer_JXGc · 2025-10-29

**Soundness:** 4
**Presentation:** 3
**Contribution:** 3
**Rating:** 6
**Confidence:** 3

**Summary:**

The paper introduces a numerically stable, chunkwise-parallel Mesa layer. At each timestep it uses a conjugate-gradient (CG) solver to compute optimal fast weights, yielding a recurrent alternative to attention with dynamically adjustable test-time compute.

**Strengths:**

1. The proposed method is well-motivated. It presents a numerically stable Mesa layer that solves $q^*_t=(H_t + \Lambda)^{-1} q_t$ per timestep via a CG solver with gated state recurrences, yielding a well-posed recurrent formulation.
2. Results are competitive with strong RNN-style baselines and broadly comparable to a Transformer at similar scale, and the accompanying analysis is generally sound.

**Weaknesses:**

1. The paper shows per-layer timings and training throughput (Fig. 2), but there’s no single, end-to-end table that reports latency (ms/token), tokens/s, and GPU memory alongside quality across CG step counts or the stopping policy, and across multiple context lengths on the same hardware.
2. It’s unclear when to prefer Mesa over MHA. Although the paper acknowledges that compute grows with CG steps and may exceed MHA past some step/key sizes, it lacks concrete recipes or tables mapping k/tolerance choices to context length, latency targets, and hardware, making deployment decisions difficult.
3. On mean-so-far long-context perplexity, a strong SWA-1024 baseline is often competitive or better, which tempers any blanket "long-range advantage" message. Please expand the long-context suite and report where Mesa wins/loses with matched budgets.

**Questions:**

Please see the concerns detailed in the Weaknesses.

---

> ### Author Response · Authors · 2025-11-21
> **Rebuttal**
>
> Thank you for your thoughtful and encouraging review that helped us to improve our work. We are glad to hear that you found our method to be well-motivated and appreciate our results & analysis.
>
> Below, we reply point-by-point to your questions and concerns:
>
> > The paper shows per-layer timings and training throughput (Fig. 2), but there’s no single, end-to-end table that reports latency (ms/token), tokens/s, and GPU memory alongside quality across CG step counts or the stopping policy, and across multiple context lengths on the same hardware.
>
> Thank you for bringing up this point. We report now in Figure 6 of the appendix the token throughput when fixing a certain sequence length, and varying the batch size, as well as token throughput when varying the sequence length when fixing the batch size. We report numbers, averaged over 10 runs, on a TPUv3 as well as an H100. For running the experiment in GPU we used the https://github.com/fla-org/flash-linear-attention open source implementation for the related work as well as a triton-based MesaNet implementation within the same framework.
>
> We observe that on both hardwares, our 1B MesaNet when compared to gated DeltaNet shows very competitive token throughput even with a fixed size of 30 CG steps per iteration. This highlights that although the MesaLayer consumes significantly more flops, the decode mode of these layers is deeply memory bound. Encouraged by the reviewer's question, we will move this figure to the main text of the paper and highlight these findings more in the main text.
>
> > It’s unclear when to prefer Mesa over MHA. Although the paper acknowledges that compute grows with CG steps and may exceed MHA past some step/key sizes, it lacks concrete recipes or tables mapping k/tolerance choices to context length, latency targets, and hardware, making deployment decisions difficult.
>
> Based on the reviews, we created a new Appendix Figure 6 for an overview of token throughput / second for the 1B MesaNet, GLA, Gated DeltaNet as well as a Transformer model. As we can see, the MesaNet (even with a fixed amount of 30 CG steps) performs competitively with the Gated DeltaNet using an equivalent architecture. GLA reaches significantly higher throughput throughout our scans, and Transformers (with global softmax) throughput deteriorate given longer sequence lengths, as expected.
> We therefore conclude that although the MesaNet consumes significantly more flops vs. Gated DeltaNet, (our strongest contender and used in e.g. stron open source models auch as Kimi, see https://huggingface.co/papers/2510.26692), token latency is very competitive.
>
> In terms of deployment decisions, we argue when aiming towards strong language modeling which includes recall benchmarks, global softmax Transformers or softmax-RNN hybrids, which we leave for future work, can not be matched by purely recurrent models yet, including the MesaNet.
> However, when only comparing fully recurrent models, we argue that MesaNet, also based on these new analyses, is a state-of-the-art model which should be considered.
> Thank you again for raising this important point - which we believe will significantly strengthen our work.
>
> > On mean-so-far long-context perplexity, a strong SWA-1024 baseline is often competitive or better, which tempers any blanket "long-range advantage" message. Please expand the long-context suite and report where Mesa wins/loses with matched budgets.
>
> Thank you for bringing up this important point. First of all, our paper, despite introducing and analyzing a scalable version of the Mesa layer at scale, tries to highlight that SWA models are strong baselines – easy-to-implement and easy-to-run baselines that are rarely reported in related work. In particular, we would like to highlight again that MesaNet is often the only model with superior performance compared to SWA-1024, in particular in in-context recall (see the aggregated performance results in the updated Table 3 – we switched to a table for better readability).
>
> Note however that all recurrent models we analyse do use less flops and memory than SWA-1024, as they store only key_size x key_size floating point values (with mesa 2 x key_size x key_size) where as the state size of SWA is 2 x key_size x 1024 - which is with common choices of key_size (128, 256) significant less. Nevertheless, we would like to stress again that SWA layers and its many variants should be a considered a strong baseline when studying RNN models which constant compute and memory wrt. sequence length.

---

### Official Review · Reviewer_oZt9 · 2025-11-01

**Soundness:** 3
**Presentation:** 4
**Contribution:** 3
**Rating:** 8
**Confidence:** 3

**Summary:**

The paper proposes MesaNet, a sequence layer that replaces recurrent fast-weight updates with a per-token optimal linear regression solved by conjugate gradients (CG). The layer maintains two matrix states
$G_t = \\sum_i \\zeta_{ti}, v_i k_i^\\top$ and $H_t = \\sum_i \\zeta_{ti}, k_i k_i^\\top$ (equivalently via gated recurrences $G_t=\\gamma_t G_{t-1}+\\beta_t v_t k_t^\\top, H_t=\\gamma_t H_{t-1}+\\beta_t k_t k_t^\\top$), and outputs $ o_t = G_t (H_t+\\Lambda)^{-1} q_t $ computed with a chunkwise-parallel CG solve implemented with GEMMs. Models at 140M/440M/1B on SlimPajama match or surpass strong linear-RNN baselines and are competitive with a transformer in average perplexity, with notably stronger early-sequence performance and solid results on some global-reasoning and few-shot tasks.

**Strengths:**

* Clear objective with a closed-form fast-weight map and practical CG solve; principled dynamic test-time compute via stopping criteria.
* Strong empirical controls (same backbone/tokenizer/data order) enabling clean comparisons.
* Useful diagnostics: early-sequence NLL gains, length extrapolation, grouped tasks; dynamic stopping achieves near-parity to larger fixed $k$ at substantially fewer steps on average.

**Weaknesses:**

* Missing a concise accuracy–efficiency summary at inference (for one representative long-sequence setting), including per-token latency and peak memory under fixed batch, hardware, and precision.
* Stability/conditioning analysis is qualitative; small ablations on the softplus scale for $\\Lambda$ and the diagonal preconditioner or $x_0$ initializer are needed to establish sensitivity and recommend defaults.

**Questions:**

1. Accuracy–efficiency summary (1B). Using the existing 1B checkpoint and kernels, please provide a compact table for a representative long sequence reporting validation perplexity, mean per-token latency, and peak memory for a few $k$ values (e.g., $0$ and two others) or dynamic stopping with $\\epsilon$ (report average CG steps). Also state hardware, precision, batch size, sequence length, and measurement procedure (warmup, repeats). Throughput optional.
2. Sensitivity/stability ablation. How sensitive are convergence and perplexity to the softplus scale on $\\Lambda$ and to the diagonal preconditioner or $x_0$ initializer? Please report CG steps to a fixed tolerance and validation perplexity, and provide recommended defaults.

---

> ### Author Response · Authors · 2025-11-21
> **Rebuttal**
>
> Thank you for the thoughtful and encouraging review that greatly improved our paper. We appreciate your positive assessment and are especially glad to hear that you appreciate our empirical controls and developed diagnostics.
>
> Below, we reply point-by-point to your questions and comments.
>
> > Accuracy–efficiency summary (1B). Using the existing 1B checkpoint and kernels, please provide a compact table for a representative long sequence reporting validation perplexity, mean per-token latency, and peak memory for a few values (e.g., and two others) or dynamic stopping with (report average CG steps). Also state hardware, precision, batch size, sequence length, and measurement procedure (warmup, repeats). Throughput optional.
>
> Inspired by the received reviews, we introduced a new Appendix Figure 6 for an overview of token throughput / second for the 1B MesaNet, GLA, gated DeltaNet as well as Transformer model. We ran all models with same precision as during training i.e. activation precision of bfloat16 as well as weight precision of float32.
> We scanned for all models the batch size as well as the sequence length before the softmax model ran out of memory. For obtaining the GPU numbers, we used a H100 GPU and the triton-based open source implementation of  https://github.com/fla-org/flash-linear-attention and the benchmarking code within, where we integrated our MesaNet which we will also release upon acceptance.
> For the TPU numbers, we used the TPUv5 which we also used during training. As we can see, the MesaNet (even with a fixed amount of 30 CG steps) performs competitively with the Gated DeltaNet using an equivalent architecture. GLA reaches significantly higher throughput throughout our scans, and Transformers (with global softmax) throughput deteriorate given longer sequence lengths, as expected.
> We therefore conclude that although the MesaNet consumes significantly more flops than gated DeltaNet, our strongest contender, latency is not affected by much. This points to the memory boundedness of all these layers in decode mode which MesaNet leverages to improve performance by spending more flops - without affecting latency significantly.
>
> > Sensitivity/stability ablation. How sensitive are convergence and perplexity to the softplus scale on and to the diagonal preconditioner or initializer? Please report CG steps to a fixed tolerance and validation perplexity, and provide recommended defaults.
>
> Thank you for this suggestion. Note that in the experiments we used softplus to restrict the lambdas, which are constant across sequence length, to be positive. Based on initial scans on our 400mio model, we found that introducing a lambda lower bound was crucial to stabilize the training. As shown in Figure 14 of the appendix, the lambda values, after training, span a large range from the lower bound (set to 0.25 in all of our experiments) up to values around 10 - while a large number of lambdas stay around their initial value 1.0.
> Furthermore, note that the max(lambdas)/min(lambdas) gives a proxy of the condition number of the matrix to be inverted, and we speculate that when allowing for low lambdas, below our lower bound, leads to convergence issues.
> Based on your suggestion, we ran some preliminary scans for which we report results for the 400mio MesaNet model here in text: We scanned the number of CG steps used during training (10, 20, 30, 40) as well as lambda lower values of 0.1 and 0.25 - we observe that for of CG steps lower than 30, we see that evaluation perplexity degrades significantly - 8% and 14% for 20 and 10 CG steps respectively. For more, 30 or 40, CG steps this degradation drops significantly under 1%. This shows that indeed the lower bound is an important stabilizing factor, directly linked to the number CG steps used for solving the underlying linear system.
> We will include a table for this important analysis in the appendix of the paper - thank you very much for suggesting it.
>
> The initial value used for lambda at the start of training (set to 1 in our experiments) on the other hand is significantly less sensitive, note that we set the lower value to 0.25 when scanning over the initial value. Here we see that large ranges (0.5, 1.0, 1.5, 2.0) lead to evaluation perplexity degradation of around 2% for the initial lambda value of 2.0, and no significant change otherwise.
>
> Finally, we would like to point the reviewer to section 5, and in particular Figure 5 where we ablated the Mesa Layer under two settings, including a dynamic stopping criterion (fixed tolerance). On the right hand side of the figure, we plot the regret to 30 CG steps (the default setting of the Mesa layer) with respect to mean-so-far NLL on SlimPajama validation set. Moreover, we report the average number of required CG steps until the dynamic stopping criterion is reached.
>
>
> Finally, we hope that we could clarify all your questions and concerns and remain open to answer any further questions that you may have.

---

### Official Review · Reviewer_MDkC · 2025-11-03

**Soundness:** 4
**Presentation:** 3
**Contribution:** 4
**Rating:** 8
**Confidence:** 3

**Summary:**

This work introduces a novel layer, called Mesa layer, guided by the theory of online learning/ test time training. The layer’s conception is similar to modern Linear RNNs/ SSMs like Gated DeltaNet, which allows for recurrent inference and chunk-wise parallel training. The difference lies in a principled online learning objective and, consequently, the recurrent state update rule, which features not just one, but two states and is computed via several steps of conjugate gradient descent. This yields competitive results in several benchmarks including large-scale LM with other models, such as the most related Gated DeltaNet. The authors conduct thorough theoretical analysis of their method and robust validation.

I’m convinced this work is a valuable contribution to the field of efficient alternatives to Transformer/ online learning models and vote for its acceptance.

**Strengths:**

First, I stress that I liked this paper a lot and I'm excited about the latest developments in the field of linear-time Transformer alternatives, and in particular the direction where this and related works are steering the field.

* The proposed method (online learning objective, corresponding update rule, chunk-wise and recurrent implementation algorithms) is novel and original.

* The paper is very rich in content, be it new theory, connections to other literature, and especially experiments.

* The scope of the experiments and ablations is very vast and comprehensive, to the extent that it leads to important and valuable observations about the other models and the field of linear alternatives to Transformers themselves (see e.g., summary in lines 81-84).

* The paper is self-contained. For example, Appendix J explains the background from previous work leading to Mesa layer, so it’s not required to be familiar with it beforehand.

* I’d like to highlight specifically that the paper provides a detailed description of the pre-training settings and hyperparameters, including e.g., dataset preparation, which allows for greater transparency and reproducibility.

**Weaknesses:**

The below comments do not represent major weaknesses and don’t denigrate the quality of the paper. They aim for improvement of presentation or consider the pieces of text which the authors could just omit without detriment of the exposition.

1. Please state in the beginning of the paper that all vectors are column vectors to avoid confusion because many related works such as GLA or Gated DeltaNet use row vectors as a convention.

2. Lines 121-123: How do you derive the expression $\gamma_t \Phi_{t-1} + \beta_t v_t k_t^T$ out of equation 3? Also, the statement on line 127 (“we recover DeltaNet”) is not supported.
Please provide full derivations or references to a specific place in the paper(s) where the statements are proved. In the same vein, could you describe how we get update rule (equation 5) out of online loss (equation 4)?

3. Among several recent algorithms specifically designed with test-time training/ online optimization in mind, the paper discusses in depth only (Gated) DeltaNet variants, both theoretically (lines 1577-1603) and empirically. It would be great to explain in more detail MesaNet’s differences and similarities with such algos as Test-Time Training (https://openreview.net/forum?id=wXfuOj9C7L, not mentioned in the paper), LongHorn (https://openreview.net/forum?id=8jOqCcLzeO), Titans (https://openreview.net/forum?id=8GjSf9Rh7Z), Atlas (https://arxiv.org/abs/2505.23735), and Miras (https://arxiv.org/pdf/2504.13173). Lines 145-148 briefly touch on some differences, but further corroboration is required. E.g., could you provide formulas demonstrating why Atlas “corresponds exactly to a sliding-window variant of the Mesa layer”? Additionally, if you have the bandwidth and capacity, would it be possible to pre-train one of these related models and add it to comparisons (although I doubt that it would be significantly different in performance)?

4. Also, It would be informative to include a table with comparison of online-learning objectives for different models, similar to the Table 4 in LongHorn paper (https://openreview.net/forum?id=8jOqCcLzeO).

5. Lines 1452-1453 – The remark about Mamba 2 being non-gated is incorrect. Mamba 2 is gated in the same sense as GLA ($S_t = G_t * S_{t-1}$), but it calls the same mechanism “decay” and uses notation $A$ instead of $G$.

6. There are no code listings for the new layer in the paper, nor accompanying code archive in the submission. The paper would benefit from an additional appendix with a high-level implementation of the core algo in PyTorch or Jax.

7. I may have overlooked that but it seems the $O(T)$ training and $O(1)$ inference complexities aren't explicitly stated in the paper.

**Questions:**

See section “weaknesses”. Also:

1. I’m curious why the regularization term in equations 2 and 4 is $\Phi^{\top} \Lambda \Phi$ and not $\Phi \Lambda \Phi^{\top}$?

2. What are your thoughts on why MesaNet becomes significantly superior to Gated DeltaNet on in-context recall tasks from Table 14 after prolonged training (50B tokens vs 15B)?

3. Does MesaNet indeed have linear computational complexity w.r.t. sequence length T? Lines 449-450 highlight “the need for higher number of steps as t grows” indicating that it may not be the case.

4. Lines 1399-1403: Do new sequences always start with fresh documents or new documents can cross sequence boundaries? I.e. if document A has more tokens than 2048 and therefore cannot be fitted into  sequence 1, will the next sequence 2 begin with the leftover part from document A?

5. Do Transformer and SWA variants used in the experiments have any differences in architecture from LLama 2? Specifically, do the models employ RoPE?

6. Why can't you use Woodbury identity for exact matrix inverse in chunk-wise parallel form instead of costly and approximate conjugate gradient descent steps?

7. Will you release the data preparation/ configs/ pre-training code and, importantly, the weights of different models you've pre-trained for experiments in your paper? That would be greatly beneficial for the advancement of the field and provide very valuable playground and baselines for future research.

Edit: Upon second thought, I see that Woodbury identity (https://en.wikipedia.org/wiki/Woodbury_matrix_identity) in Q6 wouldn't provide any benefits because the problem of finding inverse of $d \times d$ matrix translates into finding inverse of $N \times N$ matrix where d is head dimension and N is sequence length. However, I'm still not sure why you don't use exact matrix inverse algorithm with readily available PyTorch or CuBLAS methods (or some JAX/ TPU alternatives). For dense matrices, both conjugate gradient and ordinary exact matrix inverse methods have the same $O(d^3)$ complexity. Have you considered comparing speed of an exact method against your implementation with a commonly used head size d=128?

---

> ### Author Response · Authors · 2025-11-21
> **Rebuttal - Part 1**
>
> Thank you for the thoughtful, in-depth review that greatly improved our paper. We are very grateful for this high-quality review which we will gladly acknowledge in the paper. Moreover, we are very grateful for the expressed appreciation of our work – we really appreciate it!
>
> We reply point-by-point to your questions and comments.
>
> > Please state in the beginning of the paper that all vectors are column vectors to avoid confusion because many related works such as GLA or Gated DeltaNet use row vectors as a convention.
>
> We added a remark at the beginning of section 2 (highlighted in red).
>
> > Lines 121-123: How do you derive the expression $\gamma_t \Phi_{t-1} + \beta_t v_t k_t^T$ out of equation 3? Also, the statement on line 127 (“we recover DeltaNet”) is not supported. Please provide full derivations or references to a specific place in the paper(s) where the statements are proved. In the same vein, could you describe how we get update rule (equation 5) out of online loss (equation 4)?
>
> Thank you for pointing this out. We added a new Appendix B to the updated manuscript detailing the derivations of the GLA and DeltaNet update rules as online gradient-based learning dynamics (together with LongHorn and Atlas, as per your next question). We also included a new step-by-step derivation of the Mesa layer forward pass dynamics in Appendix D.
>
> > Among several recent algorithms specifically designed with test-time training/ online optimization in mind, the paper discusses in depth only (Gated) DeltaNet variants, both theoretically (lines 1577-1603) and empirically. It would be great to explain in more detail MesaNet’s differences and similarities with such algos as Test-Time Training (https://openreview.net/forum?id=wXfuOj9C7L, not mentioned in the paper), LongHorn (https://openreview.net/forum?id=8jOqCcLzeO), Titans (https://openreview.net/forum?id=8GjSf9Rh7Z), Atlas (https://arxiv.org/abs/2505.23735), and Miras (https://arxiv.org/pdf/2504.13173). Lines 145-148 briefly touch on some differences, but further corroboration is required. E.g., could you provide formulas demonstrating why Atlas “corresponds exactly to a sliding-window variant of the Mesa layer”? Additionally, if you have the bandwidth and capacity, would it be possible to pre-train one of these related models and add it to comparisons (although I doubt that it would be significantly different in performance)?
>
> We comment on the changes point-by-point below:
> We added a citation for TTT in the main text, section 2. We also expanded the discussion on Appendix A about this method.
> We included a more detailed discussion (including equations) of the LongHorn and Atlas/Omega models in the new Appendix B.
> Besides the Appendix A discussion, we now also cite and discuss Titans in the main text, section 2.
> We also added a citation for Miras alongside Wang et al., as a unifying framework for sequence modeling as test-time training.
>
> Unfortunately, we were not able to run new experiments with these models for the rebuttal due to time and capacity constraints. We would also like to note that no official code for Titans and Atlas is publicly available.
>
> > Also, It would be informative to include a table with comparison of online-learning objectives for different models, similar to the Table 4 in LongHorn paper (https://openreview.net/forum?id=8jOqCcLzeO).
>
> Thank you for this suggestion. Please see Table 2 in the new Appendix B.
>
> > Lines 1452-1453 – The remark about Mamba 2 being non-gated is incorrect. Mamba 2 is gated in the same sense as GLA ($S_t = G_t * S_{t-1}$), but it calls the same mechanism “decay” and uses notation $A$ instead of $G$.
>
> Thanks for pointing this out. We acknowledge that the original text was confusing, as it referred to the absence of a write/input gate (as per Table 4, Mamba2 vs. GLA). We fixed this in the updated manuscript.
>
> > There are no code listings for the new layer in the paper, nor accompanying code archive in the submission. The paper would benefit from an additional appendix with a high-level implementation of the core algo in PyTorch or Jax.
>
> Thank you for pointing out this crucial point. Upon acceptance, we will provide an open source triton-based implementation in PyTorch as well as in Jax. To allow the community to quickly build upon our work. Furthermore, we will provide a Colab Notebook where we reproduce some of our simple experiments to further provide code to try our model as a drop in replacement for softmax or Gated DeltaNet.
>
> > I may have overlooked that but it seems the $O(T)$ training and $O(1)$ inference complexities aren't explicitly stated in the paper.
>
> We added a remark in the end of Section 2.
>
> > I’m curious why the regularization term in equations 2 and 4 is $\Phi^{\top} \Lambda \Phi$ and not $\Phi \Lambda \Phi^{\top}$?
>
> Very good catch – this is a typo on our side! We are very grateful that you spotted this – we fixed it in the updated manuscript.

---

> > ### Author Response · Authors · 2025-11-21
> > **Rebuttal - Part 2**
> >
> > > What are your thoughts on why MesaNet becomes significantly superior to Gated DeltaNet on in-context recall tasks from Table 14 after prolonged training (50B tokens vs 15B)?
> >
> > While numbers at low token regimes might be noisy due to undertraining, we hypothesize that MesaNet requires more data until the superior in-context recall abilities kicks in, due to higher model complexity compared to Gated DeltaNet.
> >
> > > Does MesaNet indeed have linear computational complexity w.r.t. sequence length T? Lines 449-450 highlight “the need for higher number of steps as t grows” indicating that it may not be the case.
> >
> > When fixing the number of CG-step per token this is naturally the case. When we use the stopping criterion, indeed the computational complexity is dependent on how fast the solver converges (at an exponential rate), which is indeed sequence and therefore potentially time dependent. Note however, that we in practice train the model with a fixed number of CG steps (30 in our case) and therefore expect that the model, optimized for this number of steps, will consume at most the CG steps using training in a meaningful way. This is also what we observe in practice as we show in section 5 - where we see that on average already 10 CG steps, when using the stopping criterion, are sufficient to match the performance of the model trained with 30 CG steps.
> >
> > > Lines 1399-1403: Do new sequences always start with fresh documents or new documents can cross sequence boundaries? I.e. if document A has more tokens than 2048 and therefore cannot be fitted into sequence 1, will the next sequence 2 begin with the leftover part from document A
> >
> > We pre-tokenize our datasets, add BOS tokens at the start of documents, and pack them to sequence length. We also reset the states for all models based on the BOS tokens, when multiple sequences are packed together. We will clarify these points in the experimental details. Thank you for bringing up this important point.
> >
> > > Do Transformer and SWA variants used in the experiments have any differences in architecture from LLama 2? Specifically, do the models employ RoPE?
> >
> > We clarified in section 4 (“Setup” paragraph) that both models use RoPE.
> >
> > > Why can't you use Woodbury identity for exact matrix inverse in chunk-wise parallel form instead of costly and approximate conjugate gradient descent steps? Edit: Upon second thought, I see that Woodbury identity (https://en.wikipedia.org/wiki/Woodbury_matrix_identity) in Q6 wouldn't provide any benefits because the problem of finding inverse of  d x d matrix translates into finding inverse of N x N matrix where d is head dimension and N is sequence length. However, I'm still not sure why you don't use exact matrix inverse algorithm with readily available PyTorch or CuBLAS methods (or some JAX/ TPU alternatives). For dense matrices, both conjugate gradient and ordinary exact matrix inverse methods have the same
> > complexity. Have you considered comparing speed of an exact method against your implementation with a commonly used head size d=128?
> >
> > Thank you for this interesting point. We have not investigated this as we build on our inside that well established chunkwise parallel training strategies and be readily used to compute the CG algorithm in parallel. Essentially, we execute numerous GLA steps, while loading the memory only ones. We nevertheless find this idea very interesting as, at least during encoding, we could invoke these algorithms/implementations on trained models but leave this investigation for future work. As also shown in our new appendix Figure 6, compared to our strongest contender Gated DeltaNet, we perform very competitive for train as well as inference throughput. Thank you again for bringing up this interesting point.
> >
> > > Will you release the data preparation/ configs/ pre-training code and, importantly, the weights of different models you've pre-trained for experiments in your paper? That would be greatly beneficial for the advancement of the field and provide a very valuable playground and baselines for future research.
> >
> > We agree with the reviewer on this point and aim to release everything upon acceptance of the paper.
> >
> >
> > Please let us know if you have any remaining concerns or clarifying questions. Thank you again for your very insightful feedback.

---

### Author Response · Authors · 2025-11-21
**General Response (Rebuttal)**

We would like to thank all the reviewers for their time and effort putting together these encouraging and helpful reviews. We are truly grateful for the many insightful comments that helped us to greatly improve our manuscript. We will gladly acknowledge the contribution of the anonymous reviewers in the final acknowledgements of our work.

Based on your comments and suggestions, we ...
- clarified our presentation which hopefully leads to a clearer and more accessible introduction of our method
- added an additional token throughput comparison to related work
- extended the related work discussion (including detailed derivations)

Please see the updated PDF where we highlighted all relevant changes in red. We remain open for any further questions and suggestions the reviewers might have. Thank you again!

---

### Author Response · Authors · 2025-12-03
**Thank you!**

We thank all reviewers and for the time put into their reviews and the valuable feedback. We are thankful for the general positive feedback and helpful suggestions which improved our study substantially. We will gladly acknowledge the contribution of the anonymous reviewers in the final acknowledgements of our work.

Below, we shortly summarize (i) the concerns raised by the reviewers, (ii) our rebuttal response and changes we made to our work to address them, and (iii) the reaction of the reviewers to each point.

**Improved presentation and accessibility of our work.** Based on inputs of the reviewers, we did multiple changes to make our work more accessible and improved readability:
- We significantly improved section 2 (i.e., introduction of the parallelizable mesa layer)
- Moved multiple equations from the appendix into the main text to make the flow more complete
- Change the performance figure to a more accessible performance table (see table 3)
These changes were appreciated by reviewer mKy1 (the only reviewer that replied before the reviewer communication was locked)

**Contextualization with related work.** We extended our related work section in the appendix, see B and D, which in depth introduces and contrasts related methods to our novel scalable Mesa layer. Again these changes addressed the concerns of reviewer mKy1.

**Improved Runtime / Throughput Analyses:** Based on the suggestions of multiple reviewers, we added a thorough runtime analysis in terms of various token throughput plots, see Appendix section G.7 and Figure 6. We show that although the Mesa layer does consume significantly more flops than state-of-the-art layers such as Gated DeltaNet, its token-throughput/second numbers are very competitive across a wide variety of settings. We conclude that this highlights the memory boundedness of the recurrent layers during generation which is successfully leveraged by the MesaNet by improving performance while not increasing wall-clock-time significantly.

---

### Meta-Review · Area_Chair_9qQS · 2026-01-07

**Summary:**

The paper develops MesaNet a sequence mixer for autoregressive language models that sets up an in-context regression problem for the output of a layer. The in-context regression is parameterized over the sequence by two recurrences. The result is a chunkwise parallel sequence mixer that uses constant memory and bounded compute depending on the number of conjugate gradient iterations used/needed to solve the in context regression. The majority of reviewers liked the paper with questions around the presentation and clarity on why this approach over others. Still given the extensive results and it being a relatively fresh direction for scalable test time optimization that's scalable, I recommend acceptance.

**Reviewer Concerns:**

MDkC - related work and presentation of math

oZt9 - concise difference and improvements, stability and conditioning analysis

JXGc - missing concise main table, unclear on when to use it, how performance looks on matched budgets of some form

mKy1 - presentation and motivation and missing baselines

**Reviewer Scores:**

All the reviewers would keep their score as the main common weakness of this paper is clarity. I took a look and even with the reviews as a guide, it's a bit all over the place. The results, though, are comprehensive.

---

### Decision · Program_Chairs · 2026-01-26

Accept (Poster)